**Registered report**

# Bayesian evaluation of diverging theories of episodic and affective memory distortions in dysphoria

**Sascha B. Duken** [1] ✉, **Liza Keessen**[2], **Herbert Hoijtink** [3], **Merel Kindt** [1] & **Vanessa A. van Ast** [1] ✉

People suffering from dysphoria retrieve autobiographical memories distorted in content and affect, which may contribute to the aetiology and maintenance of depression. However, key memory difficulties in dysphoria remain elusive because theories disagree how memories of different valence are altered. Here, we assessed the psychophysiological expression of affect and retrieved episodic detail while participants with dysphoria (but without a diagnosed mental illness) and participants without dysphoria relived positive, negative, and neutral memories. We show that participants with dysphoria retrieve positive memories with diminished episodic detail and negative memories with enhanced detail, compared to participants without dysphoria. This is in line with negativity bias but not overgeneral memory bias theories. According to confirmatory analyses, participants with dysphoria also express diminished positive affect and enhanced negative affect when retrieving happy memories, but exploratory analyses suggest that this increase in negative affect may not be robust. Further confirmatory analyses showed that affective responses to memories are not related to episodic detail and already present during the experience of new emotional events. Our results indicate that affective memory distortions may not emerge from mnemonic processes but from general distortions in positive affect, which challenges assumptions of memory theories and therapeutics. Protocol registration: The Stage 1 protocol for this Registered Report was accepted in principle on the 18rd of March 2021. The protocol, as accepted by the journal, can be found at https://doi.org/10.6084/m9.figshare.14605374.v1.

Autobiographical memory allows us to remember personally experienced events from our lives[1]. By mentally traveling back in time, autobiographical memory retrieval can evoke the feeling of reliving an event[2], and re-elicit affective responses[3,4] that guide beviour[5–8]. For instance, remembering positive events can improve mood[3,9–11], decrease subjective stress and cortisol levels[3], and provide confidence when facing adverse events[3,5,6]. Remembering negative events can induce emotions such as sadness and anger, but can also help to avoid rejection and failure in the future[5,6]. However, when autobiographical memory processes become distorted, they can play an important role in the etiology and maintenance of psychiatric disorders, in particular dysphoria – a persistent negative mood – and clinical depression[12–16].

[1]Department of Clinical Psychology, University of Amsterdam, Amsterdam, the Netherlands. [2]Amsterdam School of Communication Research, University of Amsterdam, Amsterdam, the Netherlands. [3]Department of Methodology and Statistics, Utrecht University, Utrecht, the Netherlands. ✉e-mail: sascha_duken@web.de; v.a.vanAst@uva.nl

Specifically, whereas healthy individuals retrieve positive memories with ease[17-20], individuals with dysphoria struggle to remember positive events[1,21,22]. Individuals with dysphoria also tend to experience blunted positive emotions during retrieval[4,23], and cannot use positive memories to repair a negative mood[4,9,23,24]. Additionally, they tend to recollect sombre memories and have difficulties remembering specific events in rich episodic detail[1,25-27]. Such autobiographical memory distortions can predict the course of depression[13], and are associated with other vulnerability factors such as rumination[28], negative self-cognitions[21], and heightened morning cortisol levels[21]. These findings suggest that autobiographical memory distortions are not merely epiphenomena of depression, but might represent a key vulnerability factor that contributes to the etiology and maintenance of clinical depression[12,14].

Given the central role of autobiographical memory distortions in dysphoria and depression, recent advances in cognitive, affective, and clinical sciences have aimed to understand and change maladaptive memory processes to improve the prevention and treatment of these mental health conditions[14,29-31]. However, most studies investigated declarative memory components (e.g., episodic detail or memory accuracy)[32-34], even though affective responses during autobiographical memory retrieval are thought to be most crucial in the etiology of depression[30]. Studies that did investigate affective responses mostly assessed self-reported subjective feelings[9,24,35,36], which are not only susceptible to expectancy and demand effects[12,36-38], but can even alter the affective response under investigation[39]. Furthermore, two prominent theoretical frameworks in the field, overgeneral memory bias[1,30] and negativity bias theories[16,40,41], as well as emerging memory therapies that aim to target mood disturbances by editing memory content[14,31,34], assume a critical relationship between retrieved episodic detail and intensity of experienced affect during the reliving of memories. However, previous research on autobiographical memory distortions focused on isolated aspects of memory retrieval[4,33,34], precluding more overarching insights into dysphoric memory distortions and their interrelationships. Consequently, it is not clear whether affective distortions co-occur with episodic distortions. Finally, even though overgeneral memory bias[1,30] and negativity bias theories[16,40,41] have since long aimed to reconcile and explain the isolated observations derived from empirical studies, these two frameworks make diverging predictions on how dysphoric distortions are expressed in positive versus negative memories. With this registered report, we aimed to overcome these challenges. We developed the "Re-experience Of Autobiographical Memories" task (ROAM, see Fig. 1) that allows the assessment of the psychophysiological expression of affect as well as episodic detail during memory retrieval. We used Bayesian Informative Hypothesis Testing to evaluate the strength of evidence for diverging and converging predictions derived from overgeneral memory bias[1,30] and negativity bias theories[16,40,41] with regard to dysphoric distortions in episodic detail and affective responses, and their interrelationship.

For several decades, overgeneral memory bias theories have been very influential in depression research. These theories emphasize that individuals with dysphoria experience difficulties in accessing specific episodic details of positive and negative autobiographical memories[1,25,42]. When remembering their past, they tend to retrieve overgeneral memories that comprise a category of similar events (e.g., "When I listen to music") rather than events that happened at a specific day and time (e.g., "When I went to that concert in my home town last October"). Overgeneral memories lack episodic detail, which prevents a strong sense of reliving[30]. Consequently, such memories might not be powerful enough to elicit strong affective responses[30,43]. Potential mechanisms that drive overgeneral memories have been formulated on several levels of analysis. From a cognitive perspective, overgeneral memory retrieval might be a result of functional avoidance, where individuals with dysphoria try to evade strong emotions elicited by

detailed memories[1]. From a biological perspective, overgeneral memory retrieval might be the result of an impoverished ability to distinguish similar memories due to reduced hippocampal functioning[44-46]. Regardless of the level of analysis, these perspectives converge on the hypothesis that when displaying an overgeneral memory bias, individuals with dysphoria retrieve positive and negative memories with reduced episodic detail. As a consequence, they experience diminished affect in response to both positive and negative memories.

Negativity bias theories offer an alternative perspective on memory distortions in depression[16,32,41]. These emphasize that individuals with dysphoria tend to have negative world views that bias their attention, perception, and interpretation towards negative information[16,32,40,41]. Furthermore, a persistent negative mood renders negative information more accessible[22,32,47]. Individuals with dysphoria are thus more likely to retrieve and elaborate on negative than positive memories[20,32,47]. This is thought to lead to increased negative affect during negative memory retrieval, and decreased positive affect during positive memory retrieval. For example, if individuals with dysphoria remember going to a concert, they might remember the general event, but, unlike healthy individuals, they will have problems accessing specific episodic details and reactivating the positive affect that were part of the original event. Thus, like overgeneral memory bias accounts, negativity bias theories propose that individuals with dysphoria retrieve positive memories with reduced episodic detail and diminished positive affect. However, negativity bias theories state that they retrieve negative memories with enhanced episodic detail and enhanced negative affect.

Both the overgeneral memory bias[1,30,43] as well as the negativity bias[12,14,35] frameworks assume a critical link between retrieved episodic detail and affective responses to a memory. Inspired by this theoretical tenet, emerging memory interventions such as memory specificity training[34] or method of loci[14,35] rest on the core assumption that enhanced retrieval of episodic details leads to stronger affective responses. However, this assumption remains to be critically tested, as most previous studies focused on the accessibility of episodic details without assessing affective responses[1,22,31,32,34,47], or used subjective measurements in healthy participants[11,43,48]. While it is difficult to establish the direction of causality in the assumed relationship between episodic detail and memory emotionality, it is important to critically test whether such a relationship exists in the first place. Consequently, we will investigate whether increased retrieval of episodic detail is associated with stronger affective responses. However, it is possible that dysphoria disturbs the link between episodic detail and affective responses. Thus, even when individuals with dysphoria retrieve a positive memory with rich episodic detail, the memory might nonetheless fail to elicit strong affective responses. This could reflect a trait-like difference between individuals, representing a marker of vulnerability to depression[13,21]. Therefore, we will test whether the relationship between episodic detail and affective response differs between individuals with and without dysphoria.

In order to test our hypotheses, it is essential to assess both episodic memory detail and affective responses during autobiographical memory retrieval. Previous studies of emotional autobiographical memory retrieval have relied heavily on self-report as the sole measure of affective responses[9,24,35,36]. Even though self-reports of subjective feelings provide an important source of information, they are prone to experimental biases such as expectancy and demand effects[49-52]. More importantly, self-reports require awareness, explicit categorization, and labeling of affect, which can alter the affective response under investigation[39]. Therefore, we have developed the ROAM that allows us to assess multiple read-outs of personally cued autobiographical memories, including the psychophysiological expression of affective responses, episodic memory detail, and self-reported feelings (Fig. 1a). Specifically, we employed facial electromyography (fEMG) of the

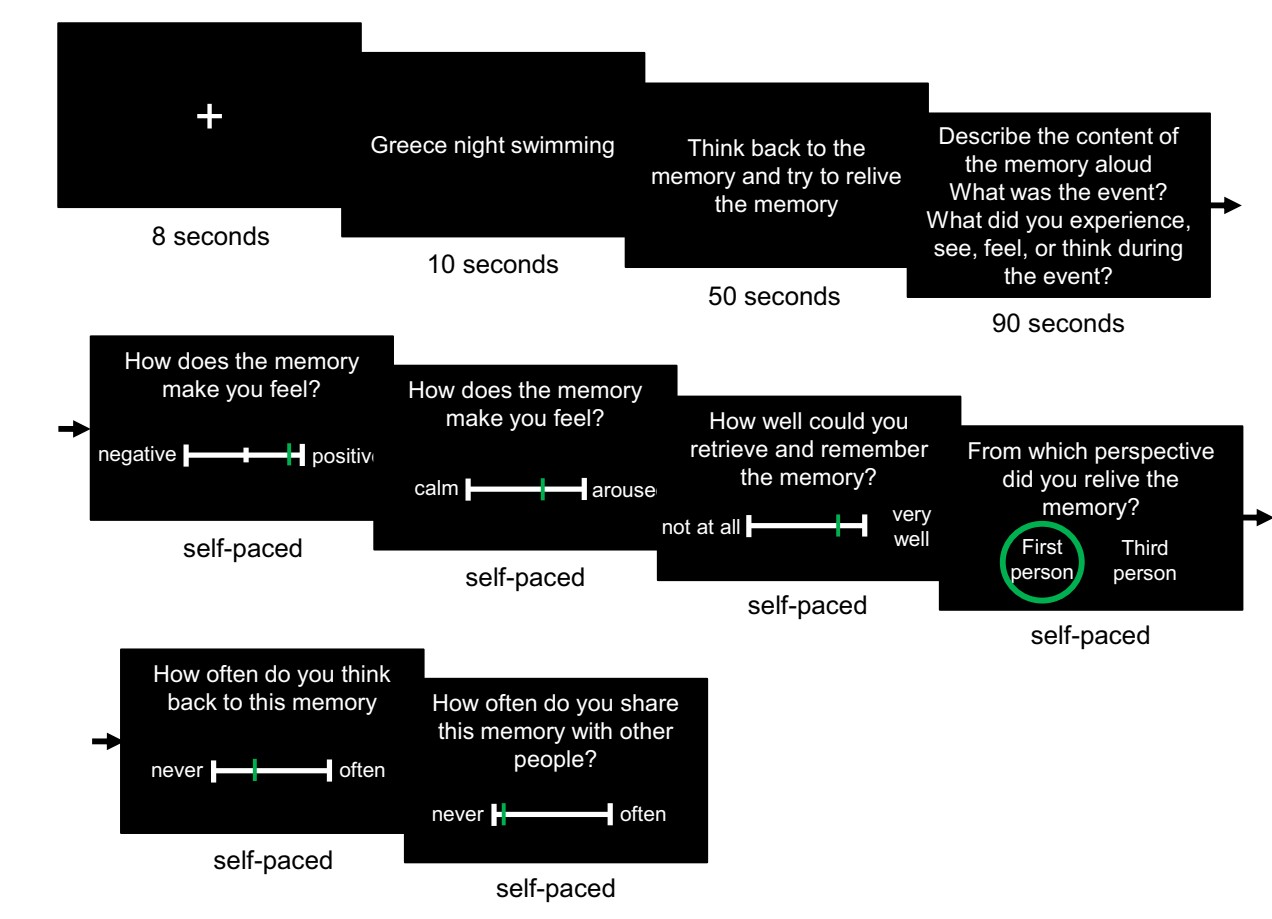

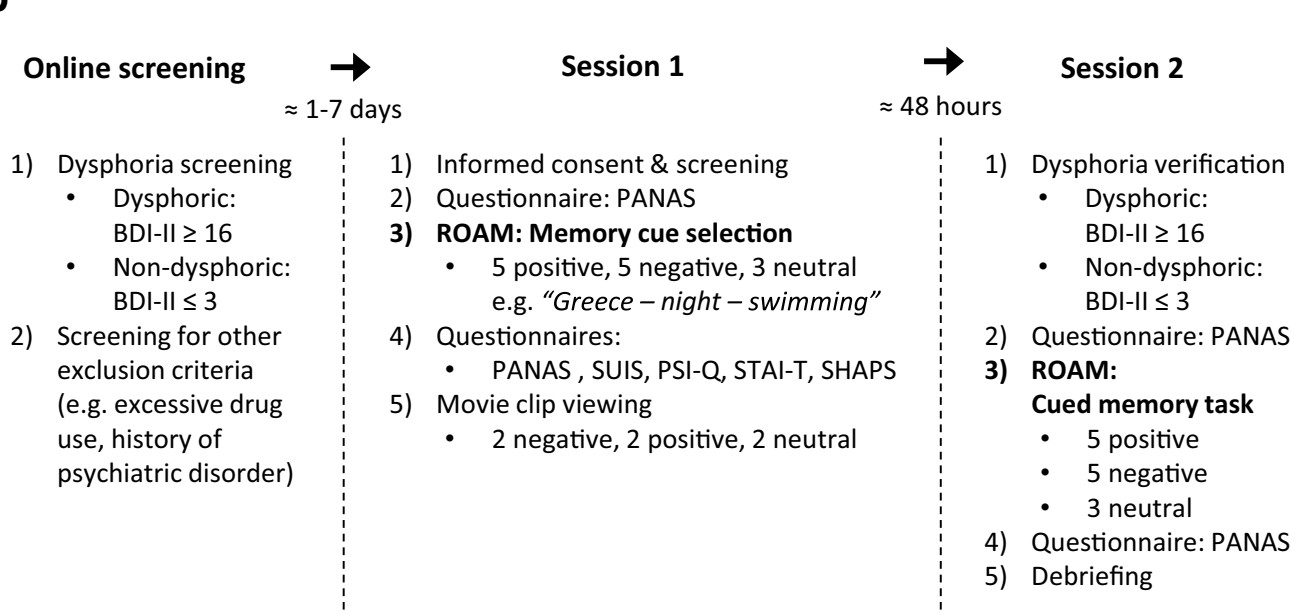

Fig. 1 | The Re-experience Of Autobiographical Memories task (ROAM) and the general procedure. a presents a schematic overview of the different computer screens and their timings presented during a cued memory trial in the ROAM during session 2. b presents a schematic overview of the complete experimental procedure including online-screening, Session 1, and Session 2. BDI-II Beck's Depression Inventory modified[92], PANAS Positive and Negative Affect Scale[93], SUIS Spontaneous Use of Imagery Scale[94], PSI-Q Plymouth Sensory Imagery Questionnaire[95], STAI-T Spielberger Stait-Trait Anxiety Inventory, trait subscale[96], SHAPS Snaith–Hamilton Pleasure Scale[97].

zygomaticus major and the corrugator supercilii regions to quantify affective responses during memory retrieval. The zygomaticus contracts to produce smiling and indicates positive affect, whereas the corrugator contracts to produce frowning and indicates the negative affect[49,50]. Facial EMG responses are thought to reflect relatively automatic affective responses[39,53–55] and may therefore be less influenced by higher-order cognitive functions and knowledge about one's self, unlike self-reports. Moreover, the strength of fEMG responses reflects variations in affect intensity, as suggested by studies that experimentally manipulated affect intensity[54,56,57] or examined naturally occurring variations in experienced affect[49,50,53]. Importantly, such variations in affect intensity can be measured both within and between participants[49,50,53,54]. The observation that fEMG can pick up even small variations in affect intensity within an individual indicates that it is possible to investigate potential relationships of expressed affect and episodic detail of individual memories. Furthermore, fEMG can be used to investigate distorted affective processes in dysphoria and depression[4,23,58,59]. Specifically, individuals with and without dysphoria showed reduced zygomaticus activity compared to healthy controls when processing positive information, but normal or even enhanced corrugator activity when processing negative information[4,58–60]. We used fEMG responses as the primary and self-reported subjective feelings as a complementary outcome measure for affective responses. Correspondingly, the conclusions from this study are mainly guided by the results of the fEMG analyses, but the insights from the self-reports help to provide a more comprehensive and nuanced insight into affective memory distortions[26,27].

In the ROAM, participants silently relive personal memories, during which fEMG data is collected. Following this, participants verbally recall the event while being audio-recorded, which allows subsequent coding of episodic memory detail[33,61,62]. After the verbal recall, participants rate their subjective experience of the memory for example in terms of emotional valence and arousal. Extensive piloting of the ROAM (see Methods and Supplementary Note 1, 2, and 3) revealed that positive autobiographical memory retrieval elicits zygomaticus activity, whereas negative autobiographical memory retrieval elicits corrugator activity. We also found initial evidence that episodic detail might indeed be positively associated with affective responses to a specific positive memory. These observations demonstrate that the ROAM allows the investigation of episodic detail and the psychophysiological expression of affect during autobiographical memory retrieval, as well as their relationship.

To recap, we collected new data with the ROAM to characterize episodic detail of and affective responses to positive and negative autobiographical memories in individuals with and without dysphoria (Manipulation checks). We employed Bayesian Informative Hypothesis Testing (BAIT) that allowed us to quantify evidence for competing hypotheses[63]. Evidence is presented as Bayes Factors (BF) and Posterior Model Probabilities (PostP). Bayes Factors indicate how much more likely it is to observe given data under one hypothesis relative to another hypothesis[63]. Posterior Model Probabilities quantify the support in the data for one hypothesis relative to one or more other hypotheses under investigation[63]. Using this analytical strategy, we assessed which theoretical framework best explains dysphoric distortions of episodic detail (Test 1 2) and affective responses (Test 2A and 2B). Overgeneral memory bias theories predict that individuals with dysphoria retrieve fewer episodic detail (Test 1 – H1) and concurrently experience diminished affect (Test 2A – H1) when reliving positive memories as well as when reliving negative memories, compared to individuals without dysphoria. Negativity bias theories predict that individuals with dysphoria retrieve fewer episodic detail when reliving positive memories, but more episodic detail when reliving negative memories (Test 1 – H2). Concurrently, they experience diminished affect when reliving positive memories, but enhanced affect when reliving negative memories (Test 2A – H2), compared to individuals

without dysphoria. Test 1 and Test 2A therefore addressed our primary research objective to evaluate evidence for competing predictions of overgeneral memory and negativity bias theories regarding episodic memory detail and affective responses to memories. The secondary objectives of our study were to determine whether any observed affective distortions are indeed specific to memories (Test 2B) and to test the critical theoretical assumption that episodic detail correlates with affect-increases (Test 3). Specifically, altered affective responses to autobiographical memories might not be specific for the re-experience of memories, but reflect general emotional or cognitive disturbances[12,64,65]. In this case, the affective distortions during autobiographical memory retrieval should already be present during the experience of novel events[12]. Therefore, we investigated whether individuals with dysphoria showed the same pattern of affective responses to new events (movie clips) as for autobiographical memories (Test 2B – H1) or a different pattern (Test 2B – Hc). Finally, we assessed whether retrieved episodic detail of a given memory was positively associated with the affective response (Test 3). For both negative and positive memories, we expected that within individuals, episodic detail would positively predict the affective response on a trial-by-trial basis. We further tested if this relationship was different for individuals with and without dysphoria.

## Results

### Participants

A total of 137 participants attended the first session of the study after being included based on the online screening. Three participants did not return for the second session and we excluded 54 participants that did not meet all inclusion criteria after participation (e.g., because their BDI score changed between online screening and the second session; see Supplementary Note 4, Supplementary Fig. 5, for an overview of the screening and inclusion procedure). The final sample consisted of $n = 40$ participants in the non-dysphoric group and $n = 40$ participants in the dysphoric group ($N = 80$). We employed Bayesian Updating and planned to collect data until there was convincing evidence for one hypothesis relative to all other hypotheses under investigation (PostP$_i \geq 0.80$) for each of the primary research questions (Test 1 and Test 2A), or until we reached a maximum sample size of $N = 80$. Supplementary Note 5 (Supplementary Fig. 6) depicts evidence with increasing sample size. Evidence in Test 1 (episodic detail) exceeded the stopping criterion already at the minimum sample size. However, since evidence in Test 2A (affective expressions) remained slightly below the stopping criterion, we continued data collection until reaching the maximum sample size. The groups were matched by self-reported sex and age (non-dysphoric: 39 self-reported female, $M_{age} = 22.07$, $SD_{age} = 3.66$; dysphoric: 39 self-reported female, age: $M_{age} = 21.50$, $SD_{age} = 4.19$). We did not conduct separate analyses or comparisons depending on self-reported sex because there was too little data on participants reporting to be male, and because the pre-registered analysis plan did not include such analyses. In line with our sampling plan, the two groups differed in depressive symptoms (non-dysphoric: $M_{BDI} = 1.25$, $SD_{BDI} = 1.19$, range$_{BDI} = 0–3$; dysphoric: $M_{BDI} = 24.15$, $SD_{BDI} = 5.67$, range$_{BDI} = 16–39$), but also in terms of trait anxiety (non-dysphoric: $M_{STAI-T} = 32.03$, $SD_{STAI-T} = 5.75$; dysphoric: $M_{STAI-T} = 54.59$, $SD_{STAI-T} = 6.30$), and anhedonia (non-dysphoric: $M_{SHAPS} = 0.55$, $SD_{SHAPS} = 0.88$; dysphoric: $M_{SHAPS} = 2.33$, $SD_{SHAPS} = 2.22$).

### Data quality checks

Descriptive statistics on the phenomenology of positive, negative, and neutral autobiographical memories retrieved by participants with and without dysphoria are presented in Table 1. Episodic detail of memories was assessed by three independent raters that coded the amount of internal episodic details during the verbal recall of autobiographical memories. In line with the analysis plan, 25% of the memories were

**Table 1 | Descriptive statistics on phenomenology of autobiographical memories and emotional movie clips**

| | | Non-dysphoric | | | Dysphoric | | |
|---|---|---|---|---|---|---|---|
| | | Negative | Neutral | Positive | Negative | Neutral | Positive |
| Autobiographical memories | Valence | 22.48 (9.77) | 53.30 (4.74) | 84.80 (9.46) | 18.48 (16.01) | 55.05 (14.02) | 86.17 (20.94) |
| | Arousal | 55.23 (14.30) | 23.98 (14.35) | 52.84 (15.80) | 62.80 (21.78) | 30.60 (15.26) | 51.45 (16.32) |
| | Vividness | 70.92 (15.13) | 62.74 (17.51) | 79.26 (12.85) | 71.38 (12.19) | 62.43 (17.07) | 76.94 (13.67) |
| | Retrieval frequency | 39.42 (17.94) | 13.28 (15.01) | 57.10 (14.45) | 46.31 (20.16) | 23.90 (21.54) | 61.31 (13.89) |
| | Sharing frequency | 32.10 (15.49) | 11.42 (12.98) | 48.12 (17.26) | 30.58 (17.13) | 18.45 (22.19) | 49.01 (16.68) |
| | Internal details | 19.44 (5.07) | 15.34 (4.96) | 23.02 (6.361) | 21.14 (4.46) | 17.20 (6.539) | 21.88 (4.98) |
| | External details | 3.40 (2.97) | 3.17 (3.09) | 2.23 (1.36) | 3.67 (2.95) | 3.70 (2.62) | 3.55 (2.20) |
| | Zygomaticus | 4.36 (17.60) | 10.16 (41.65) | 174.93 (215.4) | 3.81 (25.9) | 7.47 (61.5) | 92.21 (215.6) |
| | Corrugator | 18.33 (34.44) | 0.80 (24.22) | −11.53 (21.16) | 20.68 (45.85) | 1.06 (24.65) | −12.32 (23.45) |
| | Age in months | 11.98 (6.62) | 2.18 (4.04) | 10.12 (7.63) | 16.53 (7.26) | 2.57 (4.67) | 11.29 (8.03) |
| Movies | Valence | 16.98 (12.75) | 58.15 (12.22) | 84.44 (11.94) | 18.70 (13.49) | 56.16 (9.44) | 85.15 (11.47) |
| | Arousal | 64.20 (19.72) | 23.74 (16.21) | 46.56 (21.80) | 68.46 (16.88) | 34.26 (13.99) | 52.10 (20.05) |
| | Vividness | 70.22 (18.20) | 52.06 (21.32) | 72.31 (15.84) | 67.05 (16.36) | 42.46 (21.99) | 66.91 (15.70) |
| | Zygomaticus | −7.99 (21.71) | 24.25 (68.63) | 116.53 (175.2) | −0.36 (26.79) | 8.44 (36.47) | 76.36 (125.31) |
| | Corrugator | 85.34 (72.83) | 38.27 (52.63) | 8.95 (41.47) | 105.20 (146.3) | 47.58 (84.62) | 7.55 (37.88) |

Note. Presented are means and standard deviations in parentheses. Data was averaged within participants before calculating means and standard deviations across participants. Valence, arousal, vividness, retrieval frequency and sharing frequency were reported on visual analog scales (ranging from 0 to 100).

coded by at least two raters with a high interrater reliability (Intraclass Correlation Coefficient ICC = 0.887 and Krippendorf's alpha $\alpha_K = 0.885$). Moreover, we calculated the intraclass correlation (ICC) for episodic detail in a multilevel model in which episodic detail across all conditions is predicted by only a fixed intercept and a random intercept of participant, with ICC = 0.283, indicating that 72% of the variance of episodic detail resulted from between-participants variance and 28% resulted from within-participants variance. This underscores that our anticipated approach to investigate relationships with episodic detail both within- and across participants is valid.

We examined fEMG of the zygomaticus and corrugator to test whether the retrieval of autobiographical memories elicited psychophysiological expressions of affect in individuals with and without dysphoria. Affective expressions over time are displayed in Fig. 2. We tested whether positive and negative memories elicited psychophysiological responses in comparison to baseline measures and neutral memories, reflecting the intended valence of the memory. Bayes Factors and Posterior Probabilities for all hypotheses under consideration are listed in Table 2. Regarding zygomaticus responses, we found evidence that participants with and without dysphoria smiled more during the recall of positive memories than during the preceding baseline (non-dysphoric: $BF_{1C} = 2.286*10^{13}$, PostP$_1$ > 0.999; dysphoric: $BF_{1C} = 3796.823$, PostP$_1$ = 0.991; preregistered in Supplementary Note 13, Design Table 1: Manipulation check 1.1). Moreover, smiling responses were larger when retrieving positive compared to neutral memories (non-dysphoric: $BF_{1C} = 4.154*10^{10}$, PostP$_1$ > 0.999; dysphoric: $BF_{1C} = 9868.495$, PostP$_1$ = 0.996; Supplementary Note 13, Design Table 1: Manipulation check 1.2), which indicates that smiling responses were specific to positive memories. Regarding corrugator responses, we also found evidence that non-dysphoric as well as dysphoric participants frowned more during the recall of negative memories than during baseline (non-dysphoric: $BF_{1C} = 67.850$, PostP$_1$ = 0.796; dysphoric: $BF_{1C} = 16.654$, PostP$_1$ = 0.538; Supplementary Note 13, Design Table 1: Manipulation check 2.1). Frowning responses were larger when retrieving negative than when retrieving neutral memories (non-dysphoric: $BF_{1C} = 120774.651$, PostP$_1$ > 0.999; dysphoric: $BF_{1C} = 46760.609$, PostP$_1$ = 0.999; Supplementary Note 13, Design Table 1: Manipulation check 2.2), underscoring that these responses were specific for the retrieval of negative memories.

As can be seen in Table 1, participants' self-reports also indicated that positive memories were experienced as the most positive while negative memories were experienced as the most negative, with neutral memories scoring in between. In sum, all data quality checks were successful: the retrieval of positive and negative autobiographical memories elicited concordant affective psychophysiological expressions that could be measured to investigate potential dysphoric distortions in the affective impact of autobiographical memories.

**Confirmatory analyses**
First, we investigated dysphoric distortions in episodic detail. Specifically, we compared evidence for an overgeneral memory bias (Test 1 – H1, $BF_{1c} = 0.113$, PostP$_1$ = 0.025) and a negativity bias (Test 1 – H2, $BF_{2c} = 26.138$, PostP$_2$ = 0.902) in the number of episodic details that individuals with dysphoria retrieved when reliving autobiographical memories (Supplementary Note 13, Design Table 2: Confirmatory analyses – Test 1). In our analyses, we also considered a null hypothesis (Test 1 – H0, $BF_{0c} = 0.219$, PostP$_0$ = 0.032) and a fail-safe hypothesis (Test 1 – Hc, PostP$_c$ = 0.042). Our data provided evidence for a negativity bias: individuals with dysphoria retrieved fewer episodic detail when reliving positive memories, but more episodic detail when reliving negative memories, as illustrated in Fig. 3a. For all bar plots (Figs. 3 and 4), complementary boxplots are presented in Supplementary Note 6.

Second, we investigated expressed affect in response to autobiographical memories. We compared evidence for an overgeneral memory bias (Test 2A – H1, $BF_{1c} = 1.822$, PostP$_1$ = 0.295) and a negativity bias (Test 2A – H2, $BF_{2c} = 6.061$, PostP$_2$ = 0.700) in affective expressions of individuals with and without dysphoria when they relived autobiographical memories (Supplementary Note 13, Design Table 3: Confirmatory analyses – Test 2A). Similar to the analyses of episodic detail, we also considered a null hypothesis (Test 2A – H0, $BF_{0c} = 0.027$, PostP$_0$ = 0.006) and a fail-safe hypothesis (Test 2A – Hc, PostP$_c$ < 0.001). Given our data, a negativity bias was more likely than any other hypothesis under consideration. This suggests that individuals with dysphoria experience diminished positive affect when reliving positive memories (expressed through smiling) but enhanced negative affect when reliving negative memories (expressed through frowning). However, the posterior probability of a negativity bias did not reach the desired level of .80 at the maximum sample size.

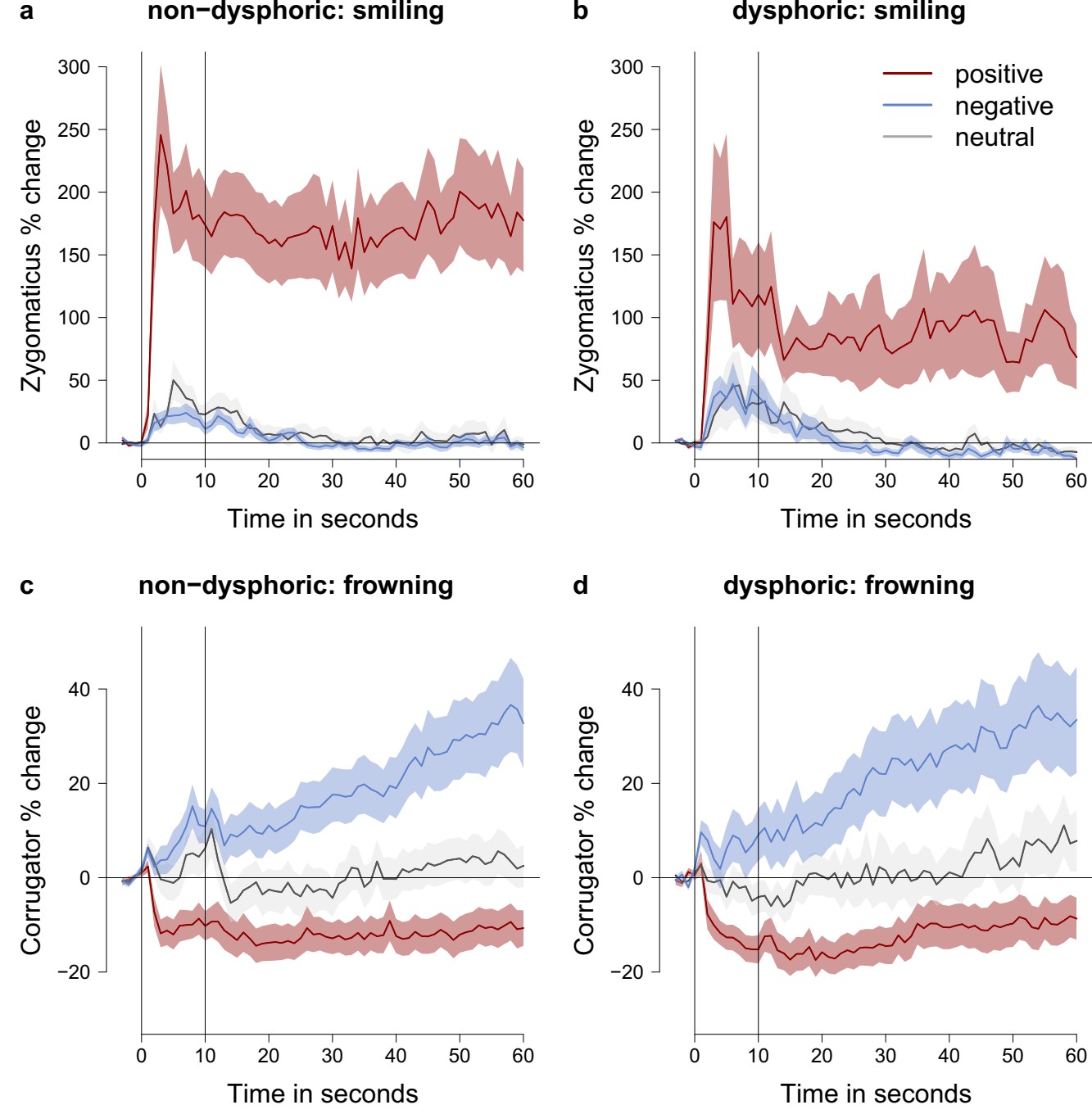

**Fig. 2 | Zygomaticus and corrugator activity over time during autobiographical memory recall. a** and **b** present zygomaticus in the non-dysphoric and dysphoric group for the positive, negative, and neutral conditions. **c** and **d** present corrugator activity. Muscle activity is presented as percentage change from the mean of the baseline (4 s). The first vertical line indicates the onset of the memory cues (0 s), the second vertical line indicates the offset of the memory cues (10 s). For the remaining 50 s participants kept silently reliving the memory. The error bands represent the standard error of the mean. The data was averaged across memories within each participant before calculating the average and standard error for each time point across participants.

Accordingly, Fig. 3b suggests that there may be an alternative explanation that we did not consider a priori: According to a positive attenuation hypothesis[64], individuals with dysphoria may experience and express diminished affect when reliving positive memories, but normal affect when reliving negative memories. This alternative hypothesis was tested in an exploratory analysis (see Exploratory Analyses).

Distorted affective responses to autobiographical memories in individuals with dysphoria may not be specific for the re-experience of memories but instead reflect general affective disturbances.

Therefore, we investigated whether individuals with dysphoria also show a negativity bias in affective responses to new events (movie clips) (Test 2B – H1, $BF_{1c} = 7.236$, $PostP_1 = 0.879$; Fig. 3c, Supplementary Fig. 9; Supplementary Note 13, Design Table 3: Confirmatory analyses – Test 2B) or whether the data suggest a different pattern of affective expressions (Test 2B – Hc, $PostP_c = 0.121$). These confirmatory analyses suggest that individuals with dysphoria show a negativity bias of affective distortions when watching movie clips that is comparable to affective distortions while remembering emotional autobiographical memories. However, positive

**Table 2 | Bayesian data quality checks for the zygomaticus (a) and corrugator (b)**

| | | Non-dysphoric | | Dysphoric | |
|---|---|---|---|---|---|
| **(a) Zygomaticus** | | $BF_{ic}$ | $PostP_i$ | $BF_{ic}$ | $PostP_i$ |
| Recall vs baseline | Recall > baseline* | >22 trillion | >0.999 | 3796.823 | 0.991 |
| | Recall < baseline | 0.000 | <0.001 | 0.000 | <0.001 |
| | Recall = baseline | 0.000 | <0.001 | 0.017 | 0.008 |
| Positive vs neutral | Positive > neutral* | >41 billion | >0.999 | 9868.495 | 0.996 |
| | Positive < neutral | 0.000 | <0.001 | 0.000 | <0.001 |
| | Positive = neutral | 0.000 | <0.001 | 0.007 | 0.002 |
| **(b) Corrugator** | | $BF_{iC}$ | $PostP_i$ | $BF_{iC}$ | $PostP_i$ |
| Recall vs baseline | Recall > baseline* | 67.850 | 0.796 | 16.654 | 0.538 |
| | Recall < baseline | 0.015 | 0.012 | 0.060 | 0.032 |
| | Recall = baseline | 0.192 | 0.192 | 1.508 | 0.430 |
| Negative vs neutral | Negative > neutral* | 120774.651 | >0.999 | 46760.609 | 0.999 |
| | Negative < neutral | 0.000 | <0.001 | 0.000 | <0.001 |
| | Negative = neutral | 0.001 | <0.001 | 0.001 | 0.001 |
| **(c) Exploratory data quality checks** | | $BF_{iC}$ | $PostP_i$ | $BF_{iC}$ | $PostP_i$ |
| Zygomaticus positive recall vs 0 | Recall > 0* | >22 trillion | >0.999 | >1 million | >0.999 |
| | Recall < 0 | 0.000 | <0.001 | 0.000 | <0.001 |
| | Recall = 0 | 0.000 | <0.001 | 0.000 | <0.001 |
| Corrugator negative recall vs 0 | Recall > 0* | >489 million | >0.999 | >2 million | >0.999 |
| | Recall < 0 | 0.000 | <0.001 | 0.000 | <0.001 |
| | Recall = 0 | 0.000 | <0.001 | 0.000 | <0.001 |

Since **a** and **b** were conducted on the fEMG data before standardization, we conducted an additional quality check in which we tested whether the standardized fEMG response during memory recall was different from 0 (**c**). Note that $BF_{iC}$ quantifies evidence for one hypothesis $i$ relative to its complement c and cannot be compared to the $BF_{ic}$ of the other hypotheses under consideration. $PostP_i$ quantifies evidence for hypothesis $i$ in comparison to all other hypotheses under consideration and, therefore, allows direct comparisons with other hypotheses under consideration. The most likely hypotheses given the data are indicated with *.

attenuation in dysphoria might offer a plausible alternative account for the pattern of affective responses to movie clips (see Exploratory Analyses). Facial EMG responses to movies over time are presented in Supplementary Note 7.

While the affective expression was the primary outcome for the reliving of affective responses during autobiographical memory recall, we conducted analyses with self-reported valence as a complementary outcome measure, given that most prior studies used self-reports to assess affective memory biases[9,24,66]. We again compared evidence for an overgeneral memory bias (Test 2A – H1, $BF_{1c} = 0.803$, $PostP_1 = 0.134$), a negativity bias (Test 2A – H2, $BF_{2c} = 0.003$, $PostP_2 = 0.001$), a null hypothesis (Test 2A – H0, $BF_{0c} = 3.907$, $PostP_0 = 0.619$), and a fail-safe hypothesis (Test 2A – Hc, $PostP_c = 0.246$; Supplementary Note 13, Design Table 3: Confirmatory analyses – Test 2A). As illustrated in Fig. 4, there was no difference in how individuals with and without dysphoria reported to feel when reliving memories, and the null hypothesis was most likely. We investigated whether the pattern of reported feelings was the same when watching movie clips or whether self-reported feelings while watching movie clips showed a different pattern (Test 2B – H0, $BF_{0c} = 57.178$, $PostP_0 = 0.983$; Supplementary Note 13, Design Table 3: Confirmatory analyses – Test 2B). The null hypothesis was extremely likely, suggesting that movies (like memories) did not result in different self-reported feelings for individuals with dysphoria compared to individuals without dysphoria.

We investigated whether memories that are retrieved with more episodic detail elicit stronger affective expressions. Specifically, we tested whether within-participant variations in episodic detail (i.e., participant-centered episodic detail) predicted affective responses to memories (Fig. 5, Table 3). Such a relationship would indicate that when a person retrieves a memory with more episodic detail, the memory also elicits stronger affective responses. For positive

memories, we found evidence against a relationship of episodic detail and zygomaticus responses in the non-dysphoric ($BF_{0c} = 8.684$, $PostP_0 = 0.813$) and in the dysphoric group ($BF_{0c} = 7.114$, $PostP_0 = 0.781$; Supplementary Note 13, Design Table 3: Confirmatory analyses – Test 3.1A). Given that there was no relationship in either group, the within-participant relationship of episodic detail and expressed positive affect did not differ between individuals with and without dysphoria (H0: $\gamma_{\text{non-dysphoric}} = \gamma_{\text{dysphoric}}$, $BF_{0c} = 8.350$, $PostP_0 = 0.807$ see Supplementary Note 8, Supplementary Table 11b for details on these analyses; Supplementary Note 13, Design Table 3: Confirmatory analyses – Test 3.1B).

We also found evidence against a within-participant relationship of the amount of episodic detail and corrugator responses to negative memories in the non-dysphoric ($BF_{0c} = 4.365$, $PostP_0 = 0.686$) and in the dysphoric group ($BF_{0c} = 6.361$, $PostP_0 = 0.761$; Supplementary Note 13, Design Table 3: Confirmatory analyses – Test 3.2A). Again, the absent within-participant relationship of episodic detail and negative affective responses did not differ between non- individuals with and without dysphoria (H0: $\gamma_{\text{non-dysphoric}} = \gamma_{\text{dysphoric}}$, $BF_{0c} = 7.700$, $PostP_0 = 0.794$, Supplementary Table 11d; Supplementary Note 13, Design Table 3: Confirmatory analyses – Test 3.2B).

In addition to investigating whether memories that are retrieved in more detail elicit stronger affective responses, we also tested whether individuals who retrieve memories in more detail tend to show stronger affective responses to memories. Specifically, we tested whether between-participant variations in episodic detail (participant-means of episodic detail) predicted affective responses to memories (Fig. 5, Table 3). We found that individuals who on average retrieved more episodic details did not express stronger positive affect when reliving positive autobiographical memories, both in the non-dysphoric ($BF_{0c} = 4.365$, $PostP_0 = 0.686$) and in the dysphoric group ($BF_{0c} = 6.361$, $PostP_0 = 0.761$; Supplementary Note 13, Design Table 3:

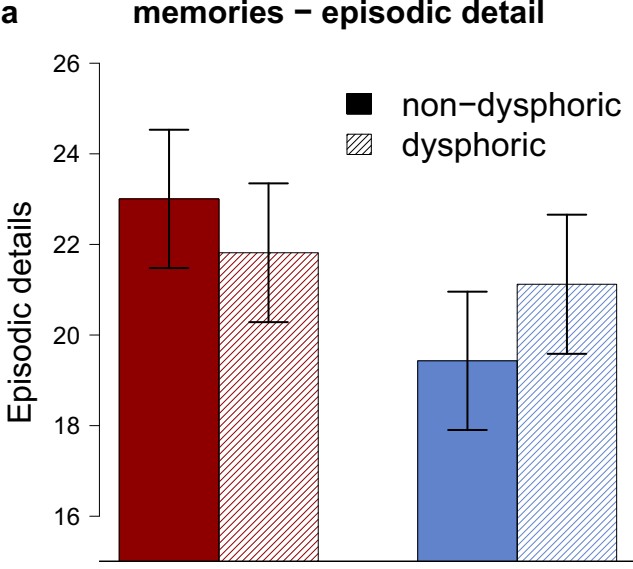

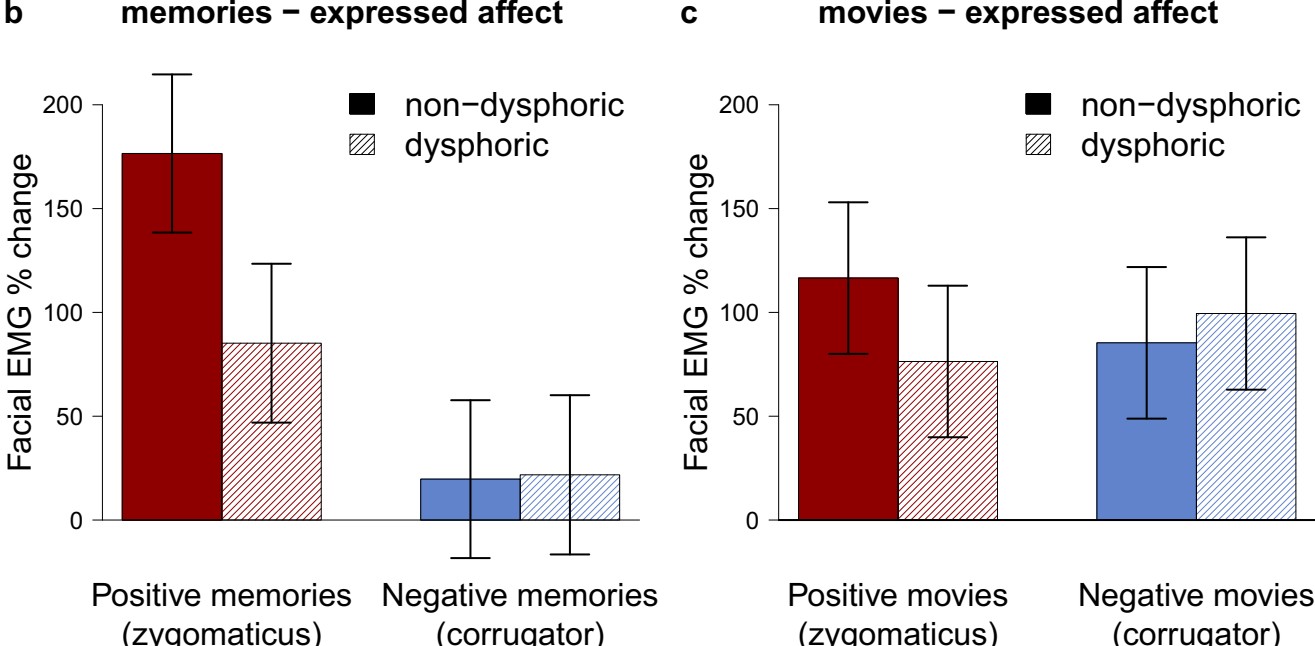

**Fig. 3 | The Re-experience Of Autobiographical Memories (ROAM) in individuals with and without dysphoria.** **a** depicts the estimated average of retrieved episodic details when reliving emotional autobiographical memories (derived from multilevel modeling which accounts for the nested data). **b** and **c** depict the estimated average of expressed affect while remembering emotional autobiographical memories and while watching emotional movie clips, respectively. The error bars represent Bayesian 0.95 central credibility intervals. The figure presents data from 779 memories (**a**), 781 memories (**b**), and 319 movie clips (**c**) of $n = 40$ participants with and $n = 40$ participants without dysphoria. Red elements refer to positive memories or movies, blue elements refer to negative memories or movies.

Confirmatory analyses – Test 3.1A). The absent between-participant relationship of episodic detail and positive affective responses was similar in the non-dysphoric and dysphoric group (H0: $\beta_{\text{non-dysphoric}} = \beta_{\text{dysphoric}}$, $BF_{0c} = 8.770$, $PostP_0 = 0.814$, Supplementary Table 11b; Supplementary Note 13, Design Table 3: Confirmatory analyses – Test 3.1B).

Individuals who on average retrieved more episodic details also did not express stronger negative affect when reliving negative autobiographical memories, both in the non-dysphoric ($BF_{0c} = 6.365$, $PostP_0 = 0.761$) and in the dysphoric group ($BF_{0c} = 5.029$, $PostP_0 = 0.715$; Supplementary Note 13, Design Table 3: Confirmatory analyses – Test 3.2A). The absent between-participant relationship of episodic detail and positive affective responses was similar in the non-dysphoric and dysphoric group (H0: $\beta_{\text{non-dysphoric}} = \beta_{\text{dysphoric}}$, $BF_{0c} = 8.237$, $PostP_0 = 0.805$, Supplementary Table 11b; Supplementary Note 13, Design Table 3: Confirmatory analyses – Test 3.2B).

In sum, we found evidence against a relationship of episodic detail and expressed affective responses both when investigating memory-by-memory variations within individuals and when investigating differences between individuals.

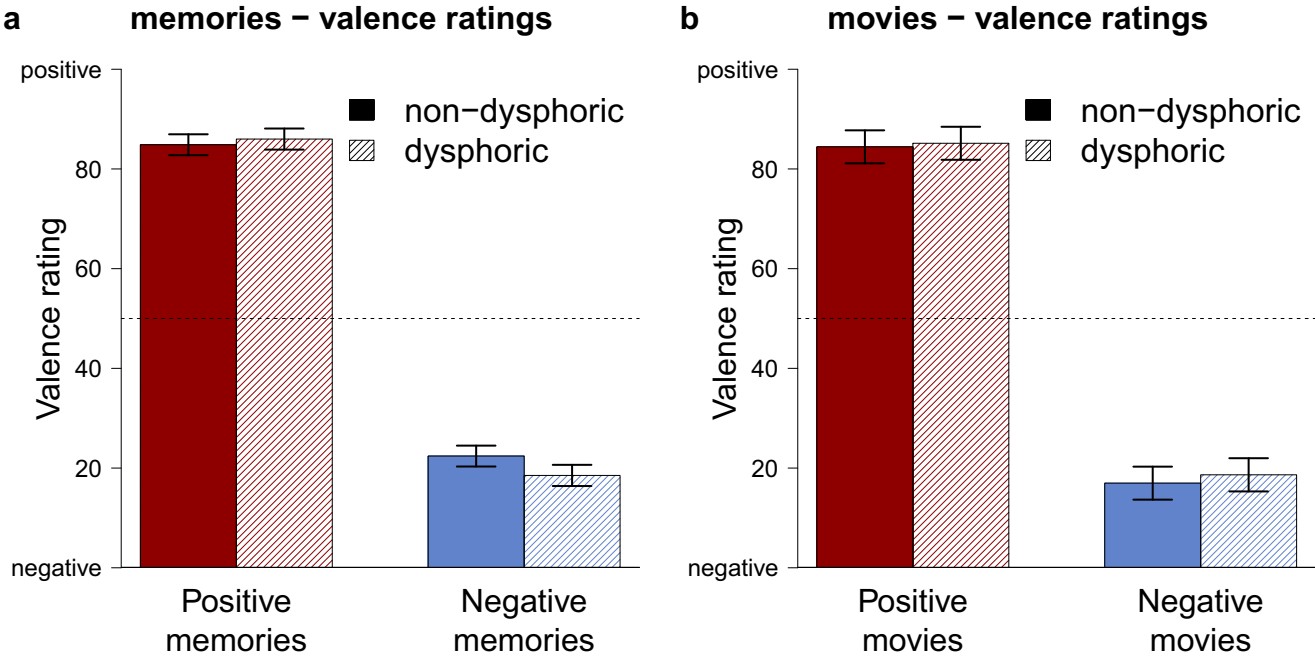

**Fig. 4 | Self-reported feelings of individuals with and without dysphoria when reliving emotional autobiographical memories or watching emotional movie clips.** Presented are the estimated average valence ratings for memories (**a**) and movie clips (**b**), derived from multilevel modeling which accounts for the nested data. The dashed line indicates neutral. The error bars represent Bayesian 0.95 central credibility intervals. The figure presents data from 781 memories (**a**), and 319 movie clips (**b**) of $n = 40$ participants with and $n = 40$ participants without dysphoria. Red elements refer to positive memories or movies, blue elements refer to negative memories or movies.

We repeated the analyses of the relationship of episodic detail and affective response with self-reported memory valence as a complementary outcome measure (instead of fEMG responses, Fig. 6, Table 4). For positive memories, we found evidence against a within-participant relationship in the non-dysphoric group ($BF_{0c} = 6.568$, $PostP_0 = 0.767$). In the dysphoric group, we found evidence that participants who retrieved more episodic details rated their memories more positively ($BF_{1c} = 2130.129$, $PostP_1 = 0.981$; Supplementary Note 13, Design Table 3: Confirmatory analyses – Test 3.1A). We found evidence that the within-participant relationship of episodic detail and positive affective responses was higher in individuals with dysphoria than in individuals without dysphoria (H0: $\gamma_{\text{non-dysphoric}} = \gamma_{\text{dysphoric}}$, $BF_{0c} = 1.670$, $PostP_0 = 0.455$; see Supplementary Note 8, Supplementary Table 12b for more details and sensitivity analyses; Supplementary Note 13, Design Table 3: Confirmatory analyses – Test 3.1B), but conclusions must be drawn with caution because the null hypothesis was almost as likely ($BF_{20} = 1.158$).

For negative memories, we found evidence against a within-participant relationship, both in the non-dysphoric ($BF_{0c} = 6.247$, $PostP_0 = 0.757$) and in the dysphoric group ($BF_{0c} = 3.157$, $PostP_0 = 0.612$; Supplementary Note 13, Design Table 3: Confirmatory analyses – Test 3.2A). We found evidence that the absent within-participant relationship of episodic detail and negative affective responses did not differ between individuals with and without dysphoria (H0: $\gamma_{\text{non-dysphoric}} = \gamma_{\text{dysphoric}}$, $BF_{0c} = 8.861$, $PostP_0 = 0.816$; Supplementary Table 12d; Supplementary Note 13, Design Table 3: Confirmatory analyses – Test 3.2B).

We also investigated whether individuals who retrieve memories with more episodic detail report stronger subjective feelings. Similar to the fEMG results (Fig. 6, Table 4), individuals who retrieved more episodic details did not report stronger positive affect when reliving positive memories (non-dysphoric: $BF_{0c} = 3.843$, $PostP_0 = 0.658$; dysphoric: $BF_{0c} = 6.074$, $PostP_0 = 0.752$; Supplementary Note 13, Design Table 3: Confirmatory analyses – Test 3.1A). There was no difference in this absent relationship of episodic detail and self-reported positive affect between individuals with and without dysphoria (H0: $\beta_{\text{non-dysphoric}} = \beta_{\text{dysphoric}}$, $BF_{0c} = 9.089$, $PostP_0 = 0.820$; Supplementary Table 12b; Supplementary Note 13, Design Table 3: Confirmatory analyses – Test 3.1B). Individuals who on average retrieved more episodic details also did not report stronger negative affect when reliving negative memories (non-dysphoric: $BF_{0c} = 9.483$, $PostP_0 = 0.826$; dysphoric: $BF_{0c} = 4.897$, $PostP_0 = 0.710$; Supplementary Note 13, Design Table 3: Confirmatory analyses – Test 3.2A). There was no difference in the absent relationship of episodic detail and self-reported negative affect between individuals with and without dysphoria (H0: $\beta_{\text{non-dysphoric}} = \beta_{\text{dysphoric}}$, $BF_{0c} = 7.448$, $PostP_0 = 0.788$; Supplementary Table 12d; Supplementary Note 13, Design Table 3: Confirmatory analyses – Test 3.2B).

To conclude, with one exception, all analyses with all outcome variables provided evidence that there was no relationship between episodic detail and affective responses to memories. Only when individuals with dysphoria retrieved positive memories in more detail, these memories were rated as more positive (but they did not elicit more smiling).

**Exploratory analyses**
The preregistered data quality check comparing fEMG activity during recall with the preceding baseline (as described above) was conducted on the fEMG data before it was standardized as percentage change from baseline. Since our confirmatory analyses concerned the standardized data, we conducted additional exploratory data checks to assess whether the percentage change from baseline was indeed different from zero (Table 2c). We found evidence that the zygomaticus response in the positive condition was larger than 0 in both groups (non-dysphoric: $BF_{1C} = 2.291 \times 10^{13}$, $PostP_1 > 0.999$; dysphoric: $BF_{1C} = 1.039 \times 10^6$, $PostP_1 > 0.999$). The corrugator response in the negative condition was also larger than 0 in both groups (non-dysphoric: $BF_{1C} = 4.895 \times 10^8$, $PostP_1 > 0.999$; dysphoric: $BF_{1C} = 2.180 \times 10^6$,

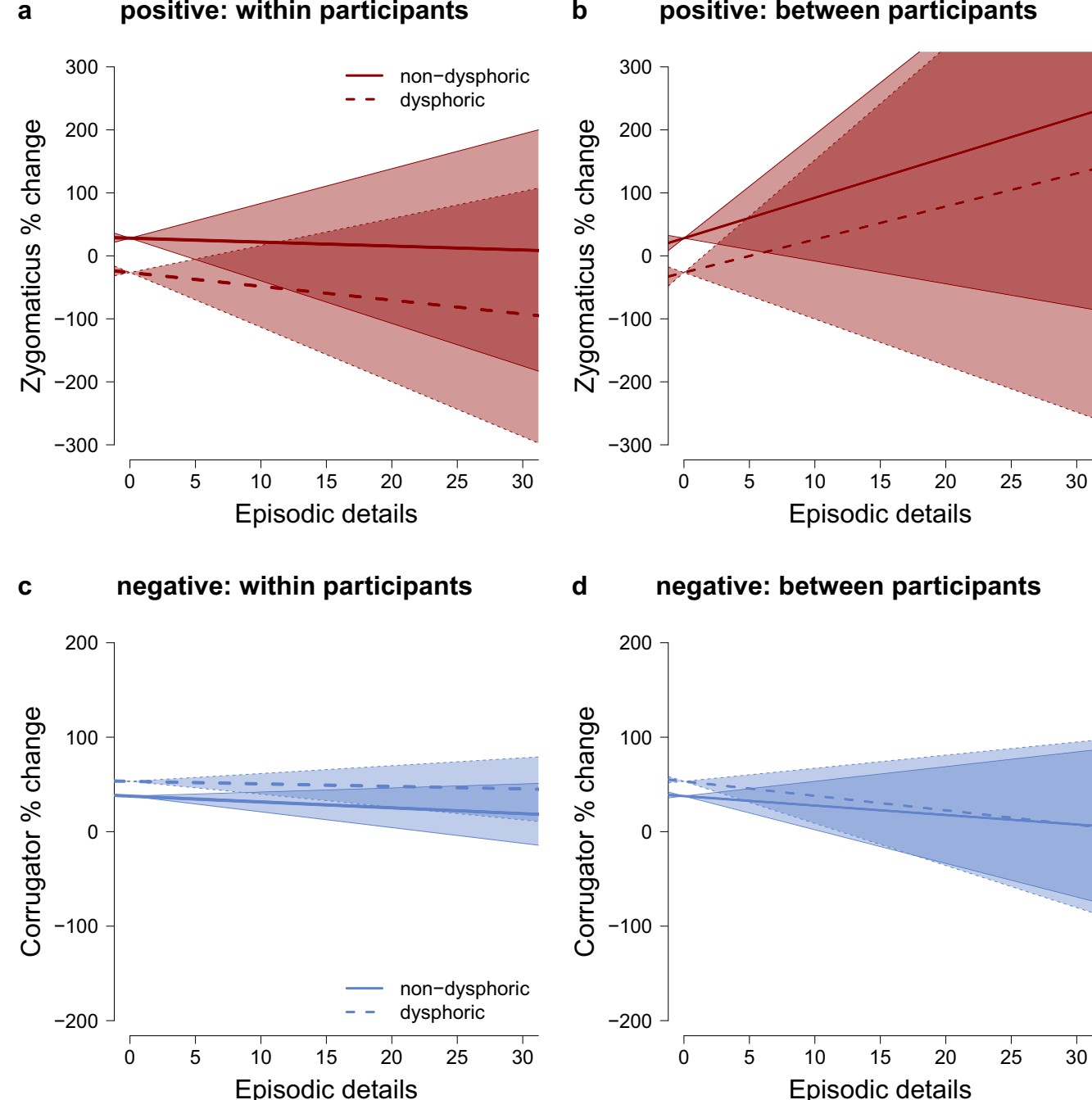

**Fig. 5 | Relationship between episodic detail and affective expression during autobiographical memory recall.** The upper panel (**a** and **b**) shows the relationship of episodic detail and positive affect measured with the zygomaticus. The lower panel (**c** and **b**) shows the relationship of episodic detail and negative affect measured with the corrugator. The left panel (**a** and **c**) depicts within-participant relationships (participant mean centered episodic detail). The right panel (**b** and **d**) depicts between-participant relationships (participant mean of episodic detail). The error bands indicate the lower and upper bound of the 0.95 credibility interval around the estimate of the slope. The figures represent data from 391 positive (**a** and **b**) and 388 negative memories (**c** and **d**) of $n = 40$ participants with and $n = 40$ participants without dysphoria.

PostP$_1 > 0.999$). These exploratory data checks further showed that emotional memories elicited facial expressions of positive and negative affect.

The data of expressed affect during the reliving of memories (analyzed in Test 2A) indicated that individuals with dysphoria might show positive attenuation[64], an hypothesis that we did not consider a priori. That is individuals with dysphoria express diminished affect when reliving positive memories, but normal affect when reliving negative memories. Therefore, we conducted an exploratory analysis

to compare evidence for a negativity bias (Exploration 1 – H1, BF$_{1c} = 5.895$, PostP$_1 = 0.162$) with evidence for positive attenuation (Exploration 1 – H2, BF$_{2c} = 17.050$, PostP$_2 = 0.815$). Given our data, positive attenuation was five times more likely than a negativity bias (BF$_{12} = 5.16$).

We also repeated the analysis of affective responses to movie clips including a positive attenuation hypothesis as a plausible alternative explanation that was not considered a priori. Specifically, we tested whether individuals with dysphoria showed a negativity bias (H1,

**Table 3 | Bayesian analyses of evidence for and against relationships of retrieved episodic detail and expressed affect when reliving autobiographical memories**

| | | Non-dysphoric | | Dysphoric | |
|---|---|---|---|---|---|
| **(a) Positive memories - zygomaticus** | | **BF$_{iC}$** | **PostP$_i$** | **BF$_{iC}$** | **PostP$_i$** |
| Within-participant relationship of affect & detail | $\gamma > 0$ | 0.727 | 0.079 | 0.340 | 0.056 |
| | $\gamma < 0$ | 1.376 | 0.108 | 2.939 | 0.164 |
| | $\gamma = 0$* | 8.684 | 0.813 | 7.114 | 0.781 |
| Between-participant relationship of affect & detail | $\beta > 0$ | 8.511 | 0.296 | 3.811 | 0.189 |
| | $\beta < 0$ | 0.117 | 0.035 | 0.262 | 0.050 |
| | $\beta = 0$* | 4.042 | 0.669 | 6.361 | 0.761 |
| **(b) Negative memories - corrugator** | | **BF$_{iC}$** | **PostP$_i$** | **BF$_{iC}$** | **PostP$_i$** |
| Within-participant relationship of affect & detail | $\gamma > 0$ | 0.141 | 0.039 | 0.463 | 0.066 |
| | $\gamma < 0$ | 7.109 | 0.275 | 2.160 | 0.142 |
| | $\gamma = 0$* | 4.365 | 0.686 | 7.620 | 0.792 |
| Between-participant relationship of affect & detail | $\beta > 0$ | 0.285 | 0.053 | 0.179 | 0.043 |
| | $\beta < 0$ | 3.512 | 0.186 | 5.591 | 0.241 |
| | $\beta = 0$* | 6.365 | 0.761 | 5.029 | 0.715 |

The most likely hypothesis given our data is indicated with *.

**Table 4 | Bayesian analyses of evidence for and against relationships of retrieved episodic detail and self-reported affect (valence) when reliving autobiographical memories**

| | | non-dysphoric | | dysphoric | |
|---|---|---|---|---|---|
| **(a) Positive memories** | | **BF$_{iC}$** | **PostP$_i$** | **BF$_{iC}$** | **PostP$_i$** |
| Within-participant relationship of affect & detail | $\gamma > 0$ | 3.810 | 0.185 | 2130.129 | *0.981 |
| | $\gamma < 0$ | 0.262 | 0.049 | 0.000 | <0.001 |
| | $\gamma = 0$ | 6.568 | *0.767 | 0.038 | 0.019 |
| Between-participant relationship of affect & detail | $\beta > 0$ | 9.644 | 0.310 | 4.471 | 0.202 |
| | $\beta < 0$ | 0.104 | 0.032 | 0.224 | 0.045 |
| | $\beta = 0$* | 3.843 | 0.658 | 6.074 | 0.752 |
| **(b) Negative memories** | | **BF$_{iC}$** | **PostP$_i$** | **BF$_{iC}$** | **PostP$_i$** |
| Within-participant relationship of affect & detail | $\gamma > 0$ | 0.209 | 0.042 | 0.071 | 0.026 |
| | $\gamma < 0$ | 4.778 | 0.201 | 13.992 | 0.362 |
| | $\gamma = 0$* | 6.247 | 0.757 | 3.157 | 0.612 |
| Between-participant relationship of affect & detail | $\beta > 0$ | 0.692 | 0.071 | 0.137 | 0.035 |
| | $\beta < 0$ | 1.445 | 0.103 | 7.297 | 0.255 |
| | $\beta = 0$* | 9.483 | 0.826 | 4.897 | 0.710 |

The most likely hypothesis given our data is indicated with *.

BF$_{1c}$ = 7.201, PostP$_1$ = 0.201) or positive attenuation when viewing movie clips (H2, BF$_{2c}$ = 12.148, PostP$_2$ = 0.771). We also included the complement hypothesis that neither a negativity bias nor positive attenuation account well for the data (Hc: PostP$_C$ = 0.028). These exploratory results suggest that positive attenuation in dysphoria might account best for affective expressions in response to movie clips, with positive attenuation being almost four times more likely than a negativity bias (BF$_{21}$ = 3.83).

We explored whether positive and negative affective expressions habituated in participants with and without dysphoria. First, we tested whether zygomaticus responses to positive memories declined over time in the non-dysphoric group (H1: trial 1 > trial 2 > trial 3 > trial 4 > trial 5, BF$_{1c}$ = 2.477, PostP$_1$ = 0.061) or whether zygomaticus responses were similar across trials (H0: trial 1 = trial 2 = trial 3 = trial 4 = trial 5, BF$_{0c}$ = 36.987, PostP$_0$ = 0.915). We also evaluated a fail-safe hypothesis (PostP$_C$ = 0.024). Second, we tested whether zygomaticus responses declined over time in the dysphoric group (H1: BF$_{1c}$ = 0.203, PostP$_1$ = 0.031) or whether zygomaticus responses were similar across trials (H0: BF$_{0c}$ = 5.355, PostP$_0$ = 0.816; fail-safe hypothesis: PostP$_C$ = 0.153). The results provided evidence that zygomaticus responses did not vary over time in the non-dysphoric nor in the dysphoric group.

We also tested whether corrugator responses to negative memories declined over time in the non-dysphoric group (H1: BF$_{1c}$ = 0.476, PostP$_1$ = 0.011) or whether corrugator responses were similar across trials (H0: BF$_{0c}$ = 41.204, PostP$_0$ = 0.965; fail-safe hypothesis: PostP$_C$ = 0.024). Finally, we tested whether corrugator responses declined over time in the dysphoric group (H1: BF$_{1c}$ = 0.042, PostP$_1$ = 0.019) or whether corrugator responses were similar across trials (H0: BF$_{0c}$ = 1.169, PostP$_0$ = 0.527; fail-safe hypothesis: PostP$_C$ = 0.454). These results provide evidence that corrugator responses did not habituate over time in participants with and without dysphoria. In the non-dysphoric group, corrugator responses were similar across trials. In the dysphoric group, responses varied somewhat across trials, but they did not decline over time (e.g., the largest average corrugator response in the dysphoric group was to the 4th of 5 negative memories).

Given that earlier work has found that a negativity bias is associated with symptom severity in depression[67], we explored whether more severe depressive symptoms were associated with more severe episodic and affective memory distortions in the dysphoric group.

We found evidence against a relationship of depressive symptoms and episodic detail for positive memories (H1: $\beta > 0$, BF$_{1c}$ = 1.536, PostP$_1$ = 0.153; H2: $\beta < 0$, BF$_{2c}$ = 0.651, PostP$_2$ = 0.099; H0: $\beta = 0$, BF$_{0c}$ = 5.928, PostP$_0$ = 0.708) and for negative memories (H1: $\beta > 0$, BF$_{1c}$ = 5.050, PostP$_1$ = 0.259; H2: $\beta < 0$, BF$_{2c}$ = 0.198, PostP$_2$ = 0.051; H0: $\beta = 0$, BF$_{0c}$ = 4.435, PostP$_0$ = 0.689). Additionally, we found evidence against a relationship of depressive symptoms and zygomaticus responses to positive memories (H1: $\beta > 0$, BF$_{1c}$ = 1.082, PostP$_1$ = 0.132; H2: $\beta < 0$, BF$_{2c}$ = 0.924, PostP$_2$ = 0.122; H0: $\beta = 0$, BF$_{0c}$ = 5.866, PostP$_0$ = 0.746) or corrugator responses to negative memories (H1: $\beta > 0$, BF$_{1c}$ = 0.218, PostP$_1$ = 0.061; H2: $\beta < 0$, BF$_{2c}$ = 4.595, PostP$_2$ = 0.280; H0: $\beta = 0$, BF$_{0c}$ = 3.867, PostP$_0$ = 0.659). These exploratory results need to be interpreted with caution because the study design only included participants with dysphoria based on a BDI score of 16 or higher without a diagnosis of depression. Consequently, the dysphoric group may have been too homogenous in symptom severity to reveal a relationship between depressive symptoms and other variables within the dysphoric group.

We explored positive and negative affect in both groups before and after they completed the ROAM, measured with the PANAS. One participant without dysphoria did not complete the PANAS and was excluded from these analyses. Regarding positive affect, participants without dysphoria scored an average of M = 27.46 (SD = 7.06) before and M = 25.87 (SD = 7.78) after completing the ROAM. Participants with dysphoria scored an average of M = 25.60 (SD = 6.83) before and M = 25.92 (SD = 7.62) after the ROAM. Overall, participants with dysphoria did not differ in terms of reported positive affect from participants without dysphoria (H1: $\beta_{dysphoric} < \beta_{non\text{-}dysphoric}$, BF$_{1c}$ = 1.508, PostP$_1$ = 0.231; H0: $\beta_{dysphoric} = \beta_{non\text{-}dysphoric}$, BF$_{0c}$ = 3.369, PostP$_0$ = 0.616; fail-safe hypothesis: PostP$_c$ = 0.153). Comparing possible changes in affect between the groups (Δ: positive affect after minus positive affect before the task), positive affect may have decreased more in the non-dysphoric group than in the dysphoric group (H1: $\beta_{\Delta dysphoric} > \beta_{\Delta non\text{-}dysphoric}$, BF$_{1c}$ = 0.049, PostP$_1$ = 0.026; H2: $\beta_{\Delta dysphoric} < \beta_{\Delta non\text{-}dysphoric}$, BF$_{2c}$ = 20.593, PostP$_2$ = 0.540; H0: $\beta_{\Delta dysphoric} = \beta_{\Delta non\text{-}dysphoric}$, BF$_{0c}$ = 1.532, PostP$_0$ = 0.434), but the null hypothesis was almost equally likely (BF$_{20}$ = 1.245). In sum, positive affect measured with the PANAS did not differ between participants with and without dysphoria, but participants without dysphoria may have felt somewhat less positive after completing the memory task.

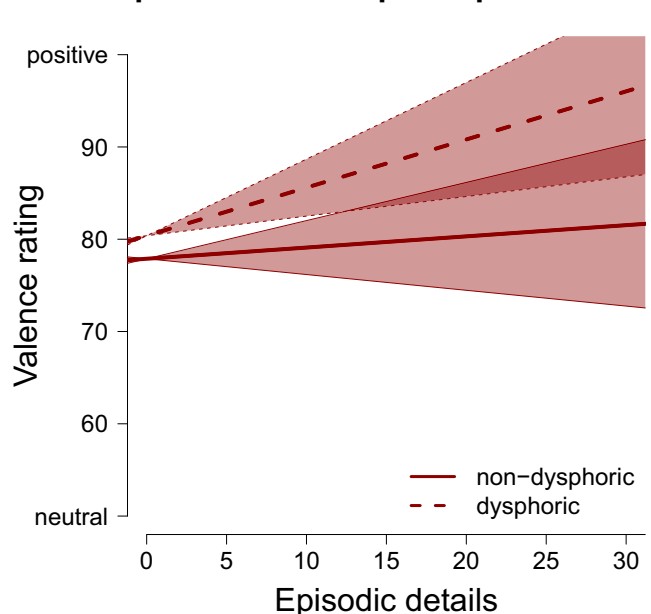

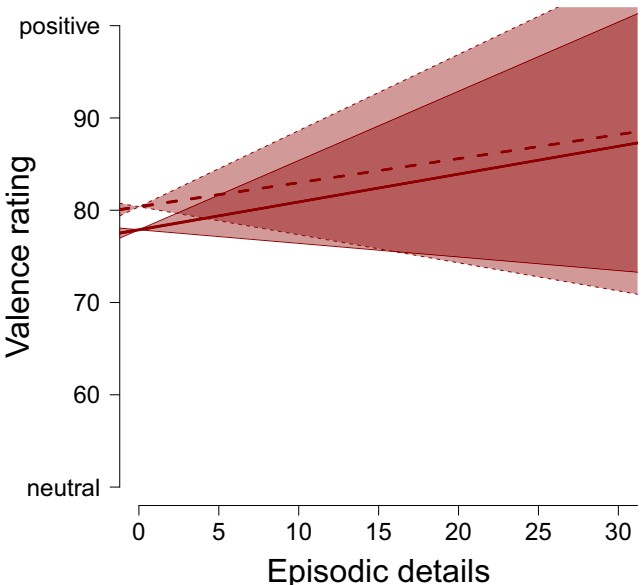

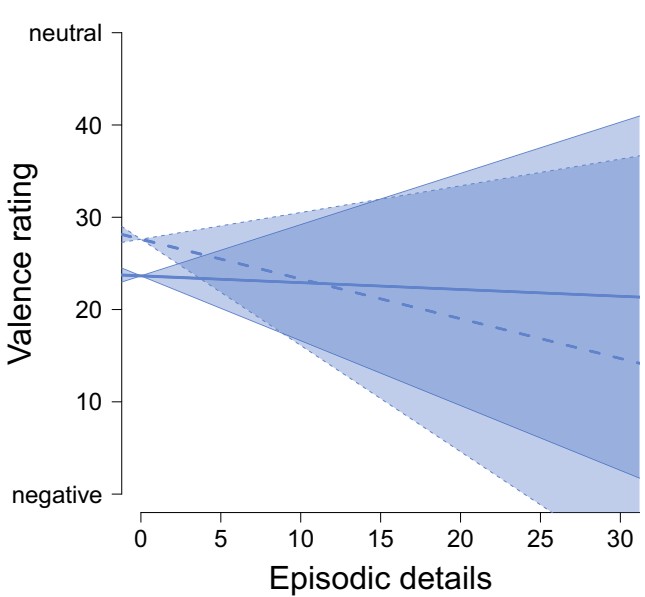

Fig. 6 | **Relationship between episodic detail and self-reported affect (valence) during autobiographical memory recall.** The upper panel (**a** and **b**) shows the relationship of episodic detail and valence when remembering positive events. The lower panel (**c** and **b**) shows the relationship of episodic detail and valence when remembering negative events. The left panel (**a** and **c**) depicts within-participant relationships (participant mean centered episodic detail). The right panel (**b** and **d**) depicts between-participant relationships (participant mean of episodic detail). The error bands indicate the lower and upper bound of the 0.95 credibility interval around the estimate of the slope. The figures represent data from 391 positive (**a** and **b**) and 388 negative memories (**c** and **d**) of $n = 40$ participants with and $n = 40$ participants without dysphoria.

Regarding negative affect, participants without dysphoria scored an average of $M = 10.85$ ($SD = 1.20$) before and $M = 12.33$ ($SD = 2.72$) after completing the ROAM. Participants with dysphoria scored an average of $M = 17.57$ ($SD = 5.63$) before and $M = 18.59$ ($SD = 6.23$) after the ROAM. Overall, participants with dysphoria reported stronger negative affect than participants without dysphoria (H1: $\beta_{dysphoric} > \beta_{non\text{-}dysphoric}$, $BF_{1c} = 8.774 \times 10^{8}$, $PostP_1 > 0.999$; H0: $\beta_{dysphoric} = \beta_{non\text{-}dysphoric}$, $BF_{0c} = 0.000$, $PostP_0 < 0.001$; $PostP_c < 0.001$). Possible changes in negative affect after retrieval did not differ between the two groups (H1: $\beta_{\Delta dysphoric} > \beta_{\Delta non\text{-}dysphoric}$, $BF_{1c} = 0.441$, $PostP_1 = 0.081$; H2: $\beta_{\Delta dysphoric} < \beta_{\Delta non\text{-}dysphoric}$, $BF_{2c} = 2.267$, $PostP_2 = 0.184$; H0: $\beta_{\Delta dysphoric} = \beta_{\Delta non\text{-}dysphoric}$, $BF_{0c} = 5.549$, $PostP_0 = 0.735$).

## Discussion

Theories of autobiographical memory distortions in dysphoria – most notably overgeneral and negativity bias theories – disagree on how positive and negative memories are altered in terms of episodic detail

and affective responses. We therefore quantified evidence for diverging theoretical predictions, using a Bayesian approach. We found that participants with dysphoria remembered fewer episodic details when reliving happy autobiographical memories but more episodic details when reliving sad memories, compared to participants without dysphoria. These results are in line with a negativity bias but not with an overgeneral memory bias, which would entail reduced episodic detail for both positive and negative memories. Similarly, participants with dysphoria expressed reduced positive affect when reliving happy memories, but normal or slightly enhanced negative affect when reliving sad memories, which is again in line with a negativity bias. However, the difference in negative affect between the groups was very small and evidence for a negativity bias in terms of expressed affect was therefore not decisive (i.e., the posterior probability bias was slightly lower than the stopping criterion $PostP_i = 0.80$). Exploratory analyses indicated that positive attenuation[64] – blunted positive affect during happy memory but normal negative affect during sad memory retrieval – may better describe affective memory distortions. These affective distortions were not unique to autobiographical memories as facing novel emotional events (i.e., movie clips) resulted in a comparable pattern of affective distortions. Overall, our findings shed nuanced insights into memory distortions in dysphoria and challenge several important assumptions of memory bias theories and emerging memory therapeutics.

Across all analyses, we found converging evidence that a negativity bias is a more plausible explanation for memory distortions in dysphoria than an overgeneral memory bias. This finding ostensibly conflicts with previous research showing that participants with dysphoria retrieve fewer specific memories than participants without dysphoria, regardless of memory valence[28,42]. However, such reduced memory specificity was traditionally assessed with the autobiographical memory test (AMT), which probes the ease of memory access during memory search. That is, memory specificity in the AMT assesses how likely a person is to retrieve specific or categorical (overgeneral) memories. In the ROAM employed in this study, episodic detail represents the recollection of specific memories after successful completion of the search process, because participants provided personal memory cues for specific events prior to reliving each memory. Therefore, our study complements findings from the AMT regarding memory access with insights on subsequent elaboration and reliving, which may represent distinct processes that rely on different cognitive and biological systems[68–71]. This approach is in line with a recent call to not only investigate memory distortions in terms of specificity, but also other related yet dissociable memory features such as memory detail and emotionality[69]. Notably, different cognitive and affective biases may interact in the etiology and maintenance of depression[72,73]. Regarding memory distortions in dysphoria, research with the AMT shows that people with dysphoria do not readily retrieve specific memories, be it positive or negative. However, our data underscore that people with dysphoria can retrieve specific memories, and when they do, their negative memories are enhanced in detail and possibly affect, while their positive memories are reduced in detail and affect, which is in line with a negativity bias.

Our observations also have consequences for other levels of analysis that build on the notions of a negativity or an overgeneral bias. For example, it has been suggested that diminished (i.e., overgeneral) positive memories in dysphoria may result from reduced neurogenesis in the hippocampus and ensuing problems with pattern separation[46], while excessively negative memories may result from amygdala sensitization[46]. The question remains, however, why problems in pattern separation would not equally cause overgeneral negative memories and why amygdala sensitization would not affect positive memories. Previous studies investigating overgeneral memory often did not distinguish between positive and negative memories[33] or only investigated memories of a particular valence (e.g., positive memories

and their capacity to alleviate a negative mood)[9,24,66,74]. Our data show that causal explanations for memory biases need to accommodate different patterns of episodic and affective distortions for positive and negative memories. In general, a more fine-grained behavioral characterization of memory distortions is necessary before any causal biological or cognitive explanations may be inferred.

While we found evidence for affective distortions when participants with dysphoria relived autobiographical memories, these were not specific to memories but already present during the experience of new events (i.e., watching movie clips). This suggests that in addition to, or even instead of, explaining affective distortions with aberrant mnemonic processes, it is important to consider general cognitive and affective disturbances that could explain memory distortions without requiring aberrant memory processes per se. Specifically, we found reduced positive affect in combination with normal or slightly enhanced negative affect for both memories and movies. Therefore, affective memory distortions could result from blunted positive affect and reward processing that perturb reactions to any emotional stimulus, regardless of whether the stimulus involves memory systems or not. This interpretation aligns with studies that showed diminished reward processing in individuals with dysphoria, as demonstrated with blunted facial expressions in response to monetary rewards[57–59] and positive social cues[75] as well as with studies on neural aberrations of the mesocorticolimbic system in depression[76,77]. The pronounced distortions of positive affect in combination with little or no distortions in negative affect also suggest that dysphoria may be driven more by alterations in the positive valence and reward system rather than the negative valence and defence system[78].

It might be possible that the magnitude of the attenuation of positive affect in dysphoria may be stronger for memories than for new experiences (a notion that seems to be consistent with a visual inspection of Fig. 3b, c). Initially small affective distortions during encoding could also exacerbate over time, resulting in even stronger affective distortions during memory retrieval[79,80]. However, the memories and movies used in this study might differ in self-relevance and intrinsic emotionality, because they do not refer to the same encoding experience. Therefore, our data do not allow for conclusions regarding differences in the magnitude of affective distortions during encoding compared to remembering episodes (as opposed to different patterns of distortions during encoding and remembering against which we found evidence). Future research could measure affective responses during encoding and subsequent retrieval of the same episodes to investigate how affective biases evolve over time, for example using movie clips and memories thereof in the lab[81].

Our study revealed affective distortions when analyzing facial expressions but no distortions in subjective self-reports of emotionality. The absence of dysphoric distortions in self-reported affect was unexpected because previous studies on affective memory distortions mainly relied on self-reports[9,24,35,36]. However, only a relatively small number of experiments investigated feelings during autobiographical memory recall[9,24,35,36]. These studies usually investigated the capability of positive memories to alleviate a previously induced negative mood[9,24,66], but not pure affective responses to a memories per se, or they did not compare affective responses between people with and without dysphoria[35,36]. The most compelling evidence for distorted subjective feelings during memory recall comes from studies on the fading affect bias (i.e., among healthy individuals, affect associated with positive memories fades slower than affect associated with negative memories), revealing that the affective load of positive memories fades quicker in individuals with dysphoria compared to individuals without dysphoria[79,80]. In these studies, participants retrospectively rated how events made them feel in the past before indicating how the memories made them feel in the present. This not only requires higher-order cognitive processes to label feelings, but may also encourage deliberate comparisons between feelings in the past

and the present. Thus, subjective distortions in affect may only become apparent when participants with dysphoria contrast their current feelings with the past.

The divergence of psychophysiological and subjective indices also corroborates that different measures of affect tap into different components or levels of affective processing[39]. Specifically, self-reported feelings require higher-order cognitive processes for the categorization and labeling of an emotional experience[39], whereas facial expressions measured with fEMG are relatively automatic affective responses that are tightly linked to motivational processes[82–84]. Our results therefore suggest that the affective processes that are compromised in dysphoria are relatively automatic and potentially related to blunted reward processing.

Expressed affect measured with fEMG was not only independent from subjective feelings, but also from remembered episodic details. The pattern of affective memory distortions differed somewhat from episodic memory distortions, and we found consistent evidence against a relationship between episodic detail and expressed affect (whether assessed in dysphoria or healthy controls). The divergent observations on the three major dependent variables in our study (episodic detail, expressed affect, subjective feelings) conflict with the assumption of memory theories that the affective impact of memories is closely related to the retrieval of declarative memory components[14,30,36]. It also underscores the necessity to carefully interpret findings according to the exact response system under investigation, since these can diverge considerably and cannot be interpreted interchangeably.

Next to theoretical insights into autobiographical memory distortions, our study bears tentative clinical implications. First, affective distortions in dysphoria were particularly pronounced for positive but not negative experiences, which is in line with recent appeals that interventions for depressive symptoms need to acknowledge the importance of anhedonia[85]. Interestingly, depressed patients who seek professional help consider increasing positive affect a more important treatment goal than decreasing negative affect[86]. That said, there is plenty of evidence that negative affect does play an important role in depression[40,67]. It is possible that over the course of one or more depressive episodes a lack of positivity may be extended with increases in negative affect that are characteristic for more severely depressed samples (our sample consisted of people with dysphoria who were at risk or at the onset of a clinical depression). Second, the finding that affective responses to memories are independent of episodic detail and that affective distortions in dysphoria are not specific to memories challenge the promise of emerging memory therapeutics at least as a stand-alone treatment[31,34]. If episodic and affective memory components are not tightly intertwined, it is unlikely that boosting declarative memory components will result in an enhanced capacity of memories to increase positive affect or to promote more adaptive behavior (e.g., reward seeking). Interventions may be more effective when they target affective processes in general, for example by increasing the motivation and ability to seek positive experiences (e.g., behavioral activation therapy)[87] or by increasing reward sensitivity[76]. For example, antidepressant drugs (e.g., reboxetine) seem to alleviate depressive symptoms by improving the otherwise impoverished processing of positive information[88,89]. It may be possible to combine pharmacological with therapeutic interventions such as behavioral activation or imagery training to enhance the experience of positive events and ultimately promote more positive beliefs about oneself and the world.

While our study has several major strengths, it is also subject to limitations. First, our sample included almost exclusively young female participants. However, as mood disorders are particularly prevalent in women and often arise during adolescence or young adulthood[90], our sample represents a population for which it is particularly important to understand affective and cognitive

distortions that contribute to depressive symptoms. Second, we had to exclude a large number of participants after they had participated, mostly because depressive symptoms scores changed between the online screening and participation. Based on informal conversations during the debriefing, at least some changes in depressive symptoms represented real changes in depressive symptoms rather than minor temporal fluctuations, for example because of Covid-19 infections, break-ups, or job rejections. Consequently, it was important to assess depressive symptoms in the lab, even after rigorous online screening. Finally, it is noteworthy that the failsafe hypothesis received very little evidence in our analysis of affective memory distortions, even though exploratory analyses suggested that positive attention may be more likely than a negativity bias. A potential explanation for this discrepancy is that the negativity bias included specific predictions (high complexity) and is favored over the less specific fail-safe hypothesis unless the data is clearly not in line with a negativity bias. The clear difference of positive affect between individuals with and without dysphoria may have further reduced evidence for the fail-safe hypothesis that also included the possibility that there is similar or even more positive affect in the dysphoric group than in the non-dysphoric group. These considerations highlight the power of BAIN to find evidence for specific sets of predictions, but also suggest that possible hypotheses should not be ruled out a priori if there is no strong prior evidence against them. A potential alternative for unlikely but not impossible hypotheses would be to consider them with smaller prior probabilities.

To conclude, a major promise of emerging memory therapeutics is to provide novel evidence-based tools to alleviate the burden of memory distortions and depressive symptoms, but our results challenge several common theoretical assumptions on autobiographical memory distortions. We demonstrate the need for a critical behavioral characterization of distortions in memory and general affect before any underlying biological or cognitive process can be inferred and before empirical insights on emotional autobiographical memories can be translated into effective evidence-based interventions.

## Methods
### Ethics information
The study procedures were approved by the ethics committee of the University of Amsterdam (2019-CP-10552). The procedure, hypotheses, and analysis plan were preregistered on the Open Science Framework (https://doi.org/10.17605/OSF.IO/3D9AN) and figshare (https://doi.org/10.6084/m9.figshare.14605374.v1) and in principle accepted as a stage 1 registered report on the 18rd of March 2021[91]. Minor deviations from the preregistration as described in the stage 1 protocol are summarized in Supplementary Note 9. Participants could choose to be compensated with course credits or 10.00 EUR per hour. Participants were recruited via an online portal for volunteers of the University of Amsterdam, via advertisements on social media, as well as through posters, and flyers. All participants provided written informed consent prior to participation. Since we recruited individuals with dysphoria, there was a chance that some participants would experience severe depressive symptoms and need help from a clinician. Therefore, all participants were debriefed at the end of the second session, regardless of whether they were in the dysphoric or non-dysphoric group. During the debriefing, the experimenter provided information on depressive symptoms and gave every participant an information letter with contacts of clinicians in the area. The experimenter also asked the participant whether they reported suicidal intentions (Beck Depression Inventory modified (BDI-II)[92]; response 2 or 3 to item 9: "I would like to kill myself" or "I would kill myself if I had the chance"). In that case, the experimenter addressed this with the participant and suggested to immediately contact the Dutch suicide prevention hotline or their general practitioner.

## Piloting the ROAM

We conducted three pilot studies with $N = 8$, $N = 5$, and $N = 8$ participants, respectively, to validate and refine the ROAM in its capability to assess episodic memory detail and to track the psychophysiological expression of emotion during autobiographical memory retrieval (see Supplementary Note 1 for a summary). Detailed analyses of the final pilot are presented in Supplementary Note 2 (Supplementary Note 2 was included in the Methods section of the stage 1 registered report).

## Design

**Procedure.** The general procedure is illustrated in Fig. 1b. Interested participants completed an online screening questionnaire via the survey tool *Qualtrics* (Qualtrics, Provo, UT). The screening included the BDI-II to allocate participants to the dysphoric or non-dysphoric group as well as questions regarding other exclusion criteria such as excessive drug use (see Sampling Plan). Participants who reported a BDI-II below 4 (non-dysphoric group) or above 15 (dysphoric group) were invited to participate in the lab as soon as possible. Participants who met an exclusion criterion were not invited to the lab. This online screening was conducted by a person who was otherwise not involved in running the experiment. The person responsible for the online screening also transcribed and coded several memories but only after all identifying information was removed, so that she was blind regarding the participants' group allocation (dysphoric or non-dysphoric). Upon selection through the online screening procedure, participants completed two experimental sessions that were separated by 48 h (±4 h). All instructions, questions, and stimuli were presented in Dutch if not explicitly stated otherwise. In this report, we present English translations, but we provide the original materials for which no legal or ethical restrictions apply on OSF. Computer tasks were implemented with Neurobs Presentation software (https://www.neurobs.com, version 21).

During session 1, participants provided written informed consent and completed a screening questionnaire that again assessed all exclusion criteria. If a participant met an exclusion criterion, they were not tested further but compensated for the first session. Following, participants completed the Positive and Negative Affect Scale (PANAS)[93]. Then, participant completed the first part of the ROAM. They selected five positive, five negative, and three neutral autobiographical memories and noted three cue words per memory on a pen and paper questionnaire. Following memory cue selection, the experimenter attached the psychophysiological sensors (fEMG, skin conductance, electrocardiography) for psychophysiological data collection while participants watched movie clips. However, before watching the movie clips, participants completed several pen and paper questionnaires: PANAS[93], SUIS[94]; PSI-Q[95]; STAI-T[96], SHAPS[97] (see also Questionnaires). The participants put the questionnaires into an envelope that was not opened until the end of session 2. Finally, participants watched two negative, two positive, and two neutral movie clips. After movie watching, the electrodes were removed, and the participant was reminded of the follow-up appointment.

During session 2, the experimenter attached the psychophysiological sensors before the participants completed the BDI-II and the PANAS. The participant again put the questionnaires into an envelope that was not opened until the end of session 2. Following, participants completed the computerized memory re-experiencing component of the ROAM in which they retrieved autobiographical memories in response to the cues that they provided in session 1. After the computer task, participants completed the PANAS. Participants further reported on a written informed consent whether we are allowed to share their personal memories 1) without any restrictions on an online platform such as OSF and during presentations, 2) with restrictions on such a platform, 3) only with other researchers upon request, or 4) whether we are not allowed to share their memories. Finally, every participant was debriefed (see Ethics Information). During the debriefing, a participant could choose to be informed about their BDI-II score. In that case, the experimenter opened the envelope with the questionnaires and discussed the BDI-II with the participant. Consequently, during debriefing but after all experimental tasks were completed, the experimenters were not blind to some participants' level of dysphoria anymore.

**ROAM: memory cue selection.** During Session 1, participants selected five negative, five positive, and three neutral specific autobiographical memories. The experimenter instructed participants to select specific memories that they had experienced personally, that were emotionally meaningful to them, that happened at a particular place and time, and that did not last longer than a day[1,28,33]. The memories needed to be at least one day old[11,26,98], but not older than five years[98]. For each memory, participants noted three cue words on a paper sheet that unambiguously referred to the memory, resulting in 13 sets of three cue words[10,99]. We asked for three cue words to guarantee that the memory could be unambiguously retrieved during Session 2, while at the same time avoiding the generation of complete sentences that could guide or intervene with the free memory recall. Next to the memory cues, participants indicated the age of each memory in months. If the memory was younger than one month, they indicated the memory age in weeks. Participants were given as much time to complete the memory selection task as they needed. The order of the memory selection sheet was counterbalanced across participants (i.e., whether positive or negative memories were selected first, neutral memories were always selected last). An example for a memory selection sheet can be found in Supplementary Note 10.

**ROAM: memory re-experience.** During session 2, participants were administered the key component of the ROAM, to characterize core qualities of memory retrieval, such as episodic detail and psychophysiological expression of affect. They were presented with all thirteen sets of three memory cue words that they had selected during session 1. Fig. 1a provides a schematic representation of a trial. Participants were instructed to retrieve the memory that the cue words referred to as vividly as possible. Each set of three memory cues was preceded by an 8 s fixation cross. The cue words were presented on screen for 10 s, followed by 50 s without memory cue but with the instruction to keep thinking back to the memory. In sum, participants mentally relived each memory for a total of 60 s. After this period of silent recall, they verbally recounted their memories for 90 s, which was audio-recorded. Participants were instructed that they were allowed to stop recounting the memory before the 90 s expired if they did not recall additional information. In this case, they were instructed to sit still and wait until the verbal recall phase is over.

Following the verbal recall of each memory, participants answered several self-paced questions regarding the memory that were displayed on the computer screen. First, they indicated how the memory made them feel on a visual analog scale (VAS) from 'negative' to 'positive', and a tick-mark in the center to indicate 'neutral' (*memory valence*) and how the memory made them feel on a VAS from 'calm' to 'aroused' (*memory arousal*). Then, they indicated how vividly they could remember the memory from 'not at all' to 'very well' (*memory vividness*). Following, participants indicated whether they remembered the memory from a field (first person) or from an observer perspective (third person; *memory perspective*)[17,100]. Finally, they indicated how often they think back to the memory (*retrieval frequency*)[101], and how often they share the memory with others (*sharing frequency*)[102] on VAS scales ranging from 'never' to 'very often'.

The thirteen memory cue-sets were presented in semi-random order. The first three trials contained one negative, one positive, and one neutral trial. Trials four to eight and trials nine to thirteen each contained two negative trials, two positive trials, and one neutral trial. Within these blocks, the memory cues were presented in random

order. Between trials, there was a black screen for 14, 15 or 16 s (on average 15), during which participants could relax and were given time to return to an emotionally neutral baseline. After eight trials, there was a 60 s break, during which participants saw a countdown that informed them when the task continued. FEMG responses were recorded during the 60 s silent recall. The 4 s directly preceding retrieval onset were used as baseline.

**Movie clip viewing.** Participants watched two positive, two negative, and two neutrally valanced film excerpts (e.g., a scene from *The Champ*, 1979, in which a boxer dies and his son cries). A description of all movie clips can be found in Supplementary Note 11 (Supplementary Table 17). The movie clips were presented in English. Participants were instructed to imagine themselves as being part of the scene as a bystander or as one of the depicted persons, whatever worked best for them. Each clip was preceded by an 8 s fixation cross and took in between 105 and 164 s. The movie clips were presented in semi-random order, such that the first three trials and the second three trials both contained one movie clip of each valence. After each clip, participants reported how the clip made them feel on a visual analog scale (VAS) from 'negative' to 'positive' with a tick-mark in the center to indicate 'neutral' (*movie valence*). Participants further indicated how the clip made them feel on a VAS from 'calm' to 'aroused' (*movie arousal*), and how well they could imagine themselves in the depicted scene from 'not at all' to 'very well' (*movie immersion*). FEMG responses were recorded over the entire length of each movie clip. The 4 s directly preceding each movie onset were used as baseline.

**Questionnaires.** We collected information on sample characteristics, including age, gender, employment status, place of birth, native language, and English proficiency. We also collected information on personality characteristics including depressive feelings using Beck's Depression Inventory modified[92] (BDI-II), anxiety using the Trait sub-scale of the State-Trait Anxiety Inventory[96] (STAI), the ability to experience pleasure using the Snaith–Hamilton Pleasure Scale[97] (SHAPS, English version), as well as the use of mental imagery using the Spontaneous Use of Imagery Scale[94] (SUIS) and the Plymouth Sensory Imagery Scale[95] (PSI-Q, English version). We also assessed participants' mood at the start and end of each session, using the Positive and Negative Affect Scale (PANAS)[93].

**Additional physiological data collection.** In order to assess responses to the movies and to the retrieved autobiographical memories on other psychophysiological measures than fEMG, we collected skin conductance levels and heart rate (ECG) during the movie and the ROAM tasks. These data and analyses are not presented in this report.

**Sampling plan**

**Sample characteristics.** Participants who scored 16 or higher on the BDI-II as collected during the online screening, were invited to participate and comprised the *dysphoric group*. This was in line with previous studies that employed a cut-off between 14 and 18[103]. Participants in the *non-dysphoric group* scored 3 or lower on the BDI-II. This cut-off has been used in previous studies[100] and should guarantee very low levels of depressive symptoms. The non-dysphoric group was matched with the dysphoric group by age (±2 years) and self-reported sex. We verified the allocation to the dysphoric and non-dysphoric with a pen and paper version of the BDI-II in the beginning of Session 2, because the BDI-II has been validated as a pen and paper questionnaire and because dysphoric symptoms might have changed in the time between the screening and Session 2. Participants whose BDI-II score was no longer 3 or lower or 16 or higher were excluded from the analyses and data of an additional participant was collected instead.

Based on the online screening, we excluded participants who (1) self-reported a prior depressive episode, self-help for depression, or

professional help for depression (e.g., counseling, antidepressant medication), (2) had a current or past diagnosis of a mental disorder, (3) had a current or past neurological illness or injury, (4) had a history of a traumatic experience that still affected them, (5) consumed more than 14 alcoholic drinks per week, (6) consumed recreational drugs more than once per week, (7) used psychoactive drugs (e.g., anxiolytics), (8) had significant visual or hearing impairments that couldn't be corrected, (9) were younger than 16 or older than 35, and (10) did not speak Dutch. By excluding participants who reported a depressive episode or professional help for depression in the past, we aimed to achieve a relatively homogenous sample of participants with dysphoria that experienced prototypic cognitive and affective distortions that were not yet affected by prior depressive episodes or professional interventions. All of the exclusion criteria were assessed again with a pen and paper screening questionnaire upon arrival at the lab. Participants who met any of these criteria were excluded from the study. Further, data from participants were excluded in case of technical issues or a failure to follow instructions that affected eight or more ROAM trials in total, or that affected more than three positive or more than three negative ROAM trials. In such a case, data of an additional participant was collected to reach the final sample size. A flow-chart of participant screening and inclusion is presented in Supplementary Note 4 (Supplementary Fig. 5).

**Sample size.** We employed a modified Sequential Bayes Factor design[104], that is, we planned to collect data until we found convincing evidence for one hypothesis relative to all other hypotheses under investigation (i.e. the largest Posterior Model Probability PostP$_i$ is equal to or larger than 0.80) for each of the primary research questions (Test 1 and Test 2A), or until we reached a maximum sample size of $N = 80$. We commenced by including a minimum of 20 participants per group ($N = 40$) and computing Posterior Model Probabilities for Test 1 and Test 2A. If there was convincing evidence for a specific hypothesis over the other hypotheses within both Test 1 and Test 2A (PostP$_i \geq 0.80$), we would stop data collection at the minimum sample size. Otherwise, we would increase the sample size in incremental steps of $n = 10$ (5 per group) and repeat the testing procedure until PostP$_i \geq 0.80$ or until the maximum sample size $N = 80$ was reached. We defined a high PostP rather than a high BF as stopping criterion, because a PostP quantifies the support in the data for one hypothesis relative to all other hypotheses under investigation (as opposed to a BF that quantifies evidence for one hypothesis relative to one other hypothesis). Therefore, PostP as stopping criterion better fitted our approach to investigate evidence for more than two hypotheses. At $N = 80$, the results were reported regardless of the strength of evidence for each hypothesis. Evidence for the different hypotheses at each of the updating steps is presented in Supplementary Note 4 (Supplementary Fig. 5).

Since earlier studies to overgeneral memory biases in participants with depressive symptoms revealed medium to large effects[1,25] and small effects in the etiology of depression are taken as reason to study other etiological mechanisms in depression[105], we used medium to large effect sizes as starting point for the substantiation of our maximum sample size. Simulation studies provided estimates for the required sample size to detect medium to large effects with Bayesian Informative Testing[106]. Specifically, they have compared the required sample sizes for traditional significance testing and for Bayesian Informative Hypothesis Testing to achieve .80 full support power for an informative hypothesis that specifies the relationship of means for four different groups (for example, m1 > m2 > m3 > m4). In such a between-participants design, traditional significance testing would require at least a total sample size of $N = 360$ (90 per group) or $N = 140$ (35 per group) to detect a medium or large effect, respectively. However, Bayesian Informative Hypothesis Testing would require a total sample size of 92 (23 per group) or 16 (4 per group) to detect a medium

or large effect, respectively. Importantly, these simulations investigated four means in between-group comparisons, whereas we employed a mixed design (including between- and within-participant comparisons). Since within-participants comparisons yield more statistical power compared to between-participants comparisons, we expected to require a lower sample size than suggested by these simulations. As a consequence, a maximum sample size of $N = 60$ (30 per group) may have been sufficient. However, due to risks of overestimation of effect sizes based on earlier studies[107], sample variability, and other unpredictable factors, we tested a larger maximum sample of $N = 80$ (40 per group). This is larger than the sample size of most studies in the field[1,25], indicating that the study would have contributed to the field, even in the case of weak evidence.

The minimum sample size of $n = 20$ per group is often used as an example for the minimum sample size in several methodological articles[63,104] and aligns with the recommendation that multilevel models should comprise a sample of at least 20 clusters on the highest level (in our case 20 individuals per group) to provide unbiased estimates of the variance of fixed effects[108]. Consequently, the minimum sample size was sufficient to yield unbiased and reliable results and would have allowed strong conclusions in case of convincing evidence (note that in the absence of convincing evidence, the sample size was increased).

Most importantly though, even in the case of weak evidence, Bayesian analyses would likely not result in any inconclusive results, as the statistics will always provide evidence in the direction of one hypothesis over the other hypotheses under consideration (except if all Posterior Model Probabilities would be exactly the same). Therefore, we could have interpreted the strength and direction of the evidence, even if the evidence associated with the primary or secondary questions at the maximum sample size was not as strong as desired initially[63,104]. In case the Posterior Model Probabilities of Test 1 and Test 2A indicated that data collection can be stopped before the maximum sample size was reached, it was likely that the strength of the evidence associated with the secondary questions (Test 2B and Test 3) was equally strong or even stronger than evidence associated with the primary questions, since the pilot data indicated that fEMG responses to the movies are generally very strong (essential for Test 2B), and estimating within-participant relationships (Test 3) generally entails more power than between-participants relationships.

### Analysis plan

**Episodic detail – coding.** The audio-recordings of each retrieved memory were transcribed. Every transcription was checked by a second researcher. Based on the transcription, we assessed the amount of episodic and non-episodic details within every memory. We used a coding scheme that was developed for the autobiographical interview by Levine and colleagues[33,61,62]. It distinguishes between internal details that are specific to the remembered event (representing episodic recollection) and external details that are not specific to the remembered event (representing more semantic contributions to the autobiographical memory). The number of episodic details per memory provides an objective quantification of episodic memory retrieval for each memory. Episodic details can be about the unfolding of the event itself (*event* details; "I went to a concert"), the location (*place* details; "in my home town"), the time of the event (*time* details; "it was two years ago"), perceptions during the event (*perceptual* details "I heard how they started to play my favorite song"), or emotions or thoughts during the event (*emotion/thought* details, "I felt so happy"). If any of these details related to the event of interest (the cued memory), it was coded as internal. If it related to a different event, it was coded as external. Additionally, we coded general knowledge or facts (*semantic* details, "Oasis is a pop band"), *repetitions*, and *other* details that cannot be categorized otherwise (for example metacognitive statements such

as "this is all I can remember"). Only episodic information that was internal to the remembered event was considered a detail that represents episodic recollection. We randomly selected 25% of the memories such that they were coded by at least two independent raters to assess interrater reliability.

Since the online screening was conducted by a person who was otherwise not involved in running the experiment, transcribing and coding of the memories, the experimenter, transcriber, and coder were blind to the participants' condition. If a participant wanted to learn about their BDI score or indicated suicidal intentions during the debriefing after the experiment, a different person than the experimenter transcribed and coded the memories to ensure that the transcriber and coder were blind to the participant's condition (see Supplementary Note 12 for a schematic overview of the blinding procedure).

**FEMG acquisition and pre-processing.** For the retrieval part of the ROAM and the movie task in session 1, facial EMG was collected with two pairs of sintered Ag/AgCl EMG electrodes with six mm sensors that were placed in the zygomaticus major and the corrugator supercilii region of the left side of the face, according to established guidelines[109]. A reference electrode was placed below the hairline in the horizontal center of the forehead. Before electrode placement, the participants' skin was cleaned with a face rub gel and alcohol wipes. The electrodes were connected to a custom-made bipolar EMG amplifier with an input resistance of 1 GΩ, an amplification factor of 5200, and a bandwidth of 5–1000 Hz (6 dB/oct). The raw data were sampled at 1000 S/s. Raw data were pre-processed using the in-house analysis program VSRRP (developed by the Technical Support Group Psychology at the University of Amsterdam). First, raw data was offline filtered with a 20 Hz high-pass filter[110], a 50 Hz notch filter, and a 100 Hz notch filter (all 4th order). Then, the data was rectified and integrated using a digital contour follower with a time constant of 25 ms. Subsequently, fEMG data of every trial was down-sampled to 1000 ms segments. All following analysis steps were conducted using R[111] as implemented in RStudio[112], with the packages bain[113] and nlme[114].

EMG data typically includes artifacts due to movement, eye blinks, coughing etcetera. Some of these artifacts (e.g., blinking) were filtered out by the 20 Hz high-pass filter. To minimize the influence of artifacts from other sources (e.g., coughing), we applied automated artifact rejection. Automated artifact rejection is particularly suited for preregistered research because it employs objective criteria that can be specified before data collection and because it is 100% reproducible by other researchers. Specifically, each data segment that deviated 3 or more standard deviations from the mean of all segments was rejected as an artifact and replaced with a missing value. This artifact rejection was applied within participant, within muscle (zygomaticus and corrugator), within session (session 1 and session 2), and within condition (negative, positive, and neutral) across relevant segments. The artifact rejection was applied separately to the baseline data within participant, within muscle, and within session but across conditions. If within a trial more than 50% of the segments during the movie clip or during the 60 s recall phase were missing or rejected, the trial was excluded from all analyses. If more than 50% of the segments during the 4 s baseline were missing or rejected, the trial was also excluded from all analyses. In addition, trials with clear technical issues or a failure of a participant to follow the instructions, were excluded (e.g., trials during which a participant started to talk during the silent recall phase).

For each memory trial in session 2, we calculated the percentage change from baseline (4 s) during the 60 s retrieval phase[115]. Subsequently, we calculated the average of the percentage change during the 60 s retrieval phase per trial. For each movie trial in session 1, we calculated the percentage change from baseline (4 s) over the entire

time-course of the movie. Subsequently, we calculated the average of the percentage change while watching each movie.

**Bayesian informative hypothesis testing.** We weighed evidence for all hypotheses of the manipulation checks and Test 1-3 with Bayesian Informative Hypothesis Testing (BAIT) as implemented in the R package 'bain'[63]. Evidence is presented as Bayes Factors (BF) and Posterior Model Probabilities (PostP). A BF indicates how much more likely it is to observe given data under one hypothesis relative to another hypothesis[63]. Posterior model probabilities quantify the support in the data for one hypothesis relative to other hypotheses under investigation. For example, in the context of a comparison between two groups, one can identify evidence in the data for the alternative hypothesis that one mean is larger than another mean ($\mu_1 > \mu_2$, Ha) compared to the null hypothesis that there is no difference in means ($\mu_1 = \mu_2$, H0). A $BF_{a0} = 4$ would indicate that the data are four times more likely under the alternative hypothesis than under the null hypothesis. The order of the hypotheses can be reversed using the formula $BF_{0a} = 1/BF_{a0} = 1/4 = 0.25$. A $BF_{0a} = 0.25$ would indicate that the data are 0.25 more likely under the null hypothesis than under the alternative hypothesis. Regarding posterior model probabilities, if $PostP_1$ equals 0.80, $PostP_2$ equals 0.05, and $PostP_3$ equals 0.15, then the probability that H1 is the best of these three hypotheses is equal to 0.80. BAIT allows to simultaneously test specific predictions involving more than two means, rendering multiple comparison correction obsolete[63]. For all analyses, we employed the default settings implemented in the bain package[63] with the exception that we used a moderate fraction = 2 of the data to define the prior variance. Additionally, we conducted sensitivity analyses with a more conservative fraction = 1 and a more liberal fraction = 3 to evaluate the influence of the prior variance on our results (Supplementary Note 8).

The bain package uses a normal approximation to the posterior distribution of the means (collected in the vector $\gamma = [y_1, \ldots, y_G]$, where G denotes the number of means) used in the hypotheses under $H_a$, that is $g(\gamma|data, H_a) \approx N(\hat{\gamma}, \sum \gamma)$, where $\hat{\gamma}$ denotes the estimates of the means and $\sum \gamma$ the covariance matrix of the estimates. From this posterior distribution the adjusted fractional prior distribution of $\gamma$ under $H_a$ is derived, that is, $h(\gamma|H_a) \approx N(0, \sum \gamma/(f*b))$, where the prior mean is 'adjusted' to 0 such that the Bayes Factors based on this posterior and prior is consistent, and the prior covariance matrix is based on a 'fraction' $b = J/n_{eff}$, where $n_{eff}$ denotes the effective sample size of the information in the data (see below for an explanation of $n_{eff}$). In bain, a modification of minimal training samples is used, that is, $J$ denotes the number of independent constraints used to specify the hypotheses. The parameter f is used to determine whether once, twice (the reference value used in this paper) or thrice the minimal fraction should be used. Varying f renders a prior sensitivity analysis. The prior distributions corresponding to all other hypotheses are derived by restricting the domain of the prior under $H_a$ in accordance with the hypothesis at hand and renormalizing the density. The Bayes factors presented in this paper are computed using these posterior and prior distributions (for further elaborations and the complete statistical background, see references[116,117]).

All analyses were performed in three steps. First, we estimated a simple multilevel model that included the dependent variable of interest, fixed effects for the predictors of interest, and a random intercept for participant to account for the within-participants design. Second, we extracted the estimated effects of interest from the multilevel model as well as the variance-covariance matrix of these effects. Third, the estimates and variance-covariance matrices were used to evaluate the evidence for each hypothesis with BAIT. For the calculation of evidence in BAIT, we used the effective sample size that accounts for multiple observations within participants[118]. The effective sample size $n_{eff}$ is situated between the number of participants and the

number of observations and is calculated with Eq. (1):

$$n_{eff} = \frac{n}{1 + (n_{cluster} - 1)\text{ICC}} \quad (1)$$

With:

$$\text{ICC} = \frac{s^2_{intercept}}{s^2_{intercept} + s^2_{residual}}$$

In the equation, $n$ is the total number of observations, $n_{cluster}$ is the number of observations within participant. ICC represents the intra-class correlation. $s^2_{int}$ is the variance of the random intercept and $s^2_{res}$ is the variance of the residuals.

**Descriptive statistics.** We provided descriptive statistics on self-reported depressive symptoms (BDI scores), anhedonia (assessed with the SHAPS), participant age, and sex for the dysphoric and the non-dysphoric group to characterize our samples. Moreover, we provided a table with descriptive statistics per condition (positive, negative, neutral) and group (dysphoric, non-dysphoric) for important memory variables collected with the ROAM. Specifically, we reported self-reported valence and arousal of the memories, the number of retrieved episodic and semantic details, memory age, and memory vividness. For all of these variables, we used BAIT to test whether there are differences between the dysphoric and non-dysphoric group that need to be taken into consideration when interpreting our results. Furthermore, we provided descriptive statistics of the fEMG responses to the movie clips as well as self-reported movie valence and arousal.

**Manipulation checks.** We tested whether the retrieval of positive and negative autobiographical memories elicited affective responses that can be measured with fEMG. Since individuals with dysphoria were expected to experience affective memory distortions, the effectiveness of our manipulation as implemented by the ROAM was only evaluated within the non-dysphoric group. However, we conducted the same analyses within the dysphoric group to draw a comprehensive picture of how individuals with and without dysphoria re-experience autobiographical memories. The manipulation checks were similar to the analyses of the pilot data. Given the evidence in the pilot data that positive and negative memories elicit zygomaticus and corrugator activity, respectively, we expected our manipulation to be successful. Design Table 1 (Supplementary Table 18, design tables for all confirmatory analyses are presented in Supplementary Note 13) provides a comprehensive overview of the manipulation checks. We compared evidence for the hypotheses that individuals without dysphoria smiled more when reliving positive memories than during baseline (manipulation check 1.1 – Hypothesis 1), that they smiled less when reliving positive memories than during baseline (manipulation check 1.1 – Hypothesis 2), and that they smiled equally when reliving positive memories and during baseline (manipulation check 1.1 – Null Hypothesis). Further, we tested whether zygomaticus responses during retrieval were specific for positive memories. We compared evidence for the hypotheses that individuals without dysphoria smiled more when reliving positive memories than when remembering neutral memories (manipulation check 1.2 – Hypothesis 1), that they smiled less when reliving positive memories than when remembering neutral memories (manipulation check 1.2 – Hypothesis 2) and that they smiled equally when reliving positive memories compared to neutral memories. We conducted similar manipulation checks for corrugator activity during negative memory retrieval (manipulation checks 2.1 and 2.2).

**Test 1 – episodic memory detail.** We investigated episodic memory distortions in individuals with dysphoria compared to individuals

without dysphoria. Design Table 2 (Supplementary Table 19) provides a comprehensive overview of the confirmatory analyses. We compared evidence for four competing hypotheses about the amount of episodic detail that is retrieved when reliving autobiographical memories. Test 1 – H1: Individuals with dysphoria retrieve fewer episodic detail when reliving positive memories and when reliving negative memories, compared to individuals without dysphoria. Test 1 – H2: Individuals with dysphoria retrieve fewer episodic detail when reliving positive memories but more episodic detail when reliving negative memories, compared to individuals without dysphoria. Test 1 – H0: Individuals with dysphoria retrieve the same amount of episodic detail when reliving positive and negative memories, compared to individuals without dysphoria. Test 1 – Hu: None of the other hypotheses describes the data well.

**Test 2A – Affective responses to memories.** We investigated affective memory distortions of individuals with dysphoria compared to individuals without dysphoria (Design Table 3, Supplementary Table 20). The primary dependent variable was baseline-corrected zygomaticus and corrugator activity for positive and negative memories, respectively. Additionally, we repeated these analyses with subjective valence ratings as complementary dependent variable. We compared four competing hypotheses about affective responses to autobiographical memories. Test 2A – H1: Individuals with dysphoria experience diminished positive affect when remembering positive memories and diminished negative affect when remembering negative memories, compared to individuals without dysphoria. Test 2A – H2: Individuals with dysphoria experience diminished positive affect when remembering positive memories and enhanced negative affect when remembering negative memories, compared to individuals without dysphoria. Test 2A – H0: Individuals with dysphoria experience normal positive affect when remembering positive memories and normal negative affect when remembering negative memories, compared to individuals without dysphoria. Test 2 – Hu: None of the other hypotheses describes the data well.

**Test 2B – affective responses to movies.** We further tested whether the pattern of altered affective responses in dysphoria is specific for memories or whether it is already present during the encoding of novel events[12]. Design Table 3 (Supplementary Table 20) provides a detailed overview of all analyses for Test 2B depending on the outcome of Test 2A. Specifically, we assessed whether individuals with dysphoria showed the same pattern of affective responses to new events (movie clips) as for autobiographical memories (Test 2B – H1) or a different pattern (Test 2B – Hc). As a consequence, Test 2B depended on the outcome of Test 2A. For the case that Test 2A provided evidence for a negativity bias, we planned to compare evidence for the hypothesis that individuals with dysphoria show reduced affective responses to positive movies but enhanced affective responses to negative movies compared to individuals without dysphoria (Test 2B – H1) and evidence for the hypothesis that individuals with dysphoria show any other pattern of affective responses to movies (Test 2B – Hc). If Test 2A provided evidence for the null hypothesis or the complement hypothesis, we planned to not conduct Test 2B as a confirmatory analysis.

**Test 3 – the relationship between episodic detail and affective responses.** We investigated whether higher retrieved episodic detail relates to stronger affective responses. We further tested whether the relationship between episodic detail and affective responses differed between individuals with and without dysphoria. We conducted these analyses separately for positive and negative memories (since these data are collected with different muscles), using the exact same approach. Design Table 4 (Supplementary Table 21) provides a comprehensive overview of the analytical approach. In short, we

calculated the grand-mean centered cluster mean of episodic detail for each participant (between person predictor) and we calculated the cluster mean centered episodic detail for each memory (within person predictor). We estimated a multilevel model, where the affective response of participant $i$ to memory $j$ was calculated with Eq. (2):

$$
\begin{aligned}
\text{affective\_response}_{i,j} = {} & \beta_1(\text{group})_{1i} + \beta_2(\text{group})_{2i} + \beta_3(\overline{\text{episodic\_detail}})_i(\text{group})_{1i} \\
& + \beta_4(\overline{\text{episodic\_detail}})_i(\text{group})_{2i} + \gamma_1(\text{episodic\_detail})_{ij}^c(\text{group})_{1i} \\
& + \gamma_2(\text{episodic\_detail})_{ij}^c(\text{group})_{2i} + \varepsilon_{ij} + U_i
\end{aligned}
$$

$$(2)$$

With:
$(\text{group})_{1i} = 1$ if dysphoric
$(\text{group})_{1i} = 0$ if non-dysphoric
$(\text{group})_{2i} = 0$ if dysphoric
$(\text{group})_{2i} = 1$ if non-dysphoric

This model was assessed separately for positive (Test 3.1) and negative memories (Test 3.2). For positive memories affective_response represents baseline-corrected zygomaticus activity and for negative memories affective_response represents baseline-corrected corrugator activity. Additionally, we repeated these analyses with subjective valence ratings as a complementary outcome measure. The factor dysphoric versus non-dysphoric is represented as (*group*). $\beta_1$ and $\beta_2$ represent the estimated *affective response* to a memory in the dysphoric group and non-dysphoric group, respectively, if $(\overline{\text{episodic\_detail}})_i$ and $(\text{episodic\_detail})_i^c$ are zero. $\beta_3$ and $\beta_4$ represent the linear effects of $(\overline{\text{episodic\_detail}})_i$ on affective_response in the dysphoric and non-dysphoric group, respectively. $\gamma_1$ and $\gamma_2$ represent the linear effects of $(\text{episodic\_detail})_{ij}^c$ on affective_response in the dysphoric and non-dysphoric group, respectively. The superscript c indicates that the variable (episodic_detail) is participant mean-centered. $(\overline{\text{episodic\_detail}})$ represents the grand-mean centered mean for each individual. The residual is indicated by $\varepsilon_{ij}$. $U_i$ represents a random intercept of a participant.

The estimated coefficients of interest and their variance-covariance matrix were extracted from this multilevel model for each separate Bayesian analysis in bain[63]. First, we tested whether episodic detail positively predicted the affective response on a trial-by-trial base. Specifically, we investigated the relationship between participant mean-centered episodic detail and affective response, separately for individuals with and without dysphoria (Test 3.1A for positive and Test 3.2A for negative memories). We compared evidence for the hypotheses that $\gamma_1$ (or $\gamma_2$ for individuals without dysphoria), is larger than 0 (H1: $\gamma > 0$), smaller than 0 (H1: $\gamma < 0$), or equal to 0 (H1: $\gamma = 0$). Second, we assessed whether the relationship between episodic detail and affective response differed between individuals with and without dysphoria (Test 3.1B for positive and Test 3.2B for negative memories). Specifically, we compared evidence for the hypotheses that the linear effect $\gamma_1$ of (episodic_detail)$^c$ on affective_response in the dysphoric group is larger than, smaller than, or equal to the linear effect $\gamma_2$ of (episodic_detail)$^c$ in the non-dysphoric group (H1: $\gamma_1 > \gamma_2$, H2: $\gamma_1 < \gamma_2$, H0: $\gamma_1 = \gamma_2$).

We additionally tested whether participants that on average retrieve more episodic detail, showed stronger affective responses to memories (Test 3.1A for positive and Test 3.2A for negative memories). These analyses were similar to test 3.1 and 3.2, but the hypotheses referred to the $\beta$ coefficients instead of the $\gamma$ coefficients, and to the participant mean of episodic detail (i.e. the $(\overline{\text{episodic\_detail}})_i$) instead of the participant mean-centered episodic detail (episodic_detail)$^c$. The analyses of the within-person regression coefficients ($\gamma$) provided insights into the relationship of episodic detail and affective responses within a person, whereas the analyses of the between-person coefficients ($\beta$) provided insights into individual differences.

## Protocol registration

The study design, materials, and data analysis plan were registered before data collection. The protocol is available on Figshare (https://doi.org/10.6084/m9.figshare.14605374.v1)[91] and on the Open Science Framework (https://doi.org/10.17605/OSF.IO/3D9AN).

## Reporting summary

Further information on research design is available in the Nature Portfolio Reporting Summary linked to this article.

## Data availability

All raw and processed data as well as study materials are publicly available on the Open Science Framework (https://doi.org/10.17605/OSF.IO/Y7346)[119] as open data with some exceptions: We will not share highly sensitive data such as audio recordings and transcriptions of the personal autobiographical memories. Personal memories often cannot be completely anonymized and sharing them online without restrictions would violate the participants' privacy. These data are stored on secured servers of the University of Amsterdam. Should other researchers need access to these sensitive data, they must make a formal request to the ethics committee of the University of Amsterdam. However, for transparency and reproducibility, we offer exemplary memories of participants who explicitly consented that their memories may be shared. We also do not share materials that are subject to copyright by third parties such as movie clips. For materials that cannot be shared, we provided detailed descriptions that allow to reproduce them.

## Code availability

All analysis code for R is publicly available (https://doi.org/10.17605/OSF.IO/Y7346)[119]. Only the conversion from raw data to processible data is conducted with the custom in-house program VSRRP (developed by the Technical Support Group Psychology at the University of Amsterdam). The (C/C++) source code of VSRRP that executes this conversion is available upon request from the Technical Support Group Psychology at the University of Amsterdam (https://lab-fmg.uva.nl/contact/contact.html) because it is not property of the authors of this manuscript.

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

## Acknowledgements

This research project is supported by a Research Talent grant (grant number: 406.17.564; awarded to S.D., V.v.A., M.K.), awarded by the Netherlands Organisation for Scientific Research (NWO). Vanessa van Ast was supported by a Veni grant (grant number: 451.16.021; V.v.A.), awarded by NWO. Merel Kindt was supported by an ERC Advanced Grant (grant number: 743263; M.K.), awarded by the European Research Council. The funders had no role in study design, data collection and analysis, decision to publish or preparation of the manuscript. We sincerely thank Susanne Schulz and Jamie Elsey for inspiring discussions, feedback, and proof-reading versions of this manuscript, as well as Bert Molenkamp for technical assistance with the psychophysiological measures. Moreover, we thank Donna Meyer, Roxanne Bongers, and Thomas Willems for their tremendous help with recruitment and data collection, as well as Rosa Breed and Femke Verbeten for their help with transcribing memory recordings. Finally, we thank all participants who so openly shared their personal memories with us.

## Author contributions

S.D.: conceptualization, data curation, formal analysis, funding acquisition, investigation, methodology, project administration, resources, software, validation, visualization, writing – original draft, writing – review & editing. L.K.: conceptualization, data curation, investigation, validation, writing – review & editing. H.H: formal analysis, methodology, software, writing – review & editing. M.K.: conceptualization, funding acquisition, supervision, writing – review & editing. V.v.A.: conceptualization, funding acquisition, methodology, project administration, resources, supervision, writing – original draft, writing – review & editing.

## Competing interests
The authors declare no competing interests.
