## [Peer Review File · Nature Communications]

Review Study Plan (Stage 1)

Reviewers' Comments:

Reviewer #1:

Remarks to the Author:

Review of "Episodic and affective memory distortions in dysphoria: Bayesian testing of diverging theories" for *Nature Communications*

This paper outlines plans for a proposed study in which dysphoric and non-dysphoric (assessed via scores on an online BDI-II questionnaire) participants would in a first session generate five positive, five negative and three autobiographical memories and three cue words per memory, and then complete questionnaires, finally followed by watching emotional movie clips while psychophysiological signals (fEMG, skin conductance, and electrocardiography) are collected. In a second session, they will silently recall each memory for 60 s in response to a memory cue and then verbally recount each memory for 90 s, then rate the memory for valence, arousal and vividness. They will also assess memory perspective, retrieval frequency and sharing frequency. Audio recordings will be coded to assess the amount of episodic and non-episodic details for each memory.

The objective of this testing is to test the general assumption that depression is associated with overly general (i.e., lacking in details) memories. They will pit this theory against another idea in the field, that depressed individuals have a negative bias in their attention and memory that leads them to be more likely to retrieve and elaborate on negative than positive memories. Both theories predict that dysphoric individuals should retrieve positive memories with reduced episodic detail and reduced positive affect. However, they make opposing predictions about recall of negative memories. This is an interesting and important question for the field, as it has implications for potential therapeutic approaches as well as for theoretical frameworks about depression and memory.

The investigators propose collecting data until there is strong evidence for one hypothesis or when they reach a maximum sample size of $N = 60$. The *Nature Communications* journal guidelines for methods involving Bayesian hypothesis testing for registered reports state that, "Authors with resource limitations are permitted to specify a maximum feasible sample size at which data collection must cease regardless of the Bayes factor; however to be eligible for advance acceptance this number must be sufficiently large that inconclusive results at this sample size would nevertheless be an important message for the field." The proposed maximum sample size provides relatively low power. For instance, a simple t-test comparing number of episodic memories between the two groups would need to have 95 participants per group to have 80% power to detect differences between groups (Cohen, 1992). The authors have not reported on the effect size expected from prior literature, but this is a key issue for this preregistered report that was not satisfactorily addressed. It is also important to provide justification for the minimum N at which statistical testing will start and the potential to end the study will start (Schönbrodt & Wagenmakers, 2017). Also, the authors state that they "will collect data until there is strong evidence for one hypothesis within Test 1 and Test 2A" - what does this mean about all the other tests they have planned? Does having sufficient data not matter for these questions or are they for some reason assumed to have higher power?

Participants will be informed that they should retrieve positive and negative memories that are at least a week old but not older than five years. The neutral memories, however, only need to be older than one day while still not older than five years. The authors state that this is because neutral memories are typically forgotten quickly. However, this is problematic for allowing for clear conclusions when comparing emotional and neutral memories, as they may differ in their level of detail due to the age of the memories rather than their emotional nature. In particular, the difference in instructions between the types of memories may suggest to participants that more recent memories are expected for neutral memories.

It is not clear who will conduct the transcriptions and coding of the audio recordings and whether they will be blind to condition or whether they will be the same researchers who ran the sessions and assessed the BDI-II with the participants at the end.

Mara Mather
(signed review)

Reviewer #2:

Remarks to the Author:

Thank you for the opportunity to review this interesting study. Simultaneous consideration of overgeneral and negative memory biases will have both theoretical and clinical implications, and the authors have articulated clear hypotheses made by each of these theories. Key strengths of the study are the sampling plan and analysis approach, which will be easily reproducible, and good consideration of clinical risk issues. The combination of physiological and behavioural measures is critically important in this type of research. However, when examining cognitive processes it is important to ensure that there are established links between the behavioural and physiological measures you intend to use to measure the same construct.

A key concern I have is that there does not appear to be a pre-established association between the strength of facial electromyography (fEMG) activity and the intensity of affect? That is, the pilot data seem to suggest that mentally well individuals frown more/have higher corrugator activity when recalling negative memories relative to neutral memories, and smile more/have higher zygomaticus activity when recalling positive memories relative to neutral. However, your key hypotheses centre on the intensity of affect. If this (apparently previously unestablished) relationship between your physiological measure and construct of interest is not evident, you will be unable to assess your key research questions.

To put this in context, there is no clear rationale/prior evidence presented for why there would be a linear relationship between the strength of fEMG activity and the phenomenological intensity of negative affect experienced by the individual. Surely it is possible to experience intense negative affect without a more intense/deeper frown (at least this would seem to be the case when working clinically with patients experiencing strong negative affect, who often appear quite blank when reporting intensively negative memories). Further justification of the construct validity of your psychophysiological measure of affect intensity is needed.

Only completing a manipulation check (page 20) for the non-dysphoric group is somewhat concerning, in light of the above, and as the method does not seem to have been used in clinical samples before (the studies referenced were done with healthy samples – and one study only had female participants). Consideration of whether factors such as psychomotor retardation (a symptom of depression) would impact fEMG measurements seems important.

The experimental design also seems unable to establish the direction of the relationship – i.e., that episodic detail causes increased intensity of affect measured via fEMG, which is what the Introduction seems to suggest is a vitally needed step in evaluating current theories (e.g., 'Consequently, it is not clear whether affective distortions co-occur with, or are even driven by, episodic distortions.') Indeed, you hypothesise that episodic detail positively predicts the affective response on a trial-by-trial basis. However, your design sees that you are only able to analyse cross-sectional associations. Prior experiments have manipulated the amount of episodic detail with which memories are recalled to try and get at this issue.

More minor comments:

- Methods -

Exclusion/inclusion criteria require further clarification. You have stated that you will screen out those with a history of psychiatric disorder (page 17) but a BDI of over 16 would indicate the likely presence of disorder, even if there has been no formal diagnosis.

What is the reasoning for not reporting the obtained skin conductance levels and heart rate (ECG) during the movie and the ROAM tasks? These measures would provide valuable data and may help address the intensity of affect issue I have raised.

Will data from memories that are not specific be excluded?

I suggest you increase the number of memories that are second rated for episodic detail to a minimum of 25% as this is a key outcome of interest.

Including a measure of anhedonia would also be informative, and may help to explore whether fEMG activity is associated with the phenomenological experience of affect.

Why is fEMG activity being baseline corrected for memory recall but not movie clips?

- Introduction -

More needs to be done to emphasise why we can't rely on self-report of affect. From a clinical perspective, it is the patient's felt experience of the emotion which is important, as it is this felt experience which drives their decision-making and behaviour.

No clear distinction is made between the prior experience of depression and dysphoria – individuals remitted from depression may be dysphoric, but individuals with no prior history of depression may also be dysphoric. When it comes to cognitive factors that drive disorder, this is an important distinction. If there is a prior history of depression, cognitive distortions may have emerged during that prior episode, indicating that the cognitive factor may not be a primary predictor of symptoms, but rather a scar from a prior experience of depression, and thereby only a secondary risk factor (i.e., for relapse of depression).

It would be useful to highlight that a combination of self-report and neurobiological approaches have been used to explore how the experience of affect differs between normative, and depressed or high-risk samples, e.g., <https://jamanetwork.com/journals/jamapsychiatry/article-abstract/1688033>
<https://ajp.psychiatryonline.org/doi/full/10.1176/appi.ajp.2015.15010119>

Reviewer #3:

Remarks to the Author:

The current pilot study addresses the effect of autobiographical memory content with positive or negative affective color or emotional movie clips on EMG responses of two facial muscles: corrugator supercilii, presumed to indicate negative affective responses, and zygomaticus major, presumed to indicate positive affective responses. Participants were asked to retrieve positive, negative, or neutral experiences from their autobiographic memory and to label these memories using key words. During an experimental session, an 8-s lasting fixation cross was followed by a 10 s lasting presentation of cue words, representing a particular memory experience, and a 50-s period during which participants silently retrieved autobiographical memories in response to these cues. During another experimental condition, participants were presented an 8-s lasting fixation cross followed by the presentation of a positive, negative, or neutral movie clip during which participants had to imagine themselves as being part of the scene. These clips were of variable duration.

EMG measurements during autobiographical memory tasks were either baseline-corrected or not corrected. In case of baseline correction, a baseline EMG value determined during a 4-s period preceding the 10-s presentation of cue words was subtracted from the values obtained during the subsequent 60-s autobiographical memory task. If no baseline correction was applied, raw EMG amplitude responses were quantified from the beginning of the 60-s task period. EMG responses to movies were not baseline-corrected.

I have several problems with this study, in particular with the methodology of the EMG measurements:

(1) EMG responses of both muscles were averaged across different stimuli within subjects and subsequently across subjects. Such averaging is only meaningful if responses are adequately standardized so that they can be compared between different stimuli presented to a subject, between different experimental conditions to which a subject is exposed, or between different subjects. Such standardization also allows a comparison between different muscles. The problem of unstandardized EMG responses is that such comparisons cannot be made due to a multitude of

variables influencing absolute EMG amplitude in terms of microvolts. For example, EMG amplitude may considerably differ between individuals, is on average larger in males than in females, is larger in younger than in older persons, and may differ between different recording sessions performed on the same individuals. Absolute EMG amplitude also depends on the exact location of the recording electrodes. Removing electrodes from the face and placing them again on the same location may result in a different amplitude. Therefore, EMG responses should be standardized. The authors performed such a standardization for autobiographical memory tasks in pilot 3 (but not for tasks in pilots 1 or 2, neither for movies). As stated above, they expressed EMG values during the 60-s memory task as difference scores with the mean value during a 4-s period preceding the task. Within the behavioral sciences, calculating such difference scores in terms of microvolts between experimental and baseline conditions is common practice for EMG responses. Unfortunately, such interval scale scores cannot be considered an adequate standardization of EMG responses since these are still expressed in terms of microvolts, and are thus dependent on anatomical and physiological differences between muscles. Because during a state of complete muscle relaxation EMG activity has a zero amplitude it should be expressed on a ratio scale with a zero origin. Using a ratio scale implies that EMG responses should be expressed as a percentage of baseline activity rather than as a difference score. This method is common practice within medical or physiological disciplines (Merletti, R., 1999, Standards for reporting EMG data. International Society of Electrophysiology and Kinesiology. *J. Electromyogr. Kinesiol.*, 9, III-IV, 1999). EMG responses of a specific muscle in terms of percentage scores can be averaged across stimuli, experimental conditions, or subjects. This method should also be used in the current study.

(2) A basic problem of recording facial EMG activity is crosstalk from other muscles. To reduce such interfering effects, electrodes with small contact areas should be used which are placed at close distance from each other. Information about these details is not provided in the current manuscript.

(3) The authors applied automated artifact rejection implying that data segments deviating 3 or more standard deviations from the mean of all segments were rejected and were considered as missing values. Such a statistical procedure is not uncommon within the behavioral sciences. However, it is not an accurate procedure. Facial EMG responses may be very dynamic. For example, an emotional event or experience may be accompanied by a sudden short-lasting increase in EMG activity of the involved muscle. Removing such responses merely based on statistical criteria might bias the results. I therefore recommend that artifact removal should be based on visual inspection of raw EMG data by an expert having experience with such data.

Reviewer #4:

Remarks to the Author:

This proposal seeks to provide a direct test of whether the theory of overgeneral memory bias or the theory of negativity bias can better account for various discrepant behaviors between dysphoric and non-dysphoric individuals. This is not my specific research area, but I was asked to evaluate the proposed analysis as my expertise is in Bayesian statistics.

Informed Bayesian hypothesis tests are to be performed here, wherein (often multiple) parameter constraints are specified on the statistical model to represent the predictions from a given theory. Then evidence from the data is used to evaluate how much each set of predictions is supported to produce Bayes factors and posterior model probabilities. This would all be done using the *bain* r package, which is an excellent piece of software.

The analysis is well described, and I found the tables to be especially informative. I was glad to see the Bayesian framework leveraged to its fullest extent with such informative hypotheses implemented, including both sets of directional contrasts and point hypotheses where appropriate. I am also glad to see a catch-all "none of the above hypothesis" to account for potential theoretical misspecification (i.e., one theory could be supported over another, while both are poor accounts of the data).

Overall, the proposed analysis looks quite sound. The only potential (minor) qualm I would raise is with the sampling plan. The proposed stopping rule is to collect data until Bayes factors of 10 or more are found. But, with these multiple compound-directional hypotheses it is entirely possible to

find medium-to-strong Bayes factors without obtaining correspondingly high posterior model probabilities (because the prior probability of each hypothesis decreases the more hypotheses are being tested). For example, in the pilot analysis manipulation checks, there are cases where one model is obtaining a high Bayes factor in its favor (e.g. $BF > 20$) but only attains a posterior probability of $\sim .65$. So, the *evidence* can be strong while there is simultaneously much uncertainty remaining about which hypothesis is true. It has become standard to use the Bayes factor as a stopping criterion when testing a simple focal hypothesis, but perhaps achieving high posterior model probability would be a more useful one here? There is no right answer and it is ultimately up to the authors to decide, but I thought I might bring the issue to attention.

Rebuttal letter concerning the manuscript "Episodic and affective memory distortions in dysphoria: Bayesian testing of diverging theories"

We would like to thank the editor and reviewers for their thoughtful and very constructive comments. We appreciate that the reviewers evaluated the research question as important for the field and the proposed analytical plan as one of the key strengths of our study. Below, we have responded to each point raised by the reviewers and listed changes that we have made in the manuscript.

Reviewer #1 (Remarks to the Author):

Review of "Episodic and affective memory distortions in dysphoria: Bayesian testing of diverging theories" for Nature Communications

This paper outlines plans for a proposed study in which dysphoric and non-dysphoric (assessed via scores on an online BDI-II questionnaire) participants would in a first session generate five positive, five negative and three autobiographical memories and three cue words per memory, and then complete questionnaires, finally followed by watching emotional movie clips while psychophysiological signals (fEMG, skin conductance, and electrocardiography) are collected. In a second session, they will silently recall each memory for 60 s in response to a memory cue and then verbally recount each memory for 90 s, then rate the memory for valence, arousal and vividness. They will also assess memory perspective, retrieval frequency and sharing frequency. Audio recordings will be coded to assess the amount of episodic and non-episodic details for each memory.

The objective of this testing is to test the general assumption that depression is associated with overly general (i.e., lacking in details) memories. They will pit this theory against another idea in the field, that depressed individuals have a negative bias in their attention and memory that leads them to be more likely to retrieve and elaborate on negative than positive memories. Both theories predict that dysphoric individuals should retrieve positive memories with reduced episodic detail and reduced positive affect. However, they make opposing predictions about recall of negative memories. This is an interesting and important question for the field, as it has implications for potential therapeutic approaches as well as for theoretical frameworks about depression and memory.

We're grateful for the on-point summary of our proposed study and the positive comments on the importance of our study objective.

Reviewer 1 – comment 1:

The investigators propose collecting data until there is strong evidence for one hypothesis or when they reach a maximum sample size of $N = 60$. The Nature Communications journal guidelines for methods involving Bayesian hypothesis testing for registered reports state that, "Authors with resource limitations are permitted to specify a maximum feasible sample size at which data collection must cease regardless of the Bayes factor; however to be eligible for advance acceptance this number must be sufficiently large that inconclusive results at this sample size would nevertheless be an important message for the field." The proposed maximum sample size provides relatively low power. For instance, a simple t-test comparing number of episodic memories between the two groups would need to have 95 participants per group to have 80% power to detect differences between groups (Cohen, 1992). The authors have not reported on the effect size expected from prior literature, but this is a key issue for this preregistered report that was not satisfactorily addressed.

We are grateful for this important point. We did not include a formal power analysis because such an analysis is currently not possible for Bayesian tests of informative hypotheses that comprise more than one prediction (i.e., more than one $>$ or $<$). Nevertheless, we do agree that it is important to include a rationale for the determination of our minimum and maximum sample sizes. The power for our study is considerably higher than a classic power analysis for a simple t-test based on a medium effect size would suggest. Our confirmatory analyses consist of very specific directed hypotheses. Already within a classic t-test, a directed hypothesis increases power (because alpha changes from .05 to .10). Our confirmatory analyses comprise sets of multiple specific directed predictions (i.e., more than one $>$ or $<$). By including more than one constraint, power increases rapidly (Klugkist, Post, Haarhuis, & Van Wesel, 2014; Kuiper & Hoijtink, 2010).

But indeed, the question remains, how many participants are needed to achieve sufficient power using Bayesian Informative Hypothesis Testing, based on effect sizes that are expected from prior literature. It is not possible to provide an exact estimate of the expected effect size because we compare evidence from different theories and empirical approaches. Nonetheless, as a starting point, we base our sample size considerations on medium to large effects for two reasons. First, studies of overgeneral memory biases in participants with depressive symptoms compared to healthy controls have revealed medium to large effects with an average Hedge's g of approximately 1 (Liu, Li, Xiao, Yang, & Jiang, 2013, abstract and figure 1 & 2; Söderlund et al., 2014, p. 56; Williams et al., 2007, table 1). Second, one might argue that only medium to large effects are worthy further investigation (suggesting pronounced cognitive and affective biases), whereas small effects might indicate that future research should focus on other factors in the etiology of depression (Lakens, 2021).

Stimulation studies provided estimates for the required sample size to detect medium to large effects with Bayesian Informative Testing (Klugkist et al., 2014). Specifically, they have compared the required sample sizes for traditional significance testing and for Bayesian Informative Hypothesis Testing to achieve .80 full support power for an informative hypothesis that specifies the relationship of four group means for four different groups (for example, $m_1 > m_2 > m_3 > m_4$). In such a between-subjects design, traditional significance testing would require at least a total sample size of $N = 360$ (90 per group) or $N = 140$ (35 per group) to detect a medium or large effect, respectively. However, Bayesian Informative Hypothesis Testing would require a total sample size of 92 (23 per group) or 16 (4 per group) to detect a medium or large effect, respectively. Importantly, these simulations investigated four means in between-group comparisons, whereas we employ a mixed design (including between- and within-subject comparisons). Since within-subjects comparisons yield more statistical power compared to between-subjects comparisons, we require a somewhat lower sample size than suggested by these simulations (see for example <http://daniellakens.blogspot.com/2016/11/why-within-subject-designs-require-less.html>). As a consequence, we initially assumed a maximum sample size of $N = 60$ (30 per group) to be sufficiently large. However, due to risks of overestimation of effect sizes based on earlier studies (Gelman & Carlin, 2014), sample variability, and other unpredictable factors, it is better to increase the maximum sample size to $N = 80$ (40 per group). This is larger than the sample size of most studies in the field (Liu et al., 2013; Williams et al., 2007), indicating that the study would contribute to the field, even in the case of weak evidence.

Most importantly though, even in the case of weak evidence, Bayesian analyses will likely not result in any inconclusive results, as the statistics will always provide evidence in the direction of one hypothesis over another hypothesis (except if the Bayes Factor is exactly 1). Therefore, even if we do not reach a $PostP_i$ of .80 at our maximum sample size, the evidence may not be as strong as desired initially, but the direction and the strength of the evidence can still be interpreted (Schönbrodt & Wagenmakers, 2018, p. 132). For example, a Bayes Factor $BF = 3$ is not as strong evidence as we hope to achieve, but it still means that given the collected data a specific hypothesis is three times more likely than its complement hypothesis.

We have now further clarified our rationale for the maximum sample size in the methods. Note that, in line with recommendations by reviewer 4 and the editor, we used the Posterior Model Probability ($PostP_i$) as a stopping criterion for our study instead of $BF > 10$ as described in the original submission.

Methods, p. 18 & 19:

"We will employ a modified Sequential Bayes Factor design⁷⁹, that is, we will collect data until we find convincing evidence for one hypothesis relative to all other hypotheses under investigation (i.e., the largest Posterior Model Probability $PostP_i$ is equal to or larger than .80) for each of the primary research questions (Test 1 and Test 2A), or until we reach a maximum sample size of $N = 80$. We will commence by including a minimum of 20 participants per group ($N=40$) and compute Posterior Model Probabilities for Test 1 and Test 2A. If there is convincing evidence for a specific hypothesis over the other hypotheses within both Test 1 and Test 2A ($PostP_i \geq .80$), we will stop data collection at the minimum sample size. Otherwise, we will increase the sample size in incremental steps of $n = 10$ (5 per group) and repeat the testing procedure until $PostP_i \geq .80$ or until the maximum sample size $N = 80$ is reached. At $N = 80$, the results will be reported regardless of the strength of evidence for each hypothesis.

Since earlier studies to overgeneral memory biases in participants with depressive symptoms revealed medium to large effects^{1,25} and small effects in the etiology of depression are taken as reason to study other etiological mechanisms in depression⁸⁰, we used medium to large effect sizes as starting point for the substantiation of our maximum sample size. Simulation studies provide estimates for the required sample size to detect medium to large effects with

*Bayesian Informative Testing*⁸¹. Specifically, they have compared the required sample sizes for traditional significance testing and for Bayesian Informative Hypothesis Testing to achieve .80 full support power for an informative hypothesis that specifies the relationship of means for four different groups (for example, $m_1 > m_2 > m_3 > m_4$). In such a between-subjects design, traditional significance testing would require at least a total sample size of $N = 360$ (90 per group) or $N = 140$ (35 per group) to detect a medium or large effect, respectively. However, Bayesian Informative Hypothesis Testing would require a total sample size of 92 (23 per group) or 16 (4 per group) to detect a medium or large effect, respectively. Importantly, these simulations investigated four means in between-group comparisons, whereas we employ a mixed design (including between- and within-subject comparisons). Since within-subjects comparisons yield more statistical power compared to between-subjects comparisons, we require a lower sample size than suggested by these simulations. As a consequence, a maximum sample size of $N = 60$ (30 per group) might be sufficient. However, due to risks of overestimation of effect sizes based on earlier studies⁸², sample variability, and other unpredictable factors, we will test a larger maximum sample of $N = 80$ (40 per group). This is larger than the sample size of most studies in the field^{1,25}, indicating that the study would contribute to the field, even in the case of weak evidence.

The minimum sample size of $n = 20$ per group is often used as an example for the minimum sample size in several methodological articles^{63,79} and aligns with the recommendation that multilevel models should comprise a sample of at least 20 clusters on the highest level (in our case 20 individuals per group) to provide unbiased estimates of the variance of fixed effects⁸³. Consequently, the minimum sample size is sufficient to yield unbiased and reliable results and would allow strong conclusions in case of convincing evidence (note that in the absence of convincing evidence, the sample size will be increased).

Most importantly though, even in the case of weak evidence, Bayesian analyses will likely not result in any inconclusive results, as the statistics will always provide evidence in the direction of one hypothesis over the other hypotheses under consideration (except if all Posterior Model Probabilities would be exactly the same). Therefore, we can interpret the strength and direction of the evidence, even if the evidence at the maximum sample size is not as strong as desired initially^{63,79}.

Reviewer 1 – comment 2:

It is also important to provide justification for the minimum N at which statistical testing will start and the potential to end the study will start (Schönbrodt & Wagenmakers, 2017).

We agree that it would be best practice to justify the minimum n and thank the reviewer for pointing this out. To the best of our knowledge, there are currently no established criteria to determine a minimum N . Nevertheless, several considerations allow to decide on a minimum N . The most important requirement is that the sample must be sufficiently large such that the results are trustworthy, reliable, and not driven by one or two outliers. As we pointed out in our previous answer, simulation studies have shown that 23 or 4 participants per group are sufficient to reach a power of 0.80 to detect evidence for a medium or large effect, respectively, when investigating a specific hypothesis regarding four group means (e.g., $m_1 > m_2 > m_3 > m_4$). Since we investigate a combination of between-group and within-group comparisons, we assume our power to be larger than in these simulations and therefore a smaller sample would suffice. In case of a large anticipated effect, very little participants would then be sufficient to detect this effect (a considerably smaller sample than our minimum sample size).

Initially, we intended to collect a minimum sample of $N = 30$ participants, which would be more than enough to find evidence for a strong and possibly even for a medium effect (based on the considerations above). However, a study with $N = 30$ would still be relatively small and might therefore provide a less significant contribution to the field compared to larger studies. Therefore, we have decided to increase the minimum sample size to $N = 40$ (20 per group). In line with this adjustment, a group sample size of $n = 20$ is often used as an example for the minimum sample size in several methodological articles (even though this recommendation is still somewhat arbitrary; Hoijtink, Mulder, van Lissa, & Gu, 2019; Schönbrodt & Wagenmakers, 2018). Moreover, a minimum sample size of 20 per group aligns with the recommendation that multilevel models should comprise a sample of at least 20 clusters on the highest level (in our case 20 individuals per group) to provide unbiased estimates of the variance of fixed effects (Stegmueller, 2013).

We have added a brief justification of the minimum sample size in the methods (see also response to comment 1):

Methods, p. 18:

"The minimum sample size of $n = 20$ per group is often used as an example for the minimum sample size in several methodological articles^{63,79} and aligns with the recommendation that multilevel models should comprise a sample of at least 20 clusters on the highest level (in our case 20 individuals per group) to provide unbiased estimates of the variance of fixed effects⁸³. Consequently, the minimum sample size is sufficient to yield unbiased and reliable results and would allow strong conclusions in case of convincing evidence (note that in the absence of convincing evidence, the sample size will be increased)."

Reviewer 1 – comment 3:

Also, the authors state that they "will collect data until there is strong evidence for one hypothesis within Test 1 and Test 2A" - what does this mean about all the other tests they have planned? Does having sufficient data not matter for these questions or are they for some reason assumed to have higher power?

We are grateful for the opportunity to clarify our reasoning. The primary objective of our study is to quantify evidence for competing predictions of overgeneral memory and negativity bias theories regarding dysphoric distortions in retrieved detail and expressed affect, described in Test 1 and Test 2A, respectively. The secondary objectives of our study are to determine whether any observed alterations for dysphoria are indeed specific to memories (Test 2B) and to test the critical assumption of these theories (and emerging memory therapeutics) that episodic detail correlates with affect-increases and investigate whether this relationship is altered in dysphoria (Test 3). These latter tests are informative and will hopefully help to provide a more nuanced insight into episodic and affective memory distortions, but these questions did not drive the design of our study. In line with recommendations for clinical trials, we have based our sample size calculation on our primary objectives (Van Meter & Charnigo, 2013). This was not described sufficiently clear in the manuscript.

At the same time, we agree that is important to have sufficient data to also draw convincing conclusions for the secondary objectives. For the analyses of the movie clips (Test 2B), it can be observed in the pilot data that the evidence for the emotional responses to movie clips is stronger than responses to memories (a pattern that we consistently find in our research). Considering that fEMG responses to movies are stronger than fEMG responses to memories, we expect that if we find strong evidence for affective memory distortions (Test 2A), we have likely even more statistical power to find evidence for affective distortions when viewing movie clips (Test 2B). For the analysis of the relationship between episodic detail and affective responses (Test 3), we reasoned that since the estimation of these relationships involves within-subject relationships, it has more power than instances in which between-subject relationships are investigated. In addition, our pilot data with $N=8$ already indicated evidence for such a relationship in positive memories. This suggests that the registered report with a larger sample size should have sufficient power to draw strong conclusions regarding the relationship between episodic detail and affective responses. Taken together, as soon as we stop data collection when $PostP_i$ is equal to or larger than .80 for each of the primary research questions (Test 1 and Test 2A), it is likely that the strength of the evidence associated with the secondary questions (Test 2B and test 3) is equally strong or even stronger.

Finally, the use of Bayesian statistics will always provide evidence in the direction of one hypothesis over the other hypotheses (see also response above about power considerations; Schönbrodt & Wagenmakers, 2018). Therefore, we will be able to draw carefully weighted conclusions from all of our analyses, even when the evidence might not be as strong as for Test 1 and Test 2A.

We have now explicitly mentioned the distinction between primary and secondary research questions in the introduction. Moreover, we have stated explicitly that the stopping criterion is evaluated based on the primary research questions in the methods section and added an argument why we also have sufficient power to draw conclusions about the secondary research questions (see also response to comment 1).

Introduction, p. 7:

"Test 1 and Test 2A will therefore address our primary research objective to evaluate evidence for competing predictions of overgeneral memory and negativity bias theories regarding episodic memory detail and affective responses to memories. The secondary objectives of our study are to determine whether any observed affective distortions are indeed specific to memories (Test 2B) and to test the critical theoretical assumption that episodic detail correlates with affect-increases (Test 3)."

Methods, p. 19:

"Most importantly though, even in the case of weak evidence, Bayesian analyses will likely not result in any inconclusive results, as the statistics will always provide evidence in the direction of one hypothesis over the other hypotheses under consideration (except if all Posterior Model Probabilities would be exactly the same). Therefore, we can interpret the strength and direction of the evidence, even if the evidence associated with the primary or secondary questions at the maximum sample size is not as strong as desired initially^{63,79}. In case the Posterior Model Probabilities of Test 1 and Test 2A indicate that data collection can be stopped before the maximum sample size is reached, it is likely that the strength of the evidence associated with the secondary questions (Test 2B and Test 3) is equally strong or even stronger than evidence associated with the primary questions, since the pilot data indicated that fEMG responses to the movies are generally very strong (essential for Test 2B), and estimating within-subject relationships (Test 3) generally entails more power than between-subjects relationships."

Reviewer 1 – comment 4:

Participants will be informed that they should retrieve positive and negative memories that are at least a week old but not older than five years. The neutral memories, however, only need to be older than one day while still not older than five years. The authors state that this is because neutral memories are typically forgotten quickly. However, this is problematic for allowing for clear conclusions when comparing emotional and neutral memories, as they may differ in their level of detail due to the age of the memories rather than their emotional nature. In particular, the difference in instructions between the types of memories may suggest to participants that more recent memories are expected for neutral memories.

This is an important point and we have modified our methods accordingly. We agree that it is better to keep the instructions for the selection of different types of memories consistent. Therefore, we have changed the minimum age for all memories to one day, in line with other autobiographical memory studies that requested memories to be at least one day old (van Schie, Chiu, Rombouts, Heiser, & Elzinga, 2019; Werner-Seidler, Tan, & Dalgleish, 2017; Young, Bellgowan, Bodurka, & Drevets, 2013; Young, Siegle, Bodurka, & Drevets, 2016). This strategy keeps the instructions consistent and thereby avoids the possible situation where participants may erroneously assume that they are expected to provide memories of varying age, while additionally ensuring that the memories are consolidated over at least one night of sleep (Poe, 2017; Sara, 2017).

We have adapted the memory selection sheet and the information regarding memory selection in the Methods section:

Methods, p. 15:

"The experimenter will instruct participants to select specific memories that they have experienced personally, that are emotionally meaningful to them, that happened at a particular place and time, and that did not last longer than a day^{1,28,32}. The memories need to be at least one day old^{11,26,27}, but not older than five years⁷²."

Given the notion that neutral memories are more readily forgotten than emotional ones (Yonelinas & Ritchey, 2015), it may still be possible that the provided neutral memories are on average younger than the emotional memories. However, this is unlikely to compromise our main analyses and conclusions. The neutral memories are solely used as a control condition for the manipulation check that addresses whether the zygomaticus and the corrugator respond specifically to positive and negative memories, respectively. Data of the neutral memories are not included in any of the confirmatory analyses (i.e., all hypotheses refer to positive and/or negative memories, but not to neutral memories). Hence, if our results reveal that the neutral memories are on average younger than the emotional memories, this would not explain why emotional memories elicit stronger emotional responses than neutral memories, because younger memories are typically associated with higher emotionality and vividness (Cooper, Kensinger, & Ritchey, 2019; Walker, Skowronski, & Thompson, 2003). Taken together, any potential differences between neutral and emotional memories are unlikely to readily alter the conclusions of the study.

We have added a paragraph to the Methods section in which we specify that we will provide descriptive statistics for important variables in our study that may be taken into consideration in the interpretation of our results.

Methods, p. 23:

"We will provide descriptive statistics on self-reported depressive symptoms (BDI scores), anhedonia (assessed with the SHAPS), participant age, and sex for the dysphoric and the non-dysphoric group to characterize our samples. Moreover, we will provide a table with descriptive statistics per condition (positive, negative, neutral) and group (dysphoric, non-dysphoric) for important memory variables collected with the ROAM. Specifically, we will report self-reported valence and arousal of the memories, the number of retrieved episodic and semantic details, memory age, and memory vividness. For all of these variables, we will use BAIT to test whether there are differences between the dysphoric and non-dysphoric group that need to be taken into consideration when interpreting our results. Furthermore, we will provide descriptive statistics of the fEMG responses to the movie clips as well as self-reported movie valence and arousal."

Reviewer 1 – comment 5:

It is not clear who will conduct the transcriptions and coding of the audio recordings and whether they will be blind to condition or whether they will be the same researchers who ran the sessions and assessed the BDI-II with the participants at the end.

We agree that the experimenter, the memory transcriber and coder should be blind to the group (i.e., dysphoric or non-dysphoric) and this was not clearly communicated in the manuscript. The initial online screening involves the assessment of eligible participants in terms of BDI scores and some other variables such as gender, age, and exclusion criteria (e.g., history of depression or excessive drug use). This online screening will be conducted by a person who is otherwise not involved in running the experiment, transcribing and coding the memories (person A; e.g., a research assistant or the last author of the study). Person A will provide a list of contact details (name, email address, phone number) to person B, who will contact the eligible participants and schedule the appointments for participation. Person B will also be the experimenter. During the experiment itself, pen and paper questionnaires (including the BDI) will be put in an envelope by the participant. Following completion of the experiment, there are two possible scenarios for each participant. If a participant wants to learn about their BDI score or indicates suicidal intentions during the debriefing, the experimenter will open the envelope and inform the participant about their BDI score. In that case, Person B is blind during the experiment, but is not able to transcribe or code the memories. In this scenario, Person C transcribes and codes the memories. If a participant doesn't want to learn about their BDI score and doesn't indicate suicidal intentions during the debriefing, the envelopes will not be opened until the memories are transcribed and coded. In that case, person B can transcribe and code the memories of the participant at hand. Therefore, the transcription and coding of the memories may be conducted by person B or person C. We expect that Person B will usually be the first author and Person C the second author.

We have now explained the blinding procedure in the Methods of the revised manuscript and we have added the schematic overview below to the supplement (Supplement 4).

Methods, p. 20-21:

*"Since the online screening will be conducted by a person who is otherwise not involved in running the experiment, transcribing and coding of the memories, the experimenter, transcriber, and coder will be blind to the participants' condition. If a participant wants to learn about their BDI score or indicates suicidal intentions during the debriefing after the experiment, a different person than the experimenter will transcribe and code the memories to ensure that the transcriber and coder are blind to the participant's condition (see **Supplement 4** for a schematic overview of the blinding procedure)."*

Supplement 4, p. 45:

Mara Mather
(signed review)

We are very grateful for your supportive and constructive feedback. We hope that by responding to and incorporating your feedback, we have significantly improved our study protocol. We appreciate the signed and transparent peer review and are looking forward to being able to sign our communication as well once the peer review process has been finished.

Reviewer #2 (Remarks to the Author):

Thank you for the opportunity to review this interesting study. Simultaneous consideration of overgeneral and negative memory biases will have both theoretical and clinical implications, and the authors have articulated clear hypotheses made by each of these theories. Key strengths of the study are the sampling plan and analysis approach, which will be easily reproducible, and good consideration of clinical risk issues. The combination of physiological and behavioural measures is critically important in this type of research. However, when examining cognitive processes it is important to ensure that there are established links between the behavioural and physiological measures you intend to use to measure the same construct.

We thank the reviewer for pointing out key strengths of our manuscript such as the sampling plan and analysis approach, as well as for pointing out important ways to improve our study. We hope to successfully address the reviewer's comments in our responses below.

Reviewer 2 – comment 1:

A key concern I have is that there does not appear to be a pre-established association between the strength of facial electromyography (fEMG) activity and the intensity of affect? That is, the pilot data seem to suggest that mentally well individuals frown more/have higher corrugator activity when recalling negative memories relative to neutral memories, and smile more/have higher zygomaticus activity when recalling positive memories relative to neutral. However, your key hypotheses centre on the intensity of affect. If this (apparently previously unestablished) relationship between your physiological measure and construct of interest is not evident, you will be unable to assess your key research questions.

We are grateful for this important comment and agree that we did not sufficiently elaborate on evidence for a pre-established link between the strength of fEMG responses and affect intensity, while such a link is indeed critical to answer our research questions. The assumed association is inferred from previous work showing convincing evidence that stronger facial expressions measured with fEMG reflect stronger affect intensity (Golland, Hakim, Aloni, Schaefer, & Levit-Binnun, 2018; Heller, Lapate, Mayer, & Davidson, 2014; Lang, Greenwald, Bradley, & Hamm, 1993; Larsen, Norris, & Cacioppo, 2003). Specifically, the strength of fEMG responses covaries with subjective self-reports of the experienced intensity of positive and negative affect in response to emotional pictures and sounds (Lang et al., 1993; Larsen et al., 2003), as well as in response to emotional imagery (Brown & Schwartz, 1980). Other studies revealed that when the intensity of emotional stimuli is independently manipulated, fEMG strength is modulated accordingly (Brown & Schwartz, 1980; Golland et al., 2018). Specifically relevant for our analytical approach is that EMG – affect intensity relationships have been shown both across individuals and within individuals (Golland et al., 2018; Heller et al., 2014; Lang et al., 1993; Larsen et al., 2003). The relationship across individuals is important for Test 2, when we compare the affective responses between dysphoric and non-dysphoric individuals. The relationship within individuals is important for Test 3A, when we investigate the relationship between affective responses and episodic detail. Our pilot results also provide preliminary evidence that fEMG responses to positive memories covary with episodic detail within individuals, which shows that fEMG as applied in our memory paradigm is responsive to subtle fEMG variation within subjects. In the light of the discussed studies, these early results suggest that memories richer in episodic detail are accompanied by more intense positive affective expression.

In sum, convincing evidence suggests that the strength of zygomaticus and corrugator responses measured with fEMG reflect affect intensity, and variation in intensity can be picked up both within and across individuals. We have now added some of the above-mentioned observations to better justify the assumed link of fEMG and affect intensity. We have also added a brief introduction and explanation of fEMG as a psychophysiological measure of affect intensity.

Introduction, p. 6:

"Specifically, we will employ facial electromyography (fEMG) of the zygomaticus major and the corrugator supercilii regions to quantify affective responses during memory retrieval. The zygomaticus contracts to produce smiling and indicates positive affect, whereas the corrugator contracts to produce frowning and indicates the negative affect^{49,50}. Facial EMG responses are thought to reflect relatively automatic affective responses^{39,53-55} and may therefore be less influenced by higher-order cognitive functions and knowledge about one's self, unlike self-reports. Moreover, the strength of fEMG responses reflect variations in affect intensity, as suggested by studies that experimentally manipulated affect intensity^{54,56,57} or examined naturally occurring variations in experienced affect^{49,50,53}. Importantly, such variations in affect intensity can be

measured both within and between participants^{49,50,53,54}. The observation that fEMG can pick up even small variations in affect intensity within an individual indicates that it is possible to investigate possible relationships of expressed affect and episodic detail of individual memories.”

Reviewer 2 – comment 2:

To put this in context, there is no clear rationale/prior evidence presented for why there would be a linear relationship between the strength of fEMG activity and the phenomenological intensity of negative affect experienced by the individual. Surely it is possible to experience intense negative affect without a more intense/deeper frown (at least this would seem to be the case when working clinically with patients experiencing strong negative affect, who often appear quite blank when reporting intensively negative memories). Further justification of the construct validity of your psychophysiological measure of affect intensity is needed.

In our response to comment 1, we have provided evidence from prior studies that – at least in healthy individuals – a linear link between experience of affect intensity and strength of fEMG responses can be assumed. However, in order to test the overgeneral memory and negativity bias hypotheses in our study, it is indeed necessary that a similar link between fEMG responses and affect intensity can be observed in dysphoric individuals. In previous studies, depressive symptoms were associated with reduced zygomaticus responses to positive information, but normal or even enhanced corrugator responses to negative information (Franzen & Brinkmann, 2016; Franzen et al., 2019; Lindsey, Rohan, Roeklein, & Mahon, 2011; Schwartz, Fair, Salt, et al., 1976; Teasdale & Fogarty, 1979). The notion that only zygomaticus but not corrugator responses were diminished suggests that altered fEMG responses do not reflect generally blunted facial expressions and that increases in negative affect are expressed through corrugator responses in clinical samples, even if these responses might not be visible to the eye (Schwartz, Fair, Salt, et al., 1976). Another important observation in previous studies with dysphoric individuals was that normal self-reported negative affect was accompanied by normal corrugator activity (Franzen & Brinkmann, 2016; Franzen et al., 2019), while increases in self-reported depressed mood in clinical samples were accompanied by increases in corrugator activity (Lindsey et al., 2011). Thus, previous research suggests that increases in negative affect will be expressed through corresponding increases in corrugator activity.

Based on this evidence, we expect that the linear link between experience of affect intensity and strength of fEMG responses in healthy samples can also be assumed in dysphoric samples. We agree with the reviewer that the link between fEMG responses and affect intensity in healthy and in dysphoric individuals is important to mention as it justifies the construct validity of our measure. We have therefore adjusted the introduction accordingly.

Furthermore, in light of this and other comments (see also answers to comments 3, 11, 13), we have decided to investigate self-reported subjective feelings as a complementary outcome variable in Test 2A, 2B, and 3. Should dysphoric participants – unexpectedly and in contrast to previous studies – show blunted facial expressions despite strong subjective feelings, the combination of fEMG and self-report measures will allow to detect this and we can account for it in the discussion. Furthermore, combining fEMG and self-report data will allow to gain more comprehensive and nuanced insights into affective memory distortions.

Introduction, p. 6:

“Furthermore, fEMG can be used to investigate distorted affective processes in dysphoria and depression^{4,23,58,59}. Specifically, dysphoric and depressed individuals showed reduced zygomaticus activity compared to healthy controls when processing positive information, but normal or even enhanced corrugator activity when processing negative information^{4,58-60}. We will use fEMG responses as the primary and self-reported subjective feelings as a complementary outcome measure for affective responses. Correspondingly, the conclusions from this study will mainly be guided by the results of the fEMG analyses, but the insights from the self-reports will help to provide a more comprehensive and nuanced insight into affective memory distortions^{26,27}.”

Reviewer 2 – comment 3:

Only completing a manipulation check (page 20) for the non-dysphoric group is somewhat concerning, in light of the above, and as the method does not seem to have been used in clinical samples before (the studies referenced were done with healthy samples – and one study only had female participants). Consideration of whether factors such as psychomotor retardation (a symptom of depression) would impact fEMG measurements seems important.

We aim to investigate alterations in affective responses in dysphoric compared to non-dysphoric individuals. We conduct the manipulation checks in the non-dysphoric group to test whether the ROAM task elicits fEMG responses in line with what would be expected in a healthy sample. Since we expect that the dysphoric group experiences systematic distortions in these responses, the dysphoric group probably cannot be used for a manipulation check of the success of the ROAM paradigm itself. For example, it is possible that dysphoric participants experience such a strong overgeneral memory or negativity bias that they don't respond to positive memories at all. However, we agree that it is crucial to also gain an in-depth insight into the response pattern of the dysphoric group. Therefore, we will conduct the same analyses for the dysphoric group as for the non-dysphoric group, but not as a manipulation check of the experimental paradigm.

We briefly explained this approach in the introduction (p. 6), Design table 1 (p. 41), and in the Methods (p. 20). For the revision, we now clarified that the manipulation check is intended to assess the success of the ROAM:

Methods, p. 20:

"We will test whether the retrieval of positive and negative autobiographical memories elicits affective responses that can be measured with fEMG. Since dysphoric individuals are expected to experience affective memory distortions, the effectiveness of our manipulation as implemented by the ROAM will only be evaluated within the non-dysphoric group. However, we will conduct the same analyses within the dysphoric group to draw a comprehensive picture of how dysphoric and non-dysphoric individuals re-experience autobiographical memories."

In our response to comment 2, we hope to have successfully addressed the concern of the reviewer that the fEMG measure has not been applied yet in clinical samples. To briefly recap, several studies have investigated positive and negative affect in dysphoric and depressed samples (Franzen & Brinkmann, 2016; Franzen et al., 2019; Lindsey et al., 2011; Schwartz, Fair, Mandel, et al., 1976; Schwartz, Fair, Salt, et al., 1976) and we now mention them in the revised manuscript. We expect that the linear link that exists between experience of affect intensity and strength of fEMG responses in healthy samples can also be assumed to exist in dysphoric samples.

Regarding psychomotor retardation and other alternative factors that may play a role in affective responses and fEMG, it is important to note that our analyses will allow to quantify evidence for these ideas compared to an overgeneral memory or negativity bias. For example, in Test 2B, we will assess whether dysphoric individuals show the same pattern of affective responses to new events (movie clips) as for autobiographical memories. If psychomotor retardation is at play, it would lead to blunted zygomaticus and blunted corrugator responses to positive and negative memories, respectively (Test 2A), similar to an overgeneral memory bias prediction. However, in contrast to an overgeneral memory bias that should only affect memories, psychomotor retardation should additionally lead to blunted zygomaticus and corrugator responses to positive and negative movie clips, respectively (Test 2B). It is further noteworthy that, even though psychomotor retardation may play a role in depression, fEMG results of previous studies with dysphoric and depressed samples cannot readily be explained by psychomotor retardation because similar or even enhanced corrugator responses to negative information were observed compared to healthy controls (Franzen & Brinkmann, 2016; Franzen et al., 2019; Lindsey et al., 2011; Schwartz, Fair, Salt, et al., 1976).

Furthermore, we include a 'none of the above' hypothesis in all of our analyses to account for the possibility that the data do not fit with any of our hypotheses (see also general feedback of reviewer 4). The 'none of the above' hypothesis allows to account for any pattern in the data that we did not specify a priori. If we find the strongest evidence for this 'none of the above' hypothesis, exploratory analyses will help to draw conclusions about alternative explanations and to generate hypotheses for future confirmatory studies.

In sum, if psychomotor retardation or other factors are at play, we are likely to detect it, and it is unlikely that we erroneously attribute patterns in the data to one of the main theories that we aim to critically assess.

Reviewer 2 – comment 4:

The experimental design also seems unable to establish the direction of the relationship – i.e., that episodic detail causes increased intensity of affect measured via fEMG, which is what the Introduction seems to suggest is a vitally needed step in evaluating current theories (e.g., 'Consequently, it is not clear whether affective distortions co-occur with, or are even driven by, episodic distortions.'). Indeed, you hypothesise that episodic detail positively predicts the affective response on a trial-by-trial basis. However, your design sees that you are only able to analyse

cross-sectional associations. Prior experiments have manipulated the amount of episodic detail with which memories are recalled to try and get at this issue.

We agree that the analysis designed to test the relationship between episodic detail and affect does not warrant causal claims about the idea that episodic detail would drive the fEMG responses. We have therefore removed any suggestions of causality based on these analyses from the manuscript. However, establishing a relationship between episodic detail and affective responses remains a critical step towards understanding affective memory distortions, not least because it is a necessary precondition for causality. Notably, in contrast to earlier studies that were only able to investigate episodic detail at the participant level (i.e., differences in episodic detail between individuals), our multilevel approach allows to investigate episodic detail and its associations with affective responses at the memory level (i.e., differences in episodic detail within individuals). This approach will allow to provide evidence in favour of or against the notion that - within a person - memories that are retrieved in more detail also elicit stronger affective responses.

Introduction, p. 3-4:

"However, previous research on autobiographical memory distortions focused on isolated aspects of memory retrieval^{4,33,34}, precluding more overarching insights into dysphoric memory distortions and their interrelationships. Consequently, it is not clear whether affective distortions co-occur with episodic distortions."

More minor comments:

- Methods -

Reviewer 2 – comment 5:

Exclusion/inclusion criteria require further clarification. You have stated that you will screen out those with a history of psychiatric disorder (page 17) but a BDI of over 16 would indicate the likely presence of disorder, even if there has been no formal diagnosis.

We agree that a BDI score of over 16 can indicate the presence of a depressive disorder. We plan to exclude individuals who have ever received a formal diagnosis and potentially treatment in the past, not because we want to exclude participants with current clinical levels of depression, but because it is likely that their affective and cognitive processes are already affected by prior depression or treatment. By excluding participants with past diagnosis, we therefore aim to obtain a relatively homogenous sample of participants that experience prototypic cognitive and affective distortions that are not yet affected by professional interventions.

We have added a justification for our sample to the Methods of our manuscript.

Methods, p. 18:

"We employ the BDI cut-off score in combination with the exclusion of participants who ever received treatment or a formal diagnosis because when individuals have received a diagnosis, and potentially treatment, it is likely that their affective and cognitive processes are affected by a prior depressive episode or by treatment. By excluding participants with a past diagnosis, we therefore aim to achieve a relatively homogenous sample of dysphoric participants that experience prototypic cognitive and affective distortions that are not yet affected by prior depressive episodes or professional interventions."

Reviewer 2 – comment 6:

What is the reasoning for not reporting the obtained skin conductance levels and heart rate (ECG) during the movie and the ROAM tasks? These measures would provide valuable data and may help address the intensity of affect issue I have raised.

We have addressed the relationship of affect intensity and fEMG in response to comment 1, 2, and 3, and this will hopefully justify fEMG as the best primary outcome variable to answer our research questions. Notably, facial EMG reflects relatively automatic evaluations of affective valence and intensity (see also comment 11; Dimberg et al., 2002; Larsen et al., 2003; Scherer, 2009). Unlike fEMG, skin conductance levels (SCL) and heart rate (HR) cannot distinguish among positive and negative affective states, as these are mediated by the autonomic nervous system (ANS) and these measures are therefore most susceptible to the arousing properties of affective stimuli (Lang et al., 1993). We are primarily interested in memories that may be relevant in the development of depression such as sad or happy memories. These memories may be characterized by intense

negative or positive affect despite low arousal levels (i.e., low-arousing sad or happy memories). Memories that are characterized by high negativity and low arousal might not lead to pronounced heart rate or skin conductance responses, but nonetheless to pronounced fEMG responses. Based on these considerations, fEMG is the best outcome variable for the main research questions in the proposed study.

Nonetheless, we agree with the reviewer that other measures such as SCL and HR may still provide valuable insights into the reliving of autobiographical memories in dysphoric and non-dysphoric individuals. For this reason, we will collect these variables alongside other additional variables (e.g., memory perspective and retrieval frequency), for exploratory purposes. We reason that since these indices are not necessary to answer our research questions, thoroughly analysing all of these variables is beyond the scope of a single preregistered manuscript. Rather than superficially reporting descriptive statistics on these measures, we will collect these data for future studies (given that they don't intervene with our study design). More importantly, we will make all data publicly available apart from those for which legal or ethical restrictions apply (for example highly sensitive data such as audio recordings and transcriptions of the personal autobiographical memories). Therefore, other researchers have the opportunity to reanalyse our data and to conduct new analyses with all the variables that are of interest to them.

Reviewer 2 – comment 7:

Will data from memories that are not specific be excluded?

We are not planning to exclude data of memories that are not specific. Our instructions during the first session are very clear and we make sure that the participants understand the concept of a specific memory (for example that it must occur within one day). From the memories in our pilot study, it becomes apparent that without time restrictions and with almost entirely free choice of memories in the first session, participants very rarely report memories that do not contain at least one specific event. As a consequence, we don't expect many memories that are not specific at all. Another reason to include all memories is that even if a memory is not specific in the sense that it does not contain information about a single event, this data would still be informative. Our coding would capture such a lack of episodic detail (by resulting in no or very few internal episodic details). In line with an overgeneral memory bias, such memories should elicit blunted affective responses as predicted.

Reviewer 2 – comment 8:

I suggest you increase the number of memories that are second rated for episodic detail to a minimum of 25% as this is a key outcome of interest.

We agree with the reviewer's suggestion and have increased the number of memories that are second rated to a minimum of 25%.

Methods, p. 20:

"We will randomly select 25% of the memories such that they will be coded by at least two independent raters to assess interrater reliability."

Reviewer 2 – comment 9:

Including a measure of anhedonia would also be informative, and may help to explore whether fEMG activity is associated with the phenomenological experience of affect.

We agree that a measure of anhedonia would be interesting for exploratory analyses or future studies based on our data. We have therefore added the Snaith-Hamilton Pleasure Scale (SHAPS) to the questionnaires that the participants complete in Session 1. The SHAPS is a commonly used and reliable questionnaire that can be used in healthy and clinical samples (Cogan, Shapses, Robinson, & Tronson, 2019; Snaith et al., 1995; Trøstheim et al., 2020).

We have added the questionnaire to Figure 1B and to the Methods Section.

Methods, p. 17:

"We also collect information on personality characteristics including depressive feelings using Beck's Depression Inventory modified⁶⁵ (BDI-II), anxiety using the trait subscale of the State-Trait Anxiety Inventory⁶⁹ (STAI), the ability to experience pleasure using the Snaith-Hamilton Pleasure Scale⁷⁰ (SHAPS, English version), as well as the use of mental imagery using the Spontaneous Use of Imagery Scale⁶⁷ (SUIS) and the Plymouth Sensory Imagery Scale⁶⁸ (PSI-Q, English version)."

Reviewer 2 – comment 10:

Why is fEMG activity being baseline corrected for memory recall but not movie clips?

We thank the reviewer for noticing this inconsistency and now keep the analysis pipeline consistent within the study. Specifically, we have adjusted the fEMG analysis in accordance with the recommendations of reviewer 3: We have re-analysed all pilot data (movies and memories) using percentage changes from baseline to quantify EMG responses, rather than amplitudes and difference scores. Importantly, changing the analysis pipeline did not change the pattern of the pilot results. For the confirmatory analyses of the registered report, we will also adhere to this recommendation of reviewer 3 (for more details see response to reviewer 3 – comment 1).

Reviewer 2 – comment 11:

- Introduction -

More needs to be done to emphasise why we can't rely on self-report of affect. From a clinical perspective, it is the patient's felt experience of the emotion which is important, as it is this felt experience which drives their decision-making and behaviour.

We have selected fEMG as the primary outcome variable for our analyses for several reasons (see also responses to comments 1, 2, and 3). First, psychophysiological responses are usually less prone to expectancy and demand effects than self-reports. Second and more importantly, psychophysiological recordings don't require explicit categorization and labelling of affect, which can alter the affective response under investigation. In order to self-report an affective state, a participant has to become aware of their affective state and has to reflect on it. In contrast, EMG responses can be measured continuously without interrupting the memory retrieval. Furthermore, facial expressions are assumed to be tightly linked to action readiness and motivated behaviour (Adams, Ambady, Macrae, & Kleck, 2006; Frijda & Tcherkassof, 1997). Finally, even though fEMG responses may be linked to emotional appraisal as well as to subjective feelings, empirical and theoretical frameworks suggest that fEMG responses reflect relatively automatic affective responses (Brown & Schwartz, 1980; Golland et al., 2018; Scherer, 2009, see for example Figure 1 and 2). For this reason, fEMG may be less influenced by higher-order cognitive functions and knowledge about one's self. For example, a participant who experiences depressive symptoms might generally 'know' that they are not a joyous person. As a consequence, they might hesitate to report that a positive memory elicits strong positive feelings (even if that might be the case) because it is in conflict with their current self-view or because they feel that they are not worthy to feel the positive emotions. EMG responses, on the other hand, may allow to measure positive affective states in response to positive memories even if such responses are in conflict with the current self-concept.

Nonetheless, in light of this and other comments (see also answers to comments 2, 3, and 13), we have decided to investigate self-reported subjective feelings as a complementary outcome variable. Importantly, we do not regard fEMG responses as a measure that is generally better than self-report. In our opinion, both measures may shed light on different affective and cognitive processes and provide complementary rather than competing information. While self-reported subjective feelings will often be better suited for applied research that aims at intervening in psychopathology, psychophysiological measures may be better suited to provide insights into some of the basic affective and cognitive processes that may play a role in the aetiology of a disorder. In any case, conducting the same analyses with fEMG responses and with self-reported feelings will provide a more comprehensive insight into affective memory distortions in dysphoria (see also comment 13). Furthermore, a potential disparity between self-reported feelings and the psychophysiological expression of affect during memory retrieval may generate interesting hypotheses for future research. Finally, investigating self-reported feelings in addition to fEMG responses will allow more direct comparisons with previous studies in the field that mainly relied on self-reports.

We have added the following considerations to the introduction of the manuscript.

Introduction, p. 5:

"Previous studies of emotional autobiographical memory retrieval have relied heavily on self-report as the sole measure of affective responses^{9,24,35,36}. Even though self-reports of subjective feelings provide an important source of information, they are prone to experimental biases such as expectancy and demand effects⁴⁹⁻⁵². More importantly, self-reports require awareness, explicit categorization, and labelling of affect, which can alter the affective response under investigation³⁹."

Introduction, p. 6:

"Facial EMG responses are thought to reflect relatively automatic affective responses^{39,53-55} and may therefore be less influenced by higher-order cognitive functions and knowledge about one's self, unlike self-reports."

Reviewer 2 – comment 12:

No clear distinction is made between the prior experience of depression and dysphoria – individuals remitted from depression may be dysphoric, but individuals with no prior history of depression may also be dysphoric. When it comes to cognitive factors that drive disorder, this is an important distinction. If there is a prior history of depression, cognitive distortions may have emerged during that prior episode, indicating that the cognitive factor may not be a primary predictor of symptoms, but rather a scar from a prior experience of depression, and thereby only a secondary risk factor (i.e., for relapse of depression).

We agree that it is important to distinguish between individuals with a history of depression and individuals without such a history, as this affects the ability to draw conclusions with regard to precipitating versus acquired cognitive distortions. For this reason, we aim to only test participants who experience depressive symptoms (BDI score ≥ 16), but who have never received a diagnosis of depression and who have never sought professional help for their depressive symptoms. With this approach, we hope to achieve a relatively homogenous sample of individuals that experience prototypic affective and cognitive distortions. Even though it could still be the case that a few dysphoric participants who have never sought help or received a formal diagnosis nevertheless have a history of depression, it is likely that our approach results in dysphoric participants of whom the majority has no such history. Therefore, it will be unlikely that prior depressive episodes or treatment can explain the affective memory distortions under investigation (see also comment 5).

Methods, p. 18:

"We employ the BDI cut-off score in combination with the exclusion of participants who ever received treatment or a formal diagnosis because when individuals have received a diagnosis, and potentially treatment, it is likely that their affective and cognitive processes are affected by a prior depressive episode or by treatment. By excluding participants with a past diagnosis, we therefore aim to achieve a relatively homogenous sample of dysphoric participants that experience prototypic cognitive and affective distortions that are not yet affected by prior depressive episodes or professional interventions."

Reviewer 2 – comment 13:

It would be useful to highlight that a combination of self-report and neurobiological approaches have been used to explore how the experience of affect differs between normative, and depressed or high-risk samples, e.g., <https://jamanetwork.com/journals/jamapsychiatry/article-abstract/1688033>
<https://ajp.psychiatryonline.org/doi/full/10.1176/appi.ajp.2015.15010119>

We agree that at least one of these studies should be mentioned, especially because the study from Young and colleagues in 2016 is one of the methodologically most sound studies in the field of memory distortions and depression. Moreover, we agree that combining psychophysiological approaches with self-reports can provide valuable insights. Therefore, we will investigate self-reported valence ratings as a complementary outcome measure (see also comments 2, 3, and 11). We have added the following references and explanations:

Introduction, p. 3:

"Additionally, they tend to recollect sombre memories and have difficulties remembering specific events in rich episodic detail^{1,25-27}."

References²⁶ and ²⁷ indicate Young et al (2013) and Young et al (2016), respectively.

Introduction, p. 6:

"We will use fEMG responses as the primary and self-reported subjective feelings as a complementary outcome measure for affective responses. Correspondingly, the conclusions from this study will mainly be guided by the results of the fEMG analyses, but the insights from the self-reports will help to provide a more comprehensive and nuanced insight into affective memory distortions^{26,27}."

Reviewer #3 (Remarks to the Author):

The current pilot study addresses the effect of autobiographical memory content with positive or negative affective color or emotional movie clips on EMG responses of two facial muscles: corrugator supercilii, presumed to indicate negative affective responses, and zygomaticus major, presumed to indicate positive affective responses. Participants were asked to retrieve positive, negative, or neutral experiences from their autobiographic memory and to label these memories using key words. During an experimental session, an 8-s lasting fixation cross was followed by a 10 s lasting presentation of cue words, representing a particular memory experience, and a 50-s period during which participants silently retrieved autobiographical memories in response to these cues. During another experimental condition, participants were presented an 8-s lasting fixation cross followed by the presentation of a positive, negative, or neutral movie clip during which participants had to imagine themselves as being part of the scene. These clips were of variable duration.

EMG measurements during autobiographical memory tasks were either baseline-corrected or not corrected. In case of baseline correction, a baseline EMG value determined during a 4-s period preceding the 10-s presentation of cue words was subtracted from the values obtained during the subsequent 60-s autobiographical memory task. If no baseline correction was applied, raw EMG amplitude responses were quantified from the beginning of the 60-s task period. EMG responses to movies were not baseline-corrected.

I have several problems with this study, in particular with the methodology of the EMG measurements:

Reviewer 3 – comment 1:

(1) EMG responses of both muscles were averaged across different stimuli within subjects and subsequently across subjects. Such averaging is only meaningful if responses are adequately standardized so that they can be compared between different stimuli presented to a subject, between different experimental conditions to which a subject is exposed, or between different subjects. Such standardization also allows a comparison between different muscles. The problem of unstandardized EMG responses is that such comparisons cannot be made due to a multitude of variables influencing absolute EMG amplitude in terms of microvolts. For example, EMG amplitude may considerably differ between individuals, is on average larger in males than in females, is larger in younger than in older persons, and may differ between different recording sessions performed on the same individuals. Absolute EMG amplitude also depends on the exact location of the recording electrodes. Removing electrodes from the face and placing them again on the same location may result in a different amplitude. Therefore, EMG responses should be standardized. The authors performed such a standardization for autobiographical memory tasks in pilot 3 (but not for tasks in pilots 1 or 2, neither for movies). As stated above, they expressed EMG values during the 60-s memory task as difference scores with the mean value during a 4-s period preceding the task. Within the behavioral sciences, calculating such difference scores in terms of microvolts between experimental and baseline conditions is common practice for EMG responses. Unfortunately, such interval scale scores cannot be considered an adequate standardization of EMG responses since these are still expressed in terms of microvolts, and are thus dependent on anatomical and physiological differences between muscles. Because during a state of complete muscle relaxation EMG activity has a zero amplitude it should be expressed on a ratio scale with a zero origin. Using a ratio scale implies that EMG responses should be expressed as a percentage of baseline activity rather than as a difference score. This method is common practice within medical or physiological disciplines (Merletti, R., 1999, Standards for reporting EMG data. International Society of Electrophysiology and Kinesiology. J. Electromyogr. Kinesiol., 9, III-IV, 1999). EMG responses of a specific muscle in terms of percentage scores can be averaged across stimuli, experimental conditions, or subjects. This method should also be used in the current study.

We are very grateful that the reviewer pointed out the importance of standardization of EMG responses when investigating facial expressions across stimuli, conditions, and participants. We had indeed proposed to quantify EMG responses as difference scores because this approach is common in the literature. However, we share the reviewer's concerns regarding this approach. Our analyses as presented in the unrevised manuscript did handle some of the problems that are mentioned (such as differences between individuals) by estimating averages in multilevel models that include a random intercept per participant. Nonetheless, we agree that EMG responses might better be presented as percentage changes from baseline. Therefore, we have re-analysed all of our pilot data in line with your recommendation and adjusted our pilot results in the revised manuscript. We have also implemented percentage changes from baseline for the analyses of the

movie clips. Importantly, this different analysis approach yielded very similar results that support the same conclusions. In line with the reviewer's suggestion, we will also conduct the confirmatory analyses of the registered report with percentage changes from baseline, rather than difference scores or mean amplitudes.

Following the suggestions of the reviewer, we have updated the statistics in the entire manuscript to use percentage changes instead of difference scores and mean amplitudes.

- Methods, pilot data, p. 11-13
- Supplement 1, p. 37-39
- Supplement 2, p. 40-41
- Design tables, p. 46-67

Methods, p. 22:

"For each memory trial in session 2, we will calculate the percentage change from baseline (4 seconds) during the 60 seconds retrieval phase⁸⁸. Subsequently, we will calculate the average of the percentage change during the 60 seconds retrieval phase per trial. Likewise, for each movie trial in session 1, we will calculate the percentage change from baseline (4 seconds) over the entire time-course of the movie. Subsequently, we will calculate the average of the percentage change per movie trial."

Reviewer 3 – comment 2:

(2) A basic problem of recording facial EMG activity is crosstalk from other muscles. To reduce such interfering effects, electrodes with small contact areas should be used which are placed at close distance from each other. Information about these details is not provided in the current manuscript.

We indeed did not provide sufficient information regarding the acquisition of the fEMG data. In general, when applying electrodes, we follow the guidelines from Fridlund and Cacioppo (1986). We have added this reference as well as information on the fEMG data acquisition to the manuscript. Our electrodes have a small contact area (6 mm) and are purchased from MedCaT: <https://medcat.ccvshop.nl/Gesinterte-AgCl-Elektrodensensor-6mm>

Methods, p. 21:

"For the retrieval part of the ROAM and the movie task in session 1, facial EMG is collected with two pairs of sintered Ag/AgCl EMG electrodes with six mm sensors that will be placed in the zygomaticus major and the corrugator supercilii region of the left side of the face, according to established guidelines⁸⁴. A reference electrode will be placed below the hairline in the horizontal centre of the forehead. Before electrode placement, the participants' skin will be cleaned with a face rub gel and alcohol wipes. The electrodes will be connected to a custom-made bipolar EMG amplifier with an input resistance of 1G Ω , an amplification factor of 5200, and a bandwidth of 5-1000Hz (6dB/oct)."

Reference ⁸⁴ indicates Fridlund and Cacioppo (1986).

Reviewer 3 – comment 3:

(3) The authors applied automated artifact rejection implying that data segments deviating 3 or more standard deviations from the mean of all segments were rejected and were considered as missing values. Such a statistical procedure is not uncommon within the behavioral sciences. However, it is not an accurate procedure. Facial EMG responses may be very dynamic. For example, an emotional event or experience may be accompanied by a sudden short-lasting increase in EMG activity of the involved muscle. Removing such responses merely based on statistical criteria might bias the results. I therefore recommend that artifact removal should be based on visual inspection of raw EMG data by an expert having experience with such data.

We agree that automated artefact rejection based on statistical criteria can sometimes eliminate a meaningful signal, but we believe that visual inspection may suffer from similar limitations. After all, criteria for visual inspection and automated artefact rejection should theoretically target the same artefacts, only automated artefact rejection aims to express these mathematically, which makes both approaches in theory very similar. Visual inspection by an expert could provide better results but only when combined with video recordings of the participants' faces during the task, which would allow to identify artefacts due to, for example, coughs or other movements such as scratching one's face. However, we do not record the participants' faces for two reasons. First, in the presence of others, emotional facial expressions serve as a communicative tool (Crivelli &

Fridlund, 2018; Hess, Banse, & Kappas, 1995). As a consequence, fEMG responses are less likely to be automatic or to reflect internal affective states as they normally do outside of a social context (Gehricke & Shapiro, 2000). A video camera that is directed towards a participant's face can be interpreted as a social context by some or all of the participants, which may influence our measure of interest as well as the cognitive and affective processes under investigation. This effect of social context can even vary systematically between dysphoric and non-dysphoric individuals (Gehricke & Shapiro, 2000). Second, being video-taped in combination with audio recordings of personal memories represents an excessive burden for participants and their privacy and might influence their commitment to participate and engage in the study.

A major additional advantage of automated artefact rejection over visual inspection is that such an approach is particularly suited for preregistered research because it employs an objective criterion that can be specified before data collection. For similar reasons, analysis pipelines with automated artefact rejection are 100% reproducible by other researchers.

Taken these arguments into consideration, we hesitate to employ visual inspection for artefact rejection and instead prefer to employ automated artefact rejection. However, we are open to optimize our artefact rejection based on potential suggestions by the reviewers or to include additional criteria that can be expressed in an algorithm.

Methods, p. 21:

"Automated artefact rejection is particularly suited for preregistered research because it employs objective criteria that can be specified before data collection and because it is 100% reproducible by other researchers."

Reviewer #4 (Remarks to the Author):

This proposal seeks to provide a direct test of whether the theory of overgeneral memory bias or the theory of negativity bias can better account for various discrepant behaviors between dysphoric and non-dysphoric individuals. This is not my specific research area, but I was asked to evaluate the proposed analysis as my expertise is in Bayesian statistics.

Informed Bayesian hypothesis tests are to be performed here, wherein (often multiple) parameter constraints are specified on the statistical model to represent the predictions from a given theory. Then evidence from the data is used to evaluate how much each set of predictions is supported to produce Bayes factors and posterior model probabilities. This would all be done using the `bain` r package, which is an excellent piece of software.

The analysis is well described, and I found the tables to be especially informative. I was glad to see the Bayesian framework leveraged to its fullest extent with such informative hypotheses implemented, including both sets of directional contrasts and point hypotheses where appropriate. I am also glad to see a catch-all "none of the above hypothesis" to account for potential theoretical misspecification (i.e., one theory could be supported over another, while both are poor accounts of the data).

Thank you very much for the positive evaluation of our analysis plan.

Reviewer 4 – comment 1:

Overall, the proposed analysis looks quite sound. The only potential (minor) qualm I would raise is with the sampling plan. The proposed stopping rule is to collect data until Bayes factors of 10 or more are found. But, with these multiple compound-directional hypotheses it is entirely possible to find medium-to-strong Bayes factors without obtaining correspondingly high posterior model probabilities (because the prior probability of each hypothesis decreases the more hypotheses are being tested). For example, in the pilot analysis manipulation checks, there are cases where one model is obtaining a high Bayes factor in its favor (e.g. $BF > 20$) but only attains a posterior probability of $\sim .65$. So, the *evidence* can be strong while there is simultaneously much uncertainty remaining about which hypothesis is true. It has become standard to use the Bayes factor as a stopping criterion when testing a simple focal hypothesis, but perhaps achieving high posterior model probability would be a more useful one here? There is no right answer and it is ultimately up to the authors to decide, but I thought I might bring the issue to attention.

Thank you for this comment. In general, it is hard to define a good stopping criterion because such criteria are always to some extent arbitrary (there is no consent on what can be considered strong evidence within a field). We decided to use a Bayes Factor (BF) of 10 as stopping criterion because this is recommended by Nature Communications and because a BF of 10 or larger has been suggested to represent strong evidence (Schönbrodt & Wagenmakers, 2018). However, we agree that it is better to use the Posterior Model Probability (PostP) as a stopping criterion for our study because a PostP quantifies the support in the data for one hypothesis relative to all other hypotheses under investigation (as opposed to a BF that quantifies evidence for one hypothesis relative to one other hypothesis). Therefore, PostP better fits our conceptual approach that investigates evidence for multiple sets of competing predictions. We have changed the stopping criterion to $PostP_i \geq .80$.

We have adjusted the stopping criterion in the methods section and in the design tables.

Methods, p. 18-19:

"We will employ a modified Sequential Bayes Factor design⁷⁸, that is, we will collect data until we find convincing evidence for one hypothesis relative to all other hypotheses under investigation (i.e., the largest Posterior Model Probability $PostP_i$ is equal to or larger than .80) for each of the primary research questions (Test 1 and Test 2A), or until we reach a maximum sample size of $N = 80$. We will commence by including a minimum of 20 participants per group ($N=40$) and compute Posterior Model Probabilities for Test 1 and Test 2A. If there is convincing evidence for a specific hypothesis over the other hypotheses within both Test 1 and Test 2A ($PostP_i \geq .80$), we will stop data collection at the minimum sample size. Otherwise, we will increase the sample size in incremental steps of $n = 10$ (5 per group) and repeat the testing procedure until $PostP_i \geq .80$ or until the maximum sample size $N = 80$ is reached. At $N = 80$, the results will be reported regardless of the strength of evidence for each hypothesis."

Editor comment – relevant for reviewer 4:

We decided to follow the editor's recommendation to conduct prior sensitivity analyses in the registered report. This decision might be relevant for reviewer 4 to evaluate the quality of our revised manuscript. Specifically, in the first draft of the manuscript, we have employed the most conservative priors (using the smallest possible fraction = 1 of the data to specify the variance of the prior distribution). We have decided to change this approach. For the confirmatory analyses, we will by default use a moderate fraction = 2 of the data to specify the prior distribution. In sensitivity analyses, we will then adjust this fraction to 1 (more conservative) and 3 (more liberal) to investigate how the prior distributions affect our results.

We have adjusted the methods section and the design tables accordingly. Furthermore, we have added sensitivity analyses for the analyses of the pilot data in Supplement 3.

Methods, p. 22:

"For all analyses, we will employ the default settings implemented in the bain package⁶¹ with the exception that we will use a moderate fraction = 2 of the data to define the prior variance. Additionally, we will conduct sensitivity analyses with a more conservative fraction = 1 and a more liberal fraction = 3 to evaluate the influence of the prior variance on our results."

References

- Adams, R. B., Ambady, N., Macrae, C. N., & Kleck, R. E. (2006). Emotional expressions forecast approach-avoidance behavior. *Motivation and Emotion, 30*(2), 179–188. <https://doi.org/10.1007/s11031-006-9020-2>
- Brown, S. L., & Schwartz, G. E. (1980). Relationships between facial electromyography and subjective experience during affective imagery. *Biological Psychology, 11*, 49–62. [https://doi.org/10.1016/0301-0511\(80\)90026-5](https://doi.org/10.1016/0301-0511(80)90026-5)
- Cogan, E. S., Shapses, M. A., Robinson, T. E., & Tronson, N. C. (2019). Disrupting reconsolidation: memory erasure or blunting of emotional/motivational value? *Neuropsychopharmacology, 44*(2), 399–407. <https://doi.org/10.1038/s41386-018-0082-0>
- Cooper, R. A., Kensinger, E. A., & Ritchey, M. (2019). Memories Fade: The Relationship Between Memory Vividness and Remembered Visual Salience. *Psychological Science, 30*(5), 657–668. <https://doi.org/10.1177/0956797619836093>
- Crivelli, C., & Fridlund, A. J. (2018). Facial Displays Are Tools for Social Influence. *Trends in Cognitive Sciences, 22*(5), 388–399. <https://doi.org/10.1016/j.tics.2018.02.006>
- Dimberg, U., Thunberg, M., & Grunedal, S. (2002). Facial reactions to emotional stimuli: Automatically controlled emotional responses. *Cognition and Emotion, 16*(4), 449–471. <https://doi.org/10.1080/02699930143000356>
- Franzen, J., & Brinkmann, K. (2016). Wanting and liking in dysphoria: Cardiovascular and facial EMG responses during incentive processing. *Biological Psychology, 121*, 19–29. <https://doi.org/10.1016/j.biopsycho.2016.07.018>
- Franzen, J., Brinkmann, K., Gendolla, G. H. E., & Sentissi, O. (2019). Major depression impairs incentive processing: Evidence from the heart and the face. *Psychological Medicine, 49*(6), 922–930. <https://doi.org/10.1017/S0033291718001526>
- Fridlund, A. J., & Cacioppo, J. T. (1986). Guidelines for Human Electromyographic Research. *Psychophysiology, 19*(1), 11–20. <https://doi.org/10.1111/j.1469-8986.1986.tb00676.x>
- Frijda, N. H., & Tcherkassof, A. (1997). Facial expressions as modes of action readiness. In J. A. Russell & J. M. Fernández-Dols (Eds.), *The Psychology of Facial Expression* (pp. 78–102). Paris: Cambridge University Press & Editions de la Maison des Sciences de l'Homme. <https://doi.org/10.1017/cbo9780511659911.006>
- Gehricke, J.-G., & Shapiro, D. (2000). Reduced facial expression and social context in major depression: discrepancies between facial muscle activity and self-reported emotion. *Psychiatry Research, 95*, 157–167.
- Gelman, A., & Carlin, J. (2014). Beyond Power Calculations: Assessing Type S (Sign) and Type M (Magnitude) Errors. *Perspectives on Psychological Science, 9*(6), 641–651. <https://doi.org/10.1177/1745691614551642>
- Golland, Y., Hakim, A., Aloni, T., Schaefer, S., & Levit-Binnun, N. (2018). Affect dynamics of facial EMG during continuous emotional experiences. *Biological Psychology, 139*(January), 47–58. <https://doi.org/10.1016/j.biopsycho.2018.10.003>
- Heller, A. S., Lapate, R. C., Mayer, K. E., & Davidson, R. J. (2014). The Face of Negative Affect: Trial-by-Trial Corrugator Responses to Negative Pictures Are Positively Associated with Amygdala and Negatively Associated with Ventromedial Prefrontal Cortex Activity. *Journal of Cognitive Neuroscience, 26*(9), 2102–2110. https://doi.org/10.1162/jocn_a_00622
- Hess, U., Banse, R., & Kappas, A. (1995). The Intensity of Facial Expression Is Determined by Underlying Affective State and Social Situation. *Journal of Personality and Social Psychology, 69*(2), 280–288. <https://doi.org/10.1037/0022-3514.69.2.280>
- Hojtink, H., Mulder, J., van Lissa, C., & Gu, X. (2019). A Tutorial on Testing Hypotheses Using the Bayes Factor. *Psychological Methods, 24*(5), 539–556. <https://doi.org/10.1037/met0000201>
- Klugkist, I., Post, L., Haarhuis, F., & Van Wesel, F. (2014). Confirmatory Methods, or Huge Samples, Are Required to Obtain Power for the Evaluation of Theories. *Open Journal of Statistics, 4*, 710–725. <https://doi.org/10.4236/ojs.2014.49066>
- Kuiper, R. M., & Hoijtink, H. (2010). Comparisons of Means Using Exploratory and Confirmatory Approaches. *Psychological Methods, 15*(1), 69–86. <https://doi.org/10.1037/a0018720>
- Lakens, D. (2021). Sample Size Justification. *PsyArXiv Preprints*. <https://doi.org/10.31234/osf.io/9d3yf>
- Lang, P. J., Greenwald, M. K., Bradley, M. M., & Hamm, A. O. (1993). Looking at pictures: Affective, facial, visceral, and behavioral reactions. *Psychophysiology, 30*, 261–273. <https://doi.org/10.1111/j.1469-8986.1993.tb03352.x>
- Larsen, J. T., Norris, C. J., & Cacioppo, J. T. (2003). Effects of positive and negative affect on electromyographic activity over zygomaticus major and corrugator supercilii. *Psychophysiology, 40*(5), 776–785. <https://doi.org/10.1111/1469-8986.00078>
- Lin, X. X., Sun, Y. Bin, Wang, Y. Z., Fan, L., Wang, X., Wang, N., ... Wang, J. Y. (2019). Ambiguity

- Processing Bias Induced by Depressed Mood Is Associated with Diminished Pleasantness. *Scientific Reports*, 9(1), 1–12. <https://doi.org/10.1038/s41598-019-55277-6>
- Lindsey, K. T., Rohan, K. J., Roecklein, K. A., & Mahon, J. N. (2011). Surface facial electromyography, skin conductance, and self-reported emotional responses to light- and season-relevant stimuli in seasonal affective disorder. *Journal of Affective Disorders*, 133(1–2), 311–319. <https://doi.org/10.1016/j.jad.2011.04.016>
- Liu, X., Li, L., Xiao, J., Yang, J., & Jiang, X. (2013). Abnormalities of autobiographical memory of patients with depressive disorders: A meta-analysis. *Psychology and Psychotherapy: Theory, Research and Practice*, 86(4), 353–373. <https://doi.org/10.1111/j.2044-8341.2012.02077.x>
- Poe, G. R. (2017). Sleep is for forgetting. *Journal of Neuroscience*, 37(3), 464–473. <https://doi.org/10.1523/JNEUROSCI.0820-16.2017>
- Ray, R. D., McRae, K., Ochsner, K. N., & Gross, J. J. (2010). Cognitive reappraisal of negative affect: Converging evidence from EMG and self-report. *Emotion*, 10(4), 587–592. <https://doi.org/10.1037/a0019015>
- Sara, S. J. (2017). Sleep to remember. *Journal of Neuroscience*, 37(3), 457–463. <https://doi.org/10.1523/JNEUROSCI.0297-16.2017>
- Scherer, K. R. (2009). The dynamic architecture of emotion: Evidence for the component process model. *Cognition & Emotion*, 23(7), 1307–1351. <https://doi.org/10.1080/02699930902928969>
- Schönbrodt, F. D., & Wagenmakers, E. J. (2018). Bayes factor design analysis: Planning for compelling evidence. *Psychonomic Bulletin and Review*, 25(1), 128–142. <https://doi.org/10.3758/s13423-017-1230-y>
- Schwartz, G. E., Fair, P. L., Mandel, M. R., Salt, P., & Klerman, G. L. (1976). Facial Expression and Imagery in Depression: An Electromyographic Study. *Psychosomatic Medicine*, 38(5), 337–347. <https://doi.org/10.1097/00006842-197609000-00006>
- Schwartz, G. E., Fair, P. L., Salt, P., Mandel, M. R., & Klerman, G. L. (1976). Facial Muscle Patterning to Affective Imagery in Depressed and Nondepressed Subjects. *Science*, 192(4238), 489–491.
- Snaith, R. P., Hamilton, M., Morley, S., Humayan, A., Hargreaves, D., & Trigwell, P. (1995). A scale for the assessment of hedonic tone. The Snaith-Hamilton Pleasure Scale. *British Journal of Psychiatry*, 167, 99–103. <https://doi.org/10.1192/bjp.167.1.99>
- Söderlund, H., Moscovitch, M., Kumar, N., Daskalakis, Z. J., Flint, A., Herrmann, N., & Levine, B. (2014). Autobiographical episodic memory in major depressive disorder. *Journal of Abnormal Psychology*, 123(1), 51–60. <https://doi.org/10.1037/a0035610>
- Stegmuller, D. (2013). How Many Countries for Multilevel Modeling? A Comparison of Frequentist and Bayesian Approaches. *American Journal of Political Science*, 57(3), 748–761. <https://doi.org/10.1111/ajps>
- Teasdale, J. D., & Fogarty, S. J. (1979). Differential effects of induced mood on retrieval of pleasant and unpleasant events from episodic memory. *Journal of Abnormal Psychology*, 88(3), 248–257. <https://doi.org/10.1037/0021-843X.88.3.248>
- Trøstheim, M., Eikemo, M., Meir, R., Hansen, I., Paul, E., Kroll, S. L., ... Leknes, S. (2020). Assessment of Anhedonia in Adults With and Without Mental Illness: A Systematic Review and Meta-analysis. *JAMA Network Open*, 3(8), e2013233. <https://doi.org/10.1001/jamanetworkopen.2020.13233>
- Van Meter, E., & Charnigo, R. (2013). Strengthening Interactions between Statisticians and Collaborators: Objectives and Sample Sizes. *Journal of Biometrics & Biostatistics*, 5(1), 1–4. <https://doi.org/10.4172/2155-6180.1000e127>
- van Schie, C. C., Chiu, C. De, Rombouts, S. A. R. B., Heiser, W. J., & Elzinga, B. M. (2019). When I relive a positive me: Vivid autobiographical memories facilitate auto-noetic brain activation and enhance mood. *Human Brain Mapping*, 40(16), 4859–4871. <https://doi.org/10.1002/hbm.24742>
- Walker, W. R., Skowronski, J. J., & Thompson, C. P. (2003). Life Is Pleasant-and Memory Helps to Keep It That Way! <https://doi.org/10.1037/1089-2680.7.2.203>
- Werner-Seidler, A., Tan, L., & Dalgleish, T. (2017). The Vicissitudes of Positive Autobiographical Recollection as an Emotion Regulation Strategy in Depression. *Clinical Psychological Science*, 5(1), 26–36. <https://doi.org/10.1177/2167702616647922>
- Williams, J. M. G., Barnhofer, T., Crane, C., Herman, D., Raes, F., Watkins, E., & Dalgleish, T. (2007). Autobiographical memory specificity and emotional disorder. *Psychological Bulletin*, 133(1), 122–148. <https://doi.org/10.1037/0033-2909.133.1.122>
- Yonelinas, A. P., & Ritchey, M. (2015). The slow forgetting of emotional episodic memories: An emotional binding account. *Trends in Cognitive Sciences*, 19(5), 259–267. <https://doi.org/10.1016/j.tics.2015.02.009>
- Young, K. D., Bellgowan, P. S. F., Bodurka, J., & Drevets, W. C. (2013). Behavioral and

Neurophysiological Correlates of Autobiographical Memory Deficits in Patients With Depression and Individuals at High Risk for Depression. *JAMA Psychiatry*, 70(7), 698–708. <https://doi.org/10.1001/jamapsychiatry.2013.1189>

Young, K. D., Siegle, G. J., Bodurka, J., & Drevets, W. C. (2016). Amygdala activity during autobiographical memory recall in depressed and vulnerable individuals: Association with symptom severity and autobiographical overgenerality. *American Journal of Psychiatry*, 173(1), 78–89. <https://doi.org/10.1176/appi.ajp.2015.15010119>

Reviewers' Comments:

Reviewer #1:

Remarks to the Author:

The authors have thoughtfully addressed the concerns raised by myself and the other reviewers and strengthened the design of their proposed study. I do not have further suggestions for improvement.

Mara Mather

Reviewer #2:

Remarks to the Author:

The authors have completed a thoughtful and considered response to my review. My key concerns have been addressed via additional information in the manuscript, and I believe that the addition of self-reported affect will allow for a more thorough consideration of the mechanisms at play. I look forward to seeing the results.

If it is the effects of treatment or chronic depressive episodes that the authors are trying to avoid when recruiting participants without a prior diagnosis of depression, I would suggest that the exclusion criteria are reframed. Specifically, you may benefit from allowing self-report of prior depressive episodes, self-help for depression, or professional help (e.g., counselling, antidepressant medication), not just diagnosis, as diagnoses may not be commonly given, despite symptoms being experienced or depressive intervention being received.

All the best with your study,
Caitlin Hitchcock

Reviewer #3:

Remarks to the Author:

I think that the authors have adequately dealt with my comments and suggestions. Therefore, I don't see any reasons to make any further comments or suggestions.

Anton van Boxtel

Reviewer #4:

Remarks to the Author:

I have reviewed the comments from the other reviewers, and the rebuttal, and find that all of my (R4) minor concerns have been sufficiently addressed.

Review Study Plan (Stage 2)

Reviewers' Comments:

Reviewer #2:

Remarks to the Author:

Thank you for the opportunity to review this Stage 2 Registered Report. Adaptations to the method are acknowledged and analysis is in line with the registration. This study provides an important and significant contribution to the field.

The Introduction has simplified the theories of memory distortion, but in doing so, has lost some key aspects. For example, overgeneral memory bias refers to reduced retrieval of specific single incident events, and the temporal specificity of an event is different to the level of detail contained within that event representation. Indeed, though related, internal, episodic detail is a distinct construct, measured with different tasks, see (<https://wires.onlinelibrary.wiley.com/doi/full/10.1002/wcs.1624>)

This should be considered in the Introduction, and also limits the strength of conclusions regarding the overgeneral memory bias discussed in the Discussion.

I very much like the approach of trying to separate the two biases, but there is also the possibility that the two biases may interact. This has been emphasised elsewhere, eg recent review of affective bias in autobiographical memory and mental health recently published in another Nature journal, <https://www.nature.com/articles/s44159-023-00148-1>

Negativity bias suggests that individuals with dysphoria will have enhanced negative emotions upon retrieval of specific negative memories, but in relation to what – their positive memories, or non-dysphoric individuals? This is unclear in your description of negativity bias theories, as it reads like you are hypothesising both positive detail and positive affect attenuation relative to non-dysphoric individuals, and enhanced negative detail and negative affect relative to non-dysphoric individuals. That is, it seems you are suggesting both a positive attenuation and a negative bias/enhancement. Given that your focus is on distinguishing between components of different theories, it is important that you are specific in your descriptions of the theory guiding your hypotheses.

When reading on, I see that you do analyse positive attenuation affects. You are much clearer about the contrast between positive attenuation and negative bias here, and this should be reflected earlier in the manuscript.

The attenuation of positive affect in dysphoric individuals appears larger for memories relative to movies (Figure 3 b and c). This has implications for existence of memory bias, in addition to a broader affective impairment discussed in the paper. That is, positive affect seems particularly impaired for memories.

There were a few features of the obtained data that made me slightly cautious about the reliability of the facial electromyography method, for example, there were large standard deviations for the zygomaticus response to positive stimuli (including SDs much higher than the mean). I say this in light of the fact that the pilot data with the ROAM demonstrated an opposite pattern of results for episodic detail (your primary outcome), than what you report here. I think this difference from the pilot warrants further consideration, beyond a sentence in the Methods.

Regarding imagery, the manual that you used to score internal details includes markers such as emotion details, sensory detail, and specificity of descriptions and adjectives, which are commonly used to index the strength of imagery. The argument in the Discussion that imagery may have a stronger effect, relative to the episodic details you have analysed and show a lack of effect for, is therefore not well supported.

Reviewer #3:

Remarks to the Author:

My primary - and single - task as a reviewer was to evaluate the method of facial EMG recording and quantification within this study.

My conclusion is short and clear: both recording and quantification were performed excellently.

I have only a minor remark concerning the reference list: the last name of the author of study #108 should be "van Boxtel" rather than "Boxtel".

Reviewer #4:

Remarks to the Author:

It is great to see this completed registered report. The evidence is strong, showing large Bayes factors and fairly decisive posterior probabilities overall. The Bayesian analyses are clearly described, and in line with everything initially proposed. The exploratory analyses are also thorough.

I think this will be a very well-regarded paper.

Reviewer #2 (Remarks to the Author):

Reviewer comment 1

Thank you for the opportunity to review this Stage 2 Registered Report. Adaptations to the method are acknowledged and analysis is in line with the registration. This study provides an important and significant contribution to the field.

We thank the reviewer for the positive assessment of our registered report and address the remaining questions point by point below.

The Introduction has simplified the theories of memory distortion, but in doing so, has lost some key aspects. For example, overgeneral memory bias refers to reduced retrieval of specific single incident events, and the temporal specificity of an event is different to the level of detail contained within that event representation. Indeed, though related, internal, episodic detail is a distinct construct, measured with different tasks, see (<https://wires.onlinelibrary.wiley.com/doi/full/10.1002/wcs.1624>) This should be considered in the Introduction, and also limits the strength of conclusions regarding the overgeneral memory bias discussed in the Discussion.

The suggested definition of overgeneral memory bias by the reviewer remains close to the operationalization and findings using the Autobiographical Memory Test (AMT). Specifically, in the AMT and in early theoretical perspectives¹, overgeneral memory bias refers to the reduced tendency to access specific single incident events that occurred in a specific place at a specific time (i.e., reduced memory specificity). However, more recent theoretical and clinical perspectives have moved beyond this definition and regard overgeneral memory as representative of generally biased episodic recollection, linking it to concepts such as imagery and memory vividness, episodic detail, and sometimes emotionality²⁻⁶. Based on these developments, our definition of overgeneral memory is also broader than the initial operationalization of memory specificity with the AMT and refers to a general reduction in episodic recollection, i.e., the ability to vividly relive (including concomitant emotions) specific past events.

We chose this broader definition because there may be quantitative differences in episodic recollection of autobiographical memories, even when a specific memory is accessed, and these differences may be clinically relevant. For example, one person might be able to remember when and where they married but without much further detail and concurrent emotional responses. Another person might remember their wedding day in vivid detail, including all the guests, the weather, the music, and they may easily re-enact the concomitant emotions. The AMT is not suited to capture such potentially important differences as it merely indicates an individual's general tendency to access specific or categorical memories. That is, in the example, both memories would be categorized as specific, and important information regarding any differences among memories as well as among individuals would be lost. Therefore, we employed Levine's coding scheme^{5,7} that allows to assess episodic detail as a quantitative indicator of episodic recollection on the memory level as well as on the person level. Since especially in the context of depression, a lot of work also focusses on the consequences of memory biases for emotions^{2,4,8-10}, we also added psychophysiological indices of expressed emotions as indicators of an overgeneral memory or a negativity bias.

In sum, our employed definition of overgeneral memory bias in terms of episodic recollection represents current theoretical and clinical perspectives well. As we are aware that our operationalization diverges somewhat from the field's tendency to rely on findings from the AMT, we elaborated on the interpretational considerations that come with using our approach. In the

discussion, we therefore address the difference between both approaches and highlight what can be learned from combining insights from both paradigms. This is in line with a recent call, published after we completed the stage 1 protocol, to investigate memory distortions using different operationalizations than specificity with the AMT (the paper highlighted by the reviewer)¹¹.

Revised discussion paragraph:

p. 26: Across all analyses, we found converging evidence that a negativity bias is a more plausible explanation for memory distortions in dysphoria than an overgeneral memory bias. This finding ostensibly conflicts with previous research showing that participants with dysphoria retrieve fewer specific memories than participants without dysphoria, regardless of memory valence^{12,13}. However, such reduced memory specificity was traditionally assessed with the autobiographical memory test (AMT), which probes the ease of memory access during memory search. That is, memory specificity in the AMT assesses how likely a person is to retrieve specific or categorical (overgeneral) memories. In the ROAM employed in this study, episodic detail represents the recollection of specific memories after successful completion of the search process, because participants provided personal memory cues for specific events prior to reliving each memory. Therefore, our study complements findings from the AMT regarding memory access with insights on subsequent elaboration and reliving, which may represent distinct processes that rely on different cognitive and biological systems^{6,11,14,15}. This approach is in line with a recent call to not only investigate memory distortions in terms of specificity, but also other related yet dissociable memory features such as memory detail and emotionality¹¹. Notably, different cognitive and affective biases may interact in the aetiology and maintenance of depression^{16,17}. Regarding memory distortions in dysphoria, research with the AMT shows that people with dysphoria do not readily retrieve specific memories, be it positive or negative. However, our data underscore that people with dysphoria can retrieve specific memories, and when they do, their negative memories are enhanced in detail and possibly affect, while their positive memories are reduced in detail and affect, which is in line with a negativity bias.

Reviewer comment 2

I very much like the approach of trying to separate the two biases, but there is also the possibility that the two biases may interact. This has been emphasised elsewhere, eg recent review of affective bias in autobiographical memory and mental health recently published in another Nature journal, <https://www.nature.com/articles/s44159-023-00148-1>

We agree that multiple biases may interact and also elaborate on this in the discussion. We have revised the relevant paragraph to make this even clearer and added a reference to the article of Dagleish and Hitchcock (2023). The adjustments are presented in the response to reviewer comment 1.

Reviewer comment 3

Negativity bias suggests that individuals with dysphoria will have enhanced negative emotions upon retrieval of specific negative memories, but in relation to what – their positive memories, or non-dysphoric individuals? This is unclear in your description of negativity bias theories, as it reads like you are hypothesising both positive detail and positive affect attenuation relative to non-dysphoric individuals, and enhanced negative detail and negative affect relative to non-dysphoric individuals. That is, it seems you are suggesting both a positive attenuation and a negative bias/enhancement. Given that your focus is on distinguishing between components of different theories, it is important that you are specific in your descriptions of the theory guiding your hypotheses.

When reading on, I see that you do analyse positive attenuation affects. You are much clearer about

the contrast between positive attenuation and negative bias here, and this should be reflected earlier in the manuscript.

We explicitly mention in the introduction, the methods section, and the design tables what we mean with negativity bias (including the relevant statistical comparisons): enhanced negative memories and diminished positive memories, compared to individuals without dysphoria (the same prediction applies to episodic detail and affect). Since, in our opinion, the definition of negativity bias is clear, and changes to the introduction are not permitted for stage 2 registered reports, we did not adjust the introduction.

We only explicitly considered positive attenuation (diminished positive memories but normal negative memories compared to people without dysphoria) as a post hoc explanation to explain the pattern of affective expressions that we observed in the data. To keep the distinction between a priori predictions and post hoc explanations clear, we did not adjust the introduction (also in line with the registered report guidelines).

As a sidenote, at the time, when writing the stage 1 report, we were wondering whether to include positive attenuation in our proposed analyses. However, given that meta-analytic evidence showed no evidence for positive attenuation in dysphoria¹⁸ and the majority of theories emphasize distortions in both positive and negative memories^{1,2,19,20}, we initially did not deem positive attenuation (i.e., without concurrent alteration of negative memories) a plausible account for memory distortions.

Excerpt from the introduction, denoting the predictions:

p. 7: Overgeneral memory bias theories predict that dysphoric individuals retrieve fewer episodic detail (Test 1 – H1) and concurrently experience diminished affect (Test 2A – H1) when reliving positive memories as well as when reliving negative memories, compared to non-dysphoric participants. Negativity bias theories predict that dysphoric individuals retrieve fewer episodic detail when reliving positive memories, but more episodic detail when reliving negative memories (Test 1 – H2). Concurrently, they experience diminished affect when reliving positive memories, but enhanced affect when reliving negative memories (Test 2A – H2), compared to non-dysphoric participants.

Excerpts from the methods:

p. 44: We compared evidence for four competing hypotheses about the amount of episodic detail that is retrieved when reliving autobiographical memories. Test 1 – H1: Individuals with dysphoria retrieve fewer episodic detail when reliving positive memories and when reliving negative memories, compared individuals without dysphoria. Test 1 – H2: Individuals with dysphoria retrieve fewer episodic detail when reliving positive memories but more episodic detail when reliving negative memories, compared to individuals without dysphoria [...].

p. 45, line: We compared four competing hypotheses about affective responses to autobiographical memories. Test 2A – H1: Individuals with dysphoria experience diminished positive affect when remembering positive memories and diminished negative affect when remembering negative memories, compared to individuals without dysphoria [...].

Reviewer comment 4

The attenuation of positive affect in dysphoric individuals appears larger for memories relative to movies (Figure 3 b and c). This has implications for existence of memory bias, in addition to a broader affective impairment discussed in the paper. That is, positive affect seems particularly impaired for memories.

Interestingly, a visual inspection of the graphs indeed seems to suggest that the magnitude of attenuation in positive affect is stronger for memories than for movies. This might suggest that attenuation of positive affect is particularly strong for memories. However, we are cautious with such an interpretation because the movies and memories used in this study represent stimuli that can differ in important aspects, including self-relevance and intrinsic emotionality, as the memories originate from other experiences than the movies. We do not know how emotional the autobiographical memories were at the time of encoding. Therefore, we cannot draw conclusions based on a comparison of the magnitude of distortions when viewing movie clips and when remembering autobiographical memories. Our data only allow a comparison of the pattern of distortions in response to novel emotional experiences (i.e., movie clips) and to memories, but not of the magnitude of such distortions.

Nevertheless, we share the reviewer's enthusiasm for the potential of future insights in looking at the data in this way, and added these considerations as a suggestion for future research to the discussion.

Excerpt from revised discussion:

*p. 29: It might be possible that the magnitude of the attenuation of positive affect in dysphoria may be stronger for memories than for new experiences (a notion that seems to be consistent with a visual inspection of **Figure 3b and c**). Initially small affective distortions during encoding could also exacerbate over time, resulting in even stronger affective distortions during memory retrieval^{21,22}. However, the memories and movies used in this study might differ in self-relevance and intrinsic emotionality, because they do not refer to the same encoding experience. Therefore, our data do not allow for conclusions regarding differences in the magnitude of affective distortions during encoding compared to remembering episodes (as opposed to different patterns of distortions during encoding and remembering against which we found evidence). Future research could measure affective responses during encoding and subsequent retrieval of the same episodes to investigate how affective biases evolve over time, for example using movie clips and memories thereof in the lab²³.*

Reviewer comment 5

There were a few features of the obtained data that made me slightly cautious about the reliability of the facial electromyography method, for example, there were large standard deviations for the zygomaticus response to positive stimuli (including SDs much higher than the mean). I say this in light of the fact that the pilot data with the ROAM demonstrated an opposite pattern of results for episodic detail (your primary outcome), than what you report here. I think this difference from the pilot warrants further consideration, beyond a sentence in the Methods.

The pilot data was not intended to provide a serious test of any of the hypotheses but to evaluate the procedure and to establish whether the intended approach to investigate memory detail and affective responses is feasible²⁴. For this reason, the pilot data also did not distinguish between people with and without dysphoria. That is, we do not consider the pilot data to be suitable to evaluate any of the hypotheses. Any conclusions should be driven by the confirmatory analyses presented in the results of the stage 2 report.

The observation that the SD is higher than the mean is not in itself informative because the SD and the mean describes different features of the data. To illustrate this, if one standardizes data, the mean is often centred around 0, but the standard deviation will be larger than zero (e.g., after a z-transformation).

Based on these considerations, we did not elaborate on the difference between pilot and registered report data, but rather regard the pilot result as a chance finding based on a small sample that should be disregarded.

Nevertheless, in order to further address the reviewer's reservations regarding reliability, we calculated permutation-based split half reliability of fEMG responses to memories²⁵. Using 5000 random splits, the Spearman-Brown corrected reliability estimate for zygomaticus responses to positive memories was .76 for people without dysphoria and .90 for people with dysphoria. The Spearman-Brown corrected reliability estimate for corrugator responses to negative memories was .80 for people without dysphoria and .81 for people with dysphoria. These estimates indicate at least acceptable reliability in all groups and conditions. Notably, we would not expect much higher a reliability estimates (and definitely not reliability estimates close to 1) because that would mean that there is no variation in responses that could be associated with other variables. For instance, the exact same zygomaticus response to every positive memory would be unlikely, also because memories differ in emotionality. In sum, we do not see reasons for concern regarding the reliability of the fEMG data.

Regarding imagery, the manual that you used to score internal details includes markers such as emotion details, sensory detail, and specificity of descriptions and adjectives, which are commonly used to index the strength of imagery. The argument in the Discussion that imagery may have a stronger effect, relative to the episodic details you have analysed and show a lack of effect for, is therefore not well supported.

We have removed the argument regarding imagery from the discussion and generally shortened the tentative clinical implications of the study.

Reviewer #3 (Remarks to the Author):

My primary - and single - task as a reviewer was to evaluate the method of facial EMG recording and quantification within this study.

My conclusion is short and clear: both recording and quantification were performed excellently.

I have only a minor remark concerning the reference list: the last name of the author of study #108 should be "van Boxtel" rather than "Boxtel".

We are grateful for the positive evaluation and adjusted the reference.

Reviewer #4 (Remarks to the Author):

It is great to see this completed registered report. The evidence is strong, showing large Bayes factors and fairly decisive posterior probabilities overall. The Bayesian analyses are clearly described, and in line with everything initially proposed. The exploratory analyses are also thorough.

I think this will be a very well-regarded paper.

We thank the reviewer for their positive feedback.

References

1. Williams, J. M. G. *et al.* Autobiographical memory specificity and emotional disorder. *Psychological Bulletin* **133**, 122–148 (2007).
2. Holmes, E. A., Blackwell, S. E., Burnett Heyes, S., Renner, F. & Raes, F. Mental Imagery in Depression: Phenomenology, Potential Mechanisms, and Treatment Implications. *Annual Review of Clinical Psychology* **12**, 249–280 (2016).
3. Dillon, D. G. & Pizzagalli, D. A. Mechanisms of Memory Disruption in Depression. *Trends in Neurosciences* **41**, 137–149 (2018).
4. Arditte Hall, K. A., De Raedt, R., Timpano, K. R. & Joormann, J. Positive memory enhancement training for individuals with major depressive disorder. *Cognitive Behaviour Therapy* **47**, 155–168 (2018).
5. Söderlund, H. *et al.* Autobiographical episodic memory in major depressive disorder. *Journal of Abnormal Psychology* **123**, 51–60 (2014).
6. Hallford, D. J. *et al.* Specificity and detail in autobiographical memory retrieval: a multi-site (re)investigation. *Memory* **29**, 1–10 (2021).
7. Levine, B., Svoboda, E., Hay, J. F., Winocur, G. & Moscovitch, M. Aging and autobiographical memory: Dissociating episodic from semantic retrieval. *Psychology and Aging* **17**, 677–689 (2002).
8. Pile, V., Williamson, G., Saunders, A., Holmes, E. A. & Lau, J. Y. F. Harnessing emotional mental imagery to reduce anxiety and depression in young people: an integrative review of progress and promise. *The Lancet Psychiatry* **8**, 836–852 (2021).
9. Pile, V. *et al.* A feasibility randomised controlled trial of a brief early intervention for adolescent depression that targets emotional mental images and memory specificity (IMAGINE). *Behaviour Research and Therapy* **143**, 103876 (2021).
10. Raes, F., Hermans, D., De Decker, A., Eelen, P. & Williams, J. M. G. Autobiographical memory specificity and affect regulation: An experimental approach. *Emotion* **3**, 201–206 (2003).

11. Barry, T. J. *et al.* Autobiographical memory and psychopathology: Is memory specificity as important as we make it seem? *WIREs Cognitive Science* 1–15 (2022) doi:10.1002/wcs.1624.
12. Griffith, J. W. *et al.* An item response theory/confirmatory factor analysis of the autobiographical memory test. *Memory* **17**, 609–623 (2009).
13. Debeer, E., Hermans, D. & Raes, F. Associations between components of rumination and autobiographical memory specificity as measured by a Minimal Instructions Autobiographical Memory Test. *Memory* **17**, 892–903 (2009).
14. Ford, J. H., Morris, J. A. & Kensinger, E. A. Effects of Emotion and Emotional Valence on the Neural Correlates of Episodic Memory Search and Elaboration Jaclyn. *Journal of Cognitive Neuroscience* **26**, 825–839 (2014).
15. McCormick, C., St-Laurent, M., Ty, A., Valiante, T. A. & McAndrews, M. P. Functional and effective hippocampal-neocortical connectivity during construction and elaboration of autobiographical memory retrieval. *Cerebral Cortex* **25**, 1297–1305 (2015).
16. Dalgleish, T. & Hitchcock, C. Transdiagnostic distortions in autobiographical memory recollection. *Nat Rev Psychol* (2023) doi:10.1038/s44159-023-00148-1.
17. Everaert, J., Bernstein, A., Joormann, J. & Koster, E. H. W. Mapping Dynamic Interactions Among Cognitive Biases in Depression. *Emotion Review* **12**, 93–110 (2020).
18. Bylsma, L. M., Morris, B. H. & Rottenberg, J. A meta-analysis of emotional reactivity in major depressive disorder ☆. *Clinical Psychology Review* **28**, 676–691 (2008).
19. Dalgleish, T. & Werner-Seidler, A. Disruptions in autobiographical memory processing in depression and the emergence of memory therapeutics. *Trends in Cognitive Sciences* **18**, 596–604 (2014).
20. Bower, G. H. Mood and Memory. *American Psychologist* **36**, 129–148 (1981).
21. Walker, W. R., Skowronski, J. J., Gibbons, J. A., Vogl, R. J. & Thompson, C. P. On the emotions that accompany autobiographical memories: Dysphoria disrupts the fading affect bias. *Cognition and Emotion* **17**, 703–723 (2003).

22. Marsh, C., Hammond, M. D. & Crawford, M. T. Thinking about negative life events as a mediator between depression and fading affect bias. *PLoS ONE* **14**, (2019).
23. Duken, S. B., Neumayer, F., Kindt, M., Oosterwijk, S. & Ast, V. van. Reliving emotional memories: Episodic recollection elicits affective psychophysiological responses. Preprint at <https://doi.org/10.31234/osf.io/ukt5x> (2021).
24. Hoijtink, H. *et al.* The Open Empirical Cycle for Hypothesis Evaluation in Psychology. Preprint at <https://doi.org/10.31234/osf.io/wsxbh> (2023).
25. Parsons, S., Kruijt, A.-W. & Fox, E. Psychological Science Needs a Standard Practice of Reporting the Reliability of Cognitive-Behavioral Measurements. *Advances in Methods and Practices in Psychological Science* **2**, 378–395 (2019).